# SpatialViz-Bench: A Cognitively-Grounded Benchmark for Diagnosing Spatial Visualization in MLLMs

**Siting Wang**[1,2,3] **Minnan Pei**[1,2] **Luoyang Sun**[1,2,3] **Cheng Deng**[4,†] **Yuchen Li**[5]
**Kun Shao**[4] **Zheng Tian**[6] **Haifeng Zhang**[1,†] **Jun Wang**[7,†]

[1]The Key Laboratory of Cognition and Decision Intelligence for Complex Systems
Institute of Automation, Chinese Academy of Sciences
[2]University of Chinese Academy of Sciences
[3]AI Lab, The Yangtze River Delta
[4]Huawei Noah's Ark, UK
[5]Shanghai Jiao Tong University
[6]School of Creativity and Art, ShanghaiTech University
[7]University College London

## Abstract

Humans can imagine and manipulate visual images mentally, a capability known as *spatial visualization*. While many multi-modal benchmarks assess reasoning on visible visual information, the ability to infer unseen relationships through spatial visualization remains insufficiently evaluated as a spatial skill. This reliance on publicly sourced problems from IQ tests or math competitions risks data contamination and compromises assessment reliability. To this end, we introduce *SpatialViz-Bench*, a comprehensive multi-modal benchmark for *spatial visualization* with *12* tasks across *4* sub-abilities, comprising *1,180* programmatically generated problems, a scalable framework that allows for expansion to ensure fair and continuously reliable evaluations. Our evaluation of *27* Multi-modal Large Language Models (MLLMs) reveals wide performance variations, demonstrates the benchmark's strong discriminative power, and uncovers counter-intuitive findings: Chain-of-Thought (CoT) prompting paradoxically degrades accuracy on open-source models. Through statistical and qualitative analysis of error types, SpatialViz-Bench demonstrates that state-of-the-art MLLMs exhibit deficiencies in *spatial visualization* tasks, thereby addressing a significant lacuna in the field. The benchmark data and code are publicly available. [1]

## 1 Introduction

Large Language Models (LLMs) have demonstrated strong capabilities in complex reasoning, and the integration of Vision Transformers (ViTs) has given them "eyes", extending these abilities into the multimodal domain. While many tasks focus on *visible* information, real-world challenges in fields like architectural design and medical-image–assisted surgery often demand the ability to mentally construct and manipulate *unseen* structures, a capability in which existing MLLMs still struggle. To bridge this gap, *spatial visualization* must be abstracted and assessed through targeted evaluations that isolate it from confounding factors, like a well-designed physics exam tests fundamental principles. However, current evaluations rely heavily on web-sourced problems, risking data leakage and inconsistent formulations, underscoring the need for a procedurally generated, standardized benchmark to ensure fair and reliable assessment.

This cognitive faculty for mental manipulation is known as *spatial visualization*, which was first identified by Thurstone in his work on primary mental abilities (Thurstone, 1938). Successfully

---

[†]Correspondence to Cheng Deng, Haifeng Zhang Jun Wang.
[1]Data: https://huggingface.co/datasets/PLM-Team/Spatial-Visualization-Benchmark
  Code: https://github.com/wangst0181/SpatialViz-Bench

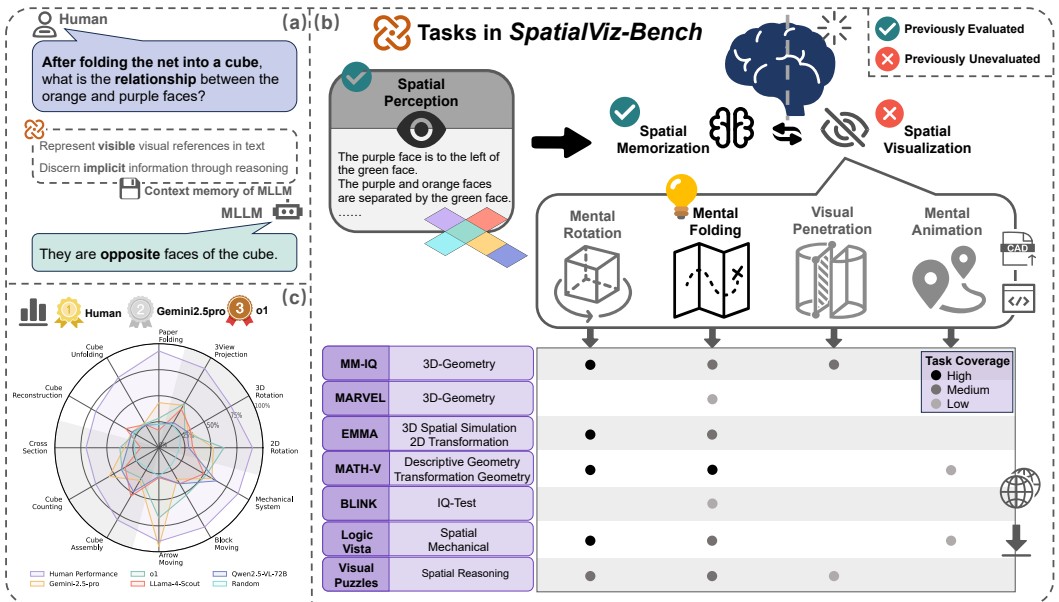

Figure 1: **Overview of SpatialViz-Bench.** (a) presents a representative task instance. (b) unfolds the reasoning behind (a): perceiving visible cues to infer unseen relationships via iterative visualization and memorization. The table highlights a systematic gap: unlike perception, *spatial visualization* remains a largely unassessed blind spot in prior benchmarks (indicated by lighter colors). (c) displays zero-shot accuracy revealing significant gaps against human performance.

performing spatial visualization tasks relies on two other fundamental spatial abilities: *Spatial perception* (Thurstone, 1950), which aims to perceive external spatial information and relationships, and *spatial memorization* (Della Sala et al., 1999), which requires temporarily storing transformation information mentally without accessing physical objects.

Despite their importance as dedicated spatial-reasoning challenges, *spatial visualization* tasks are often buried under broader categories like mathematical or logical reasoning, appearing as multimodal puzzles or 3D geometry problems. This categorization obscures the evaluation of *spatial visualization* as a distinct capability and focuses on "solving" a problem rather than driving research toward core spatial abilities. Moreover, most examples are drawn from publicly available sources, online IQ tests, administrative exams, and math contests, which risks overlap between training and evaluation data and undermines reliability. The scarcity of items per subskill also magnifies random error, while heterogeneous formats make it hard to distinguish true reasoning failures from misinterpretation. Consequently, even with potential pretraining exposure, performance remains poor. State-of-the-art systems score just *27.64* on 3D Geometry in MM-IQ (Cai et al., 2025) and *26.00* on Descriptive Geometry in MathVision (Wang et al., 2024). Beyond task difficulty, the modern paradigm of pretraining on vast, scraped internet data fundamentally challenges evaluation validity (Wu et al., 2025), a problem exacerbated by proprietary datasets that make auditing for contamination impossible. This fundamental challenge calls for a new generation of benchmarks with dynamically updatable test banks to ensure persistent evaluation integrity (Ni et al., 2025).

To address these shortcomings, we introduce ***SpatialViz-Bench***, a novel benchmark designed to formally evaluate the *spatial visualization* capabilities of MLLMs, comprising a framework of *4* key sub-abilities(mental rotation, mental folding, visual penetration, and mental animation) from which *12* targeted tasks are designed for comprehensive assessment. Inspired by benchmarks like CLEVR (Johnson et al., 2017), a diagnostic benchmark for *spatial perception*, which uses Blender (Blender Online Community, 2016) for data generation, we developed a pipeline that integrates Python with FreeCAD (FreeCAD Team, 2025) for the programmatic generation of novel test cases, enabling scalable task expansion while effectively preventing data contamination by dynamically updating the test bank through randomized generation. We employ standardized question templates to minimize errors arising from varied instructions. Furthermore, programmatic generation allows us to control task difficulty precisely and to create distractors with explanations systematically.

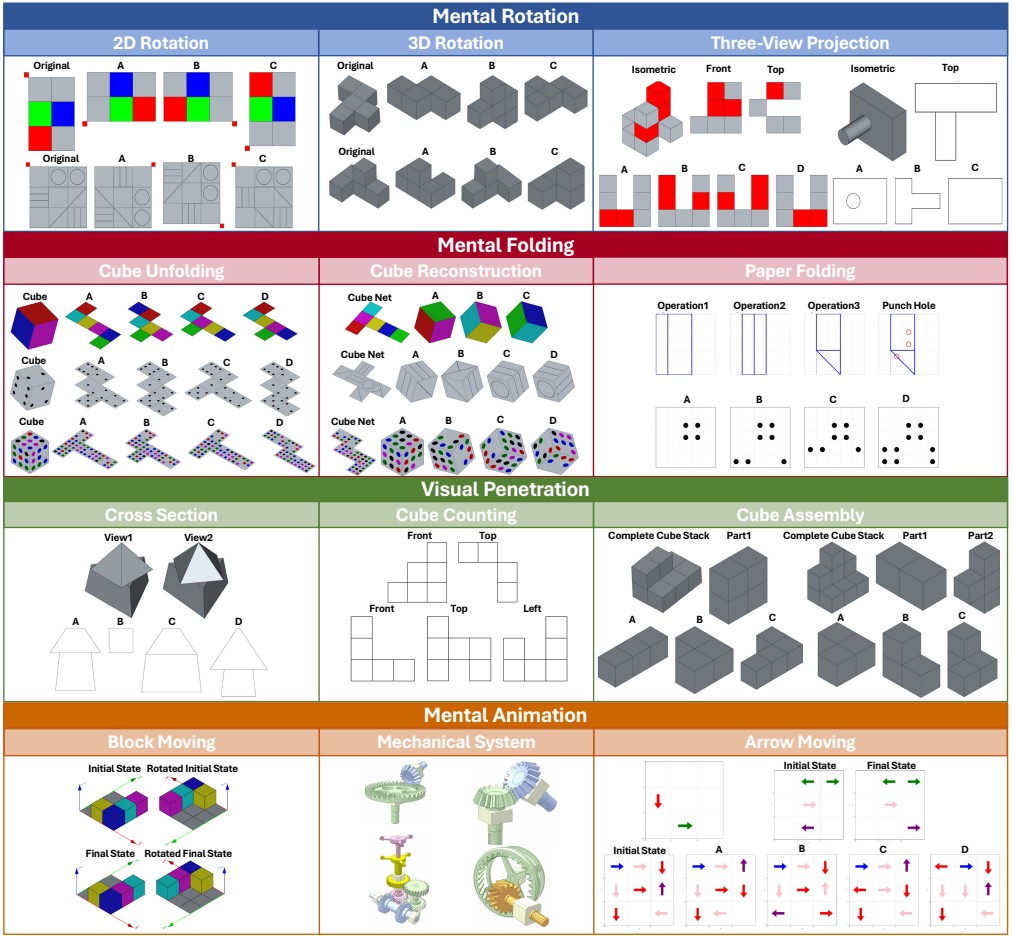

Figure 2: **Overview of Tasks in SpatialViz-Bench.** SpatialViz-Bench evaluates 4 spatial sub-abilities, mental rotation, mental folding, visual penetration, and mental animation, via 3 tasks each (12 tasks total). Each task has 2–3 difficulty levels of 40–50 cases, yielding 1,180 question–answer pairs.

Models with strong *spatial visualization* skills can serve as an **efficient internal world model**, providing a foundational capability for various downstream applications. This allows a model to run fast, lightweight internal "what-if" scenarios (e.g., "what happens if I rotate this object?", "if this gear turns clockwise, which way will the connected gear move?") to predict the outcome of actions. This is far more efficient than the current alternative of invoking large, diffusion-based video generation models to explicitly render a future state.

The main contributions of our work can be listed as follows:

- We introduce *SpatialViz-Bench*, the first benchmark to formally establish a comprehensive and challenging evaluation framework for *spatial visualization*, a core yet long-overlooked cognitive ability. It is grounded in cognitive science and assesses *4* key sub-abilities through *12* distinct tasks, resulting in a total of *1,180* examples across parameter-controlled difficulty levels.

- We establish a scalable and trustworthy programmatic generation methodology for *11* of our tasks. This approach not only enables continuous expansion of tasks but also sets a new standard for fair evaluation by preventing data contamination through dynamic updates to the test bank.

- We systematically evaluate *27* MLLMs, with top scores from Gemini-2.5-pro (*44.66%*) and o1 (*41.36%*). These results demonstrate the benchmark's challenge and high discriminative power, revealing a significant capability gap to human performance.

- We conduct a diagnostic analysis revealing that model failures stem primarily from fundamental Perceptual and Spatial Transformation deficits, rather than from high-level reasoning, which offers a clear direction for future improvements.

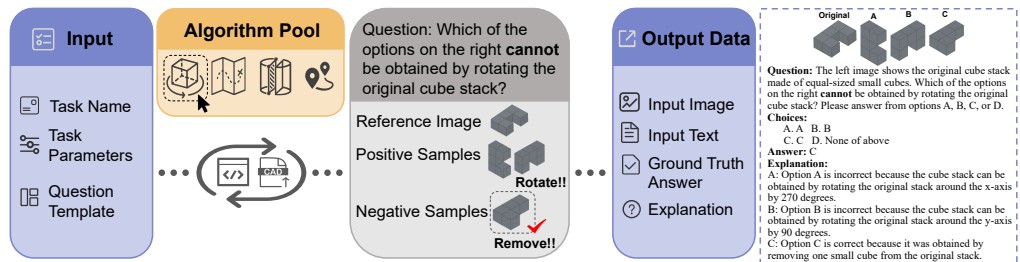

Figure 3: **The programmatic generation pipeline of a data instance.** We constructed the dataset using an programmatic generation system that integrates Python with FreeCAD, enabling precise control of difficulty, systematic generation of distractor options, and programmatic recording of explanations for incorrect choices.

## 2 RELATED WORKS

**Current Landscape in Spatial Reasoning Benchmarks** The evaluation of spatial reasoning in MLLMs has largely concentrated on abilities tied to directly observable information. Benchmarks for *spatial perception*, the ability to identify and interpret spatial relationships from visual input, are the most established. Existing benchmarks like What'sUp (Kamath et al., 2023), Blink (Fu et al., 2024), and SpatialRGPT-bench (Cheng et al., 2024) assess how models understand object- or camera-centric relationships, relative distances, sizes, and positions. Progress has also been made in evaluating *spatial memorization*, with video-based benchmarks like VCBench (Li et al., 2024) and VSI-bench (Yang et al., 2024b) challenging models to track objects in dynamic scenes. These efforts have built a foundation for assessing a type of spatial reasoning that relies on explicit visual information and applies a model's world knowledge to interpret what is perceived. However, they largely neglect the advanced capability of *spatial visualization*, the ability to infer implicit visual-spatial information through transformation of structures derived from visible inputs, leaving a significant gap in the current evaluation landscape.

**Evaluation of Spatial Visualization** Evaluating *spatial visualization* presents challenges regarding data contamination, obscured categorization, and narrow task coverage. A primary concern is contamination from public sources (Xu et al., 2025b), a risk programmatic generation seeks to mitigate, as seen in the LEGO-Puzzles benchmark (Tang et al., 2025). Furthermore, *spatial visualization* is often subject to obscured categorization, subsumed under broader domains like mathematical or logical reasoning in general benchmarks (e.g., MM-IQ (Cai et al., 2025), MathVision (Wang et al., 2024)), which diverts focus from it as a core ability. Concurrently, specialized datasets exhibit narrow task coverage, focusing on single sub-skills like mental rotation (SPARE3D (Han et al., 2020), CLEVR-MRT (Beckham et al., 2023)) or specific tasks like paper folding (SRBench (Stogiannidis et al., 2025)). Yin et al. (2025) also assess mental modeling, utilizing distinct organizational frameworks, such as relative spatial perspectives.

## 3 SPATIALVIZ-BENCH

### 3.1 SPATIAL VISUALIZATION

*Spatial visualization* is a core component of human cognitive systems and a critical capability for deployment in downstream applications. Research into this ability began with Thurstone (Thurstone, 1938), who defined it as performing mental operations on visual images and identified it as one of the key spatial factors: *spatial perception*, *spatial visualization*, and mental rotation (Thurstone, 1950).

Building on this foundation, we establish a cognitive framework that decomposes spatial visualization tasks into two phases: **observing visible information** and **discerning implicit information**. The former requires basic *spatial perception*, while the latter demands an alternation between *spatial visualization* (mentally manipulating images to find implicit information) and *spatial memorization* (temporarily storing visuospatial information) (Della Sala et al., 1999).

Our benchmark's design is guided by 4 core sub-abilities: 1) **mental rotation**: Mentally representing and rotating objects while maintaining their features; 2) **mental folding**: Mentally folding 2D patterns

Table 1: A Compact Summary of Spatial Reasoning Tasks.

| Category | Task Name | Core Objective | Negative Samples | Difficulty Scaling |
|---|---|---|---|---|
| **Mental Rotation** | 2D Rotation | Identify correct 2D rotation | Mirroring; internal pattern rotation | Non-centrally symmetric patterns |
| | 3D Rotation | Identify correct 3D rotation | View mirroring cube removal | Larger assemblies |
| | Three-View Projection | Select left view from projections | Wrong view substitution; view flipping; line deletion | Real engineering parts (DeepCAD (Wu et al., 2021)) |
| **Mental Folding** | Paper Folding | Predict unfolded hole pattern | Hole mirroring, addition, deletion, or relocation | More folds; larger grid; more holes |
| | Cube Unfolding | Select correct 2D net from view | Swapping face colors; rotating internal patterns | Asymmetric/dot patterns on faces |
| | Cube Reconstruction | Select 3D view from net; Find opposite face | Mirroring the correct 3D view | Follows Cube Unfolding |
| **Visual Penetration** | Cross-Section | Identify cross-section of solid | Altered geometric proportions | 3-solid composites; oblique slicing |
| | Cube Counting | Infer total cube count from views | Options from min/max math bounds | 2 to 3 views; larger assemblies |
| | Cube Assembly | Find complementary part of split stack | Add/remove cubes from correct part | Larger stacks; 3-part splits |
| **Mental Animation** | Arrow Moving | Predict final state or movement sequence | Incorrect endpoint from same start | Multiple arrows; interaction rules |
| | Block Moving | Predict final state with gravity | Incorrect final states | Higher complexity; longer sequences |
| | Mechanical System | Understand motion propagation | Incorrect motion outcomes | More system modules |

into 3D objects or vice versa (Glass et al., 2013); 3) **visual penetration**: Imagining the internal structure of an object from its external features (Titus & Horsman, 2009); 4) **mental animation**: Mentally visualizing the motion of components within a system (Sims & Hegarty, 1997).

## 3.2 OVERVIEW OF SPATIALVIZ-BENCH

Stemming from an availability-driven collection, current web-sourced benchmarks containing *spatial visualization* tasks lack standardization and a cognitive theory basis, resulting in inconsistent tasks and incomplete coverage. We counter this with a systematic, ability-centric methodology: we use a hierarchical framework based on cognitive principles to guide new task design and employ a unified input format with standardized templates to reduce confounds and enable fine-grained error analysis.

Based on our cognitive framework, we propose ***SpatialViz-Bench*** to comprehensively evaluate the *spatial visualization* capabilities of MLLMs. It is organized around *4* core sub-abilities—mental rotation, mental folding, visual penetration, and mental animation—with *3* assessment tasks designed for each, totaling *12* tasks. Each task includes *2* to *3* difficulty levels, with each level containing *40* or *50* test cases, comprising *1,180* question-answer pairs in total, mostly with image-based options to focus on visual reasoning. Further details on the dataset characteristics are provided in Appendix C.

## 3.3 CONSTRUCTION OF SPATIALVIZ-BENCH

*SpatialViz-Bench* is constructed through a combination of programmatic generation and manual design. For 11 of the tasks, we used a programmatic system integrating Python with FreeCAD (FreeCAD Team, 2025) (see Figure 3). By explicitly utilizing cognitive load parameters rather than heuristics, such as aligning rotational complexity (global object vs. internal pattern rotation) with mental transformation steps (Shepard & Metzler, 1971), our programmatic framework ensures precise difficulty control, while employing controlled randomness to enhance diversity and generate distractor options with explanations for deep diagnostics. Notably, the Three-View Projection task (Level 1) uses fixed DeepCAD (Wu et al., 2021) models, but we programmatically generate novel distractors (e.g., random line deletion, view flipping) to ensure novelty. Conversely, the Mechanical System task (1/12) was manually designed, as programmatic, physically-consistent generation was technically difficult. Using representative public simulations as a reference, experts designed all questions from scratch. These visual-based questions probe dynamic motion propagation (e.g., rotational dynamics from a single image), testing visual simulation rather than caption recall or theoretical derivation.

This combined methodology, leveraging both programmatic generation and the vast pool of public simulations for expert-driven question design, supports a dynamically updated test bank that proactively mitigates data contamination. A task summary is presented in Table 1, with detailed generation processes, algorithmic pseudocode, and illustrative examples deferred to Appendix B.1, B.4 and D.

Table 2: Comparison of open-source model performances. Tasks: 2D Rotation (2DR), 3D Rotation (3DR), Three-View Projection (3VP), Paper Folding (PF), Cube Unfolding (CU), Cube Reconstruction (CR), Cross-Section (CS), Cube Counting (CC), Cube Assembly (CA), Arrow Moving (AM), Block Moving (BM), Mechanical System (MS). The first and second highest accuracy of MLLMs are marked in red and blue, with open-source and closed-source models marked separately.

| Model | Overall | | Mental Rotation | | | | Mental Folding | | | | Visual Penetration | | | | Mental Animation | | | |
|---|---|---|---|---|---|---|---|---|---|---|---|---|---|---|---|---|---|---|
| | w/o CoT | w/ CoT | 2DR | 3DR | 3VP | Avg | PF | CU | CR | Avg | CS | CC | CA | Avg | AM | BM | MS | Avg |
| Human | - | 82.46 | 90.00 | 79.16 | 87.50 | 85.56 | 93.75 | 75.00 | 72.92 | 80.56 | 72.92 | 70.83 | 82.50 | 75.42 | 90.00 | 87.50 | 87.50 | 88.33 |
| Random | - | 25.08 | 23.75 | 27.50 | 31.00 | 27.69 | 19.17 | 20.00 | 25.83 | 21.67 | 30.00 | 25.00 | 30.00 | 28.12 | 28.75 | 16.25 | 25.00 | 23.33 |
| Qwen2.5-72B-Instruct(Text-only) | - | 25.86 | 15.00 | 35.00 | 15.00 | 21.67 | 23.33 | 16.67 | 26.67 | 22.22 | 20.00 | 33.33 | 45.00 | 31.25 | 25.00 | 30.00 | 30.00 | 28.33 |
| Open Source MLLMs | | | | | | | | | | | | | | | | | | |
| 3B | | | | | | | | | | | | | | | | | | |
| SAIL-VL-1.5-2B | 29.32 | 24.15 | 22.50 | 22.50 | 22.00 | 22.31 | 20.00 | 27.50 | 20.00 | 22.50 | 24.17 | 26.67 | 32.50 | 27.19 | 21.25 | 25.00 | 27.50 | 24.58 |
| InternVL3-2B | - | 26.19 | 16.25 | 33.75 | 31.00 | 27.31 | 22.50 | 25.83 | 25.00 | 24.44 | 20.00 | 30.83 | 30.00 | 26.56 | 18.75 | 32.50 | 30.00 | 27.08 |
| Deepseek-VL2-tiny(3B) | 29.58 | 21.36 | 17.50 | 22.50 | 27.00 | 22.69 | 21.67 | 20.83 | 19.17 | 20.56 | 20.83 | 22.50 | 18.75 | 20.94 | 18.75 | 21.25 | 25.00 | 21.67 |
| Qwen2.5-VL-3B-Instruct | 30.17 | 26.10 | 20.00 | 18.75 | 21.00 | 20.00 | 25.00 | 25.83 | 21.67 | 24.17 | 25.83 | 23.33 | 30.00 | 25.94 | 35.00 | 30.00 | 42.50 | 35.83 |
| 7B | | | | | | | | | | | | | | | | | | |
| Qwen2.5-VL-7B-Instruct | 30.76 | 27.97 | 25.00 | 16.25 | 29.00 | 23.85 | 34.17 | 21.67 | 30.00 | 28.61 | 16.67 | 36.67 | 28.75 | 27.19 | 22.50 | 23.75 | 51.25 | 32.50 |
| Qwen2.5-Omni-7B | 31.44 | 27.29 | 22.50 | 20.00 | 29.00 | 24.23 | 25.00 | 27.50 | 20.00 | 24.17 | 20.83 | 33.33 | 27.50 | 27.19 | 31.25 | 30.00 | 45.00 | 35.42 |
| SAIL-VL-1.6-8B | 29.15 | 25.00 | 18.75 | 21.25 | 25.00 | 21.92 | 28.33 | 25.00 | 18.33 | 23.89 | 21.67 | 19.17 | 23.75 | 21.25 | 25.00 | 35.00 | 45.00 | 35.00 |
| InternVL3-8B | 30.25 | 30.08 | 20.00 | 38.75 | 28.00 | 28.85 | 28.33 | 23.33 | 25.00 | 25.56 | 15.83 | 40.83 | 38.75 | 30.94 | 30.00 | 30.00 | 51.25 | 37.08 |
| 16B | | | | | | | | | | | | | | | | | | |
| Kimi-VL-A3B-Instruct(16B) | 32.37 | 23.90 | 16.25 | 30.00 | 36.00 | 28.08 | 25.83 | 20.00 | 26.67 | 24.17 | 21.67 | 5.00 | 28.75 | 17.19 | 15.00 | 31.25 | 37.50 | 27.92 |
| Kimi-VL-A3B-thinking(16B) | - | 28.14 | 13.75 | 20.00 | 25.00 | 20.00 | 23.33 | 24.17 | 26.67 | 24.72 | 25.00 | 36.67 | 25.00 | 29.38 | 30.00 | 43.75 | 47.50 | 40.42 |
| Deepseek-VL2-small(16B) | 25.17 | 25.17 | 31.25 | 16.25 | 26.00 | 24.62 | 22.50 | 25.00 | 26.67 | 24.72 | 9.17 | 35.00 | 35.00 | 25.31 | 26.25 | 23.75 | 28.75 | 26.25 |
| 32B | | | | | | | | | | | | | | | | | | |
| Deepseek-VL2(27B) | 30.08 | 28.31 | 25.00 | 33.75 | 30.00 | 29.62 | 31.67 | 25.00 | 22.50 | 26.39 | 18.33 | 39.17 | 28.75 | 28.75 | 26.25 | 30.00 | 31.25 | 29.17 |
| Qwen2.5-VL-32B-Instruct | 33.90 | 32.12 | 31.25 | 35.00 | 38.00 | 35.00 | 21.67 | 25.00 | 27.50 | 24.72 | 25.83 | 36.67 | 43.75 | 34.38 | 28.75 | 27.50 | 55.00 | 37.08 |
| InternVL3-38B | 29.75 | 30.34 | 22.50 | 33.75 | 29.00 | 28.46 | 20.83 | 29.17 | 30.83 | 26.94 | 21.67 | 32.50 | 41.25 | 30.63 | 25.00 | 30.00 | 56.25 | 37.08 |
| 72B | | | | | | | | | | | | | | | | | | |
| Qwen2.5-VL-72B-Instruct | 35.00 | 33.31 | 28.75 | 31.25 | 28.00 | 29.23 | 22.50 | 20.00 | 30.00 | 24.17 | 30.00 | 41.67 | 48.75 | 39.06 | 27.50 | 40.00 | 63.75 | 43.75 |
| QvQ-72B-preview | - | 28.14 | 21.25 | 30.00 | 31.00 | 27.69 | 16.67 | 19.17 | 27.50 | 21.11 | 30.00 | 22.50 | 32.50 | 27.81 | 25.00 | 50.00 | 43.75 | 39.58 |
| InternVL3-78B | 32.29 | 29.75 | 25.00 | 25.00 | 34.00 | 28.46 | 19.17 | 25.00 | 22.50 | 22.22 | 20.83 | 40.00 | 48.75 | 35.00 | 23.75 | 41.25 | 41.25 | 35.42 |
| 108B | | | | | | | | | | | | | | | | | | |
| Llama-4-Maverick-17B-128E-Instruct | - | 31.78 | 20.00 | 40.00 | 40.00 | 33.85 | 16.67 | 29.17 | 29.17 | 25.00 | 19.17 | 35.00 | 47.50 | 32.19 | 35.00 | 40.00 | 42.50 | 39.17 |
| LLama-4-Scout-17B-16E-Instruct | - | 34.24 | 32.50 | 35.00 | 43.00 | 37.31 | 16.67 | 32.50 | 36.67 | 28.61 | 17.50 | 37.50 | 53.75 | 34.06 | 28.75 | 40.00 | 50.00 | 39.58 |
| Closed Source MLLMs | | | | | | | | | | | | | | | | | | |
| GPT-4o | 30.76 | 31.10 | 32.50 | 27.50 | 33.00 | 31.15 | 29.17 | 15.83 | 30.00 | 25.00 | 19.17 | 40.83 | 40.00 | 32.50 | 22.50 | 32.50 | 60.00 | 38.33 |
| o1 | - | 41.36 | 62.50 | 28.75 | 49.00 | 46.92 | 28.33 | 34.17 | 26.67 | 29.72 | 37.50 | 40.83 | 33.75 | 37.81 | 67.50 | 52.50 | 52.50 | 57.50 |
| Claude-3.5-sonnet | 26.86 | 32.54 | 31.25 | 25.00 | 45.00 | 34.62 | 20.83 | 22.50 | 31.67 | 25.00 | 22.50 | 35.83 | 46.25 | 33.44 | 37.50 | 31.25 | 52.50 | 40.42 |
| Claude-3.7-sonnet | - | 33.90 | 32.50 | 36.25 | 44.00 | 38.08 | 18.33 | 26.67 | 29.17 | 24.72 | 24.17 | 30.83 | 43.75 | 31.56 | 66.25 | 28.75 | 43.75 | 46.25 |
| Gemini-2.5-flash | - | 36.86 | 42.50 | 30.00 | 35.00 | 35.83 | 26.67 | 30.00 | 40.83 | 32.50 | 30.00 | 38.33 | 28.75 | 32.81 | 67.50 | 33.75 | 48.75 | 50.00 |
| Gemini-2.5-pro | - | 44.66 | 52.50 | 32.50 | 47.00 | 44.23 | 43.33 | 31.67 | 30.00 | 35.00 | 33.33 | 55.00 | 36.25 | 42.19 | 95.00 | 35.00 | 58.75 | 62.92 |
| Doubao-1-5-vision-pro | 37.54 | 33.31 | 7.50 | 35.00 | 45.00 | 30.38 | 31.67 | 23.33 | 29.17 | 28.06 | 30.00 | 55.83 | 30.00 | 39.69 | 22.50 | 37.50 | 47.50 | 35.83 |
| Qwen-VL-max | 36.10 | 32.03 | 23.75 | 26.25 | 33.00 | 28.08 | 24.17 | 17.50 | 31.67 | 24.44 | 26.67 | 47.50 | 42.50 | 38.44 | 26.25 | 36.25 | 55.00 | 39.17 |

## 4 EVALUATION

### 4.1 EVALUATION SETUP

**Models** We conducted comprehensive experiments on a diverse range of MLLMs, including 8 closed-source and 19 open-source models. For **closed-source MLLMs**, we evaluated models from 5 major providers, including OpenAI series (GPT-4o (Hurst et al., 2024), o1 (Jaech et al., 2024)), Gemini series (Gemini-2.5-flash, Gemini-2.5-pro (Deepmind, 2025)), Claude series (Claude-3.5-sonnet (Anthropic, 2024), Claude-3.7-sonnet (Anthropic, 2025)), Qwen-VL-max (Bai et al., 2023), and Doubao-1.5-vision-pro (ByteDance, 2025). For **open-source MLLMs**, we assessed Qwen2.5-VL series (Bai et al., 2025), QvQ (Qwen Team, 2024), Qwen-Omni (Xu et al., 2025a), InternVL-3 series (Zhu et al., 2025), Deepseek-VL2 series (Wu et al., 2024), SAIL-VL series (Dong et al., 2025), Kimi-VL-A3B series (Team et al., 2025) and LLama-4 series (Meta AI, 2025). For **text-only LLM**, we used Qwen2.5-72B-Instruct (Yang et al., 2024a).

**Setting** For a rigorous evaluation, all experiments were performed in a zero-shot setting (Hao et al., 2025; Wang et al., 2024), comparing model performance under two prompting schemes: (1) CoT, where prompts were designed to encourage models to output their reasoning process before the final answer, and (2) Direct Answering (non-CoT), where prompts solicited the answer directly (see Appendix E.2). This methodology enabled us to not only assess the accuracy of responses but also gain deeper insights into the models' underlying reasoning mechanisms across our benchmark tasks.

**Metric Design** To evaluate models handling multimodal inputs and generating textual outputs, with most options presented as images, we formatted all tasks as Multiple-Choice Answer (MCA) with one correct answer. Option and reference images were integrated into a unified visual input. For

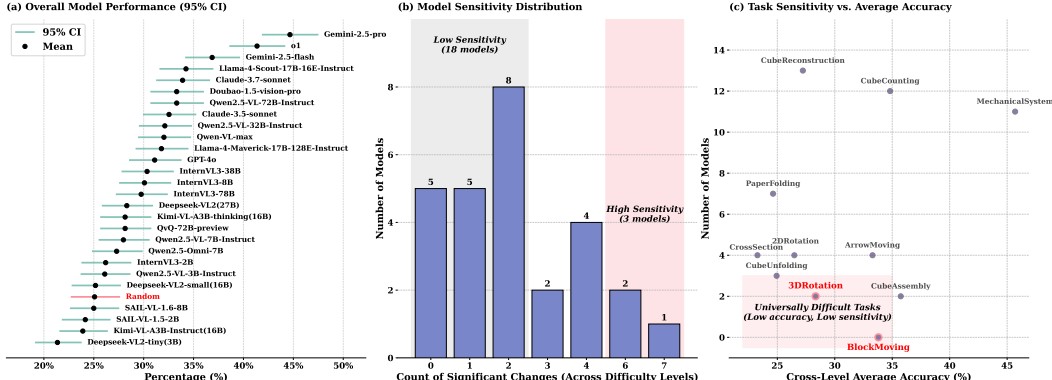

Figure 4: **Statistical Analysis of Model Performance, Difficulty Sensitivity, and Task Discriminability.** (a) presents the overall model performance with 95% Wilson confidence intervals. (b) shows the distribution of model sensitivity to difficulty gradients. (c) provides a task-centered analysis of difficulty sensitivity, revealing how difficulty levels differentiate model capabilities across tasks.

questions where answers could be expressed as simple text, we also provided a text-based answer format (detailed in Appendix E.4). Model performance was assessed using accuracy, based on the match between predicted and ground-truth answers. This standardized approach ensures consistent evaluation across tasks and enables fair comparison of multimodal understanding across models. A comparative analysis of performance on both formats is provided in Appendix F.3.

**Human Baseline** Our human baseline was established with 8 graduate students from mechanical engineering and computer science, selected for their strong spatial reasoning backgrounds. Each participant solved a 72-problem subset under strict conditions designed to be analogous to MLLM evaluation: no external aids (e.g., scratch paper) were allowed, but time was unlimited. This protocol isolates intrinsic spatial visualization abilities for a fair comparison.

## 4.2 EVALUATION RESULTS

This section first establishes the performance gaps between different models and then, through a CoT ablation study, investigates the impact of explicit reasoning to identify the core abilities required for advanced spatial reasoning.

### 4.2.1 MAIN RESULTS

**Tasks in SpatialViz-Bench are Vision-Dependent and Reasoning-Intensive** As the textual input alone is insufficient, visual input is essential for problem-solving, making the benchmark highly vision-dependent. We empirically validated this claim by evaluating a powerful text-only LLM (Qwen2.5-72B-Instruct). As detailed in Table 2, the text-only model achieved a total accuracy of 25.86%, which is negligibly different from the random-chance baseline (25.08%), quantitatively proving that the visual modality is indispensable. Most options are image-based, requiring precise visual analysis rather than simple matching, thereby increasing reasoning complexity. For both humans and MLLMs, these tasks demand multi-step spatial transformations and inferences that mirror complex CoT processes.

**Performance Gaps Reveal a Statistically Validated Hierarchy of MLLMs** All evaluated models performed well below the human baseline (82.46%), underscoring the benchmark's difficulty. Our analysis, now supported by 95% Wilson confidence intervals (CIs) (as shown in **??**), confirms this performance hierarchy is statistically robust. The top performer, Gemini-2.5-pro (44.66%, CI: [41.85%, 47.51%]), demonstrates capabilities irrefutably above the random baseline (25.08%, CI: [22.69%, 27.64%]), as their CIs do not overlap. More importantly, this analysis provides solid statistical backing for the critical capability gap between proprietary and open-source models. The CI for Gemini-2.5-pro shows no overlap with that of the top open-source model, LLaMA-4-Scout (34.24%, CI: [31.58%, 36.99%]), confirming this ∼10% performance delta is significant. Conversely, the CIs help group statistically similar models into "performance tiers"; for example, the CIs for LLaMA-4-Scout and Qwen2.5-VL-72B-Instruct (35.00%, CI: [30.67%, 36.04%]) highly overlap,

Table 3: **Robustness analysis of CoT performance.** (a) Performance remains stable across different CoT prompt templates. (b) The significant performance gap between CoT and non-CoT persists across extraction rules, ruling out parsing failures as the cause of performance drops.

| (a) Sensitivity to Prompt Variations (Accuracy %) | | | | (b) Sensitivity to Extraction Rules (Acc. Drop%) | | | |
|---|---|---|---|---|---|---|---|
| **Model** | **CoT A** | **CoT B** | $\Delta$ | **Model** | **Rule A** $\downarrow$ | **Rule B** $\downarrow$ | $\Delta$ |
| Qwen2.5-VL-72B | 33.31 | 31.19 | -2.12 | SAIL-VL-1.5-2B | -8.22 | -7.29 | +0.93 |
| GPT-4o | 31.10 | 30.81 | -0.29 | Deepseek-VL2-3B | -5.18 | -5.01 | +0.17 |
| Claude-3.5-sonnet | 32.54 | 28.31 | -4.23 | Kimi-VL-16B | -8.47 | -9.66 | -1.19 |

making their performance statistically indistinguishable. This statistically validated discriminative power highlights significant room for improvement.

**Core 3D Visualization Tasks Reveal Common Model Failures** Models with higher overall accuracy generally perform well across individual tasks. Most models show near-random accuracy on core 3D tasks like 3D Rotation, Cube Unfolding & Reconstruction, indicating common and severe perceptual and visualization limitations in 3D space. Both proprietary models perform well on the Arrow Moving task, with Gemini-2.5-pro even surpassing human performance, while most of open-source models perform at near-random levels. This suggests that, despite its relatively low visual complexity, the task requires advanced reasoning—such as understanding object-centered motion—which open-source models still lack. In most cases, model performance matched our expected difficulty levels, though some discrepancies with human perception offer valuable insights for refining task design and guiding future research. Additional evaluation results and task-specific analysis are provided in Appendix F.1.

**Difficulty Collapse Only Visible in Top-Tier Models** We first validated our intended difficulty gradient (DG) against human performance and hypothesized models would show similar scaling. However, data reveals a widespread "performance floor" at L0; 10 models showed $\leq 1$ significant DG, while the top-performing Gemini-2.5-pro was most sensitive (7 DGs) (Figure 4.b). From a task-centric perspective (Figure 4.c), three tasks induced a significant DG in 11 or more models. Notably, the stark DG contrast between CubeReconstruction (12 models) and its symmetric counterpart CubeUnfolding (1 model) suggests models better reason about symmetry from unfolded views. Conversely, BlockMoving (0 DGs) proved challenging at both levels, rendering any drop statistically invisible. Critically, on 3DRotation, the only two models exhibiting a DG were the top-two performers (Gemini-2.5-pro, o1). This confirms our core claim: only top-tier models achieved non-random L0 accuracy, and thus were the only ones capable of showing a statistically significant collapse at L1.

### 4.2.2 CoT PROMPTING ABLATION STUDY

For the non-CoT evaluation, we excluded models designed for extended reasoning (e.g., o1, Gemini-2.5 series) or those unable to adhere to the format (e.g., InternVL3-2B), proceeding only with models that could reliably provide a single-letter answer (detailed in Appendix E.2).

Our ablation study on Chain-of-Thought (CoT) prompting confirms a "CoT paradox," a phenomenon also noted by EMMA (Hao et al., 2025): CoT benefits high-performing closed-source MLLMs but often paradoxically degrades their open-source counterparts. We provide new statistical validation for this. As shown in Table 5 in Appendix F.2, the impact is significantly positive for claude-3.5-sonnet but significantly negative for several leading open-source models.

Crucially, our analysis pinpoints where this degradation occurs. The performance loss for these open-source models is not uniform but is highly concentrated in "pure-visual" spatial tasks (e.g., 3ViewProjection, 3DRotation). This strongly supports our hypothesis: for these models, the mandate to generate explanatory text (CoT) interferes with their native visual-spatial judgment, acting as a cognitive distraction rather than an aid. In contrast, top-tier closed-source models demonstrate superior resistance to this interference, likely due to specialized RL-based reasoning training, allowing them to leverage CoT effectively.

### 4.2.3 ROBUSTNESS TO PROMPTING AND EXTRACTION STRATEGIES

To rule out the possibility that the observed CoT degradation is an artifact of specific prompt engineering or parsing failures, we conducted a sensitivity analysis in Table 3. First, we tested models with an alternative CoT prompt template (detailed in Appendix E.2). As shown in Table 3(a), the

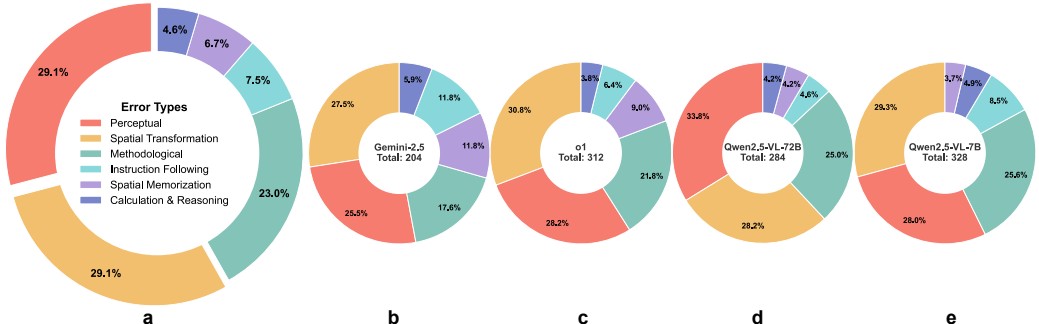

Figure 5: **Comparison of error type distributions**, with chart (a) showing the overall breakdown and charts (b-e) detailing results for specific MLLMs: (b) Gemini-2.5, (c) o1, (d) Qwen2.5-VL-72B and (e) Qwen2.5-VL-7B. Errors are classified into six categories: Perceptual, Spatial Transformation, Methodological, Instruction Following, Spatial Memorization, and Calculation & Reasoning.

performance trends remained consistent, with Qwen2.5-VL-72B still underperforming compared to its non-CoT baseline (35.00%). Second, we compared two distinct answer extraction rules (truncated letter matching as Rule A vs. full-format regex matching as Rule B, detailed in Appendix E.4). Table 3(b) reveals that the discrepancy between rules is negligible ($< 1.2\%$), confirming that the negative impact of CoT (ranging from -5% to -9%) is a genuine reasoning failure, not a parsing error.

## 4.3 ERROR ANALYSIS

This section first presents a statistical error analysis across several representative models to identify common failure modes, followed by a detailed case study of Gemini-2.5-pro to illustrate its specific reasoning processes.

### 4.3.1 STATISTICAL ERROR ANALYSIS

This evaluation was conducted primarily through manual review (2 human annotators), utilizing Gemini-2.5-pro as an assistive tool based on 6 manually defined error categories, including perceptual, spatial transformation, spatial memorization, instruction following, methodological, and calculation & reasoning error (detailed in Appendix E.6.2). To account for diversity in developers, model sizes, and open/closed-source paradigms, we selected 4 models for deeper analysis: Gemini-2.5-pro and o1 (the top-performing closed-source models), Qwen2.5-VL-72B (a leading open-source model), and its smaller counterpart, Qwen2.5-VL-7B. To ensure the reliability of our error taxonomy, two annotators independently annotated a subset of 100 errors. We calculated the Cohen's Kappa coefficient ($\kappa = 0.85$), indicating strong inter-annotator agreement. Disagreements were resolved through discussion with a third expert.

**Perceptual and Spatial Transformation Errors Dominate Failures** The dominance of Perceptual and Spatial Transformation errors, which collectively account for nearly 60% of all failures, quantitatively supports our central hypothesis that the primary MLLM bottleneck stems from fundamental failures in visual perception and transformation, not from high-level reasoning deficits. In contrast, the low frequency of Calculation & Reasoning and Instruction Following errors confirms the benchmark's effective isolation of spatial deficits. Methodological errors, the third-largest category at over 23%, indicate that models often adopt suboptimal problem-solving strategies. This highlights a clear direction for future improvements: enhancing spatial visualization capabilities by augmenting the training data with more correct solutions.

**Model Scaling Fails to Resolve Core Spatial Deficits** A model's absolute error count correlates with its performance rank: Gemini-2.5-pro had the fewest errors (204), followed by o1 (236), Qwen2.5-VL-72B (272), and Qwen2.5-VL-7B (328). Although top models show similar error profiles, Gemini-2.5-pro's lower rate of Methodological errors partly explains its superior performance. The limits of model scaling become evident when comparing Qwen2.5-VL-7B and Qwen2.5-VL-72B. Despite a tenfold parameter increase, their core error patterns remained strikingly similar, with Perceptual and Transformation errors still dominant. While the 72B model nearly eliminated Spatial Memorization and Calculation errors, it made only limited gains on these most critical error types. This reveals a

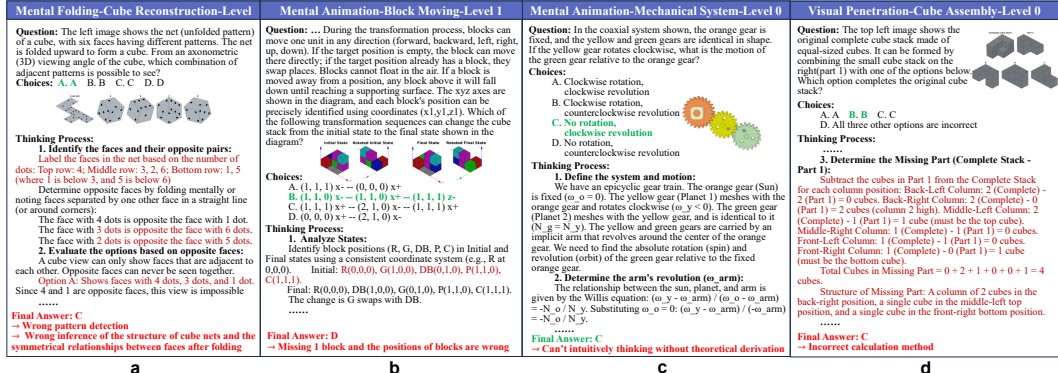

Figure 6: Case study of Gemini-2.5-pro's reasoning in different tasks.

crucial insight: scaling alone does not resolve fundamental spatial reasoning deficits. True progress will likely require innovations in training paradigms, such as (DeepSeek-AI et al., 2025), rather than merely increasing model size.

### 4.3.2 ANALYSIS OF TEST CASES

To complement the statistical analysis, we conducted a qualitative case study of Gemini-2.5-pro's reasoning processes. The model exhibited strong reasoning, following logically coherent and complete processes, validating the effectiveness of our evaluation results. This analysis reveals a significant gap between its abstract reasoning capabilities and its visuospatial processing abilities, reinforcing that the primary bottleneck is not high-level logic but fundamental perception and visualization.

**Deficiencies Found in Both Perception and Visualization** A qualitative case study of Gemini-2.5-pro's reasoning reveals errors occur at two distinct stages: perceiving visible information and reasoning about unseen spatial relationships. In processing visible information, the model exhibited deficiencies in 2D tasks like color recognition and complex pattern identification (Figure 6.a). These perceptual failures were more pronounced in 3D space, where it struggled to accurately identify the quantity, position, and spatial relationships of stacked cubes (Figure 6.b). This difficulty is quantified by a stark performance drop, with accuracy plummeting from 95% on the 2D Arrow Moving task to just 35% on analogous 3D tasks. The model's primary struggles, however, emerged when reasoning about unseen information. It consistently failed tasks requiring mental manipulation, such as accurately inferring the structure of cube nets or the symmetrical relationships between faces after folding.

**Pre-training Biases Drive Non-Simulative Problem Solving** The case study also uncovered strong pre-training biases that shape the model's problem-solving approach. For Mechanical System tasks, which were designed to be solvable via pure spatial visualization, Gemini-2.5-pro often defaulted to applying theoretical physics formulas instead of mentally simulating the motion (Figure 6.c). This behavior diverges sharply from human strategies and reveals a critical misalignment between the model's problem-solving approach and genuine spatial intelligence, suggesting its internal world model is more analytical than simulative. These qualitative examples directly illustrate the types of Methodological failures identified in our statistical analysis, forming a cohesive picture of current MLLM limitations.

## 5 CONCLUSION

We introduce *SpatialViz-Bench*, a cognitive-science–inspired for testing spatial visualization in MLLMs, designed for continuous task expansion while ensuring fair evaluation by preventing data contamination via a dynamic test bank. It comprises *12* tasks (*1,180* problems) across *4* core sub-abilities: mental rotation, mental folding, visual penetration, and mental animation. Its results show strong discriminative power, revealing the primary limitation in models is visuospatial acquisition over logical reasoning, guiding targeted optimizations in spatial skills.

# 6 ACKNOWLEDGEMENT

This work was supported by the National Natural Science Foundation of China (NO.62572471).

# 7 ETHICS STATEMENT

**Data Licensing** The SpatialViz-Bench benchmark is released under the MIT license to promote academic and non-commercial research. Its licensing fully complies with all third-party assets used in its creation, which include materials governed by the LGPL (e.g., FreeCAD), MIT (e.g., DeepCAD), CC0 1.0 Universal Public Domain Dedication (e.g., assets from public websites), and default licenses from websites that are known to support non-commercial fair use (e.g., assets from various video websites). For SpatialViz-Bench, we abide by Fair Use §107: "the fair use of a copyrighted work, including such use by · · · scholarship, or research, is not an infringement of copyright", where fair use is determined by "the purpose and character of the use, including whether such use is of a commercial nature or is for nonprofit educational purposes" and "the effect of the use upon the potential market for or value of the copyrighted work."

**Labor Practices** All manual data processing and annotation adhered to fair labor practices. Data review for the Mechanical System task was performed by non-author members of our research group, who were compensated for their work via research stipends. All other manual processes, including initial task creation, data verification, and model error analysis, were conducted by the author team as part of their standard research responsibilities.

# 8 REPRODUCIBILITY STATEMENT

To ensure the full reproducibility of our research, we have made all necessary materials available. The supplementary materials include the complete source code used for data generation (11 of 12 tasks) and model evaluation. Due to submission size constraints, we have provided a "mini" version of our benchmark data, which is sufficient to verify our experimental setup and replicate the core results. Furthermore, to facilitate a clear understanding of our methodology, Appendix B.4 provides detailed pseudocode for each key algorithm.

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

APPENDIX

## A   DETAILED RELATED WORKS

### A.1   CURRENT LANDSCAPE IN SPATIAL REASONING BENCHMARKS

Spatial reasoning is foundational to embodied intelligence, supporting critical tasks like navigation, interaction, and scene understanding. The evaluation of this ability in MLLMs has historically focused on two primary areas: spatial perception and spatial memorization, both of which rely on interpreting directly observable, explicit visual information.

**Spatial Perception**, the ability to interpret spatial relationships from static visual input, is the most established area. Early benchmarks targeted perceptual-level understanding, such as monocular depth estimation and object localization. With the rise of MLLMs, this has shifted to visual question answering formats. For instance, datasets like VSR (Liu et al., 2023) and What'sUp (Kamath et al., 2023) benchmark models' comprehension of object-centric spatial relationships. Others, including SpatialVLM (Chen et al., 2024), Spatial-MM (Shiri et al., 2024), and MMRel (Nie et al., 2024), further expand this evaluation to include relative distances, camera-object perspectives, and object size comparisons. More advanced benchmarks like Blink (Fu et al., 2024), with its Multi-view Reasoning task, and SpatialRGPT-bench (Cheng et al., 2024), which incorporates world knowledge and multi-hop reasoning, have pushed the boundaries but remain centered on interpreting what is explicitly perceived.

**Spatial Memorization**, the ability to track objects and their relationships in dynamic scenes, has been increasingly addressed by video-based benchmarks. VCBench (Li et al., 2024) evaluates this through tasks like Flash Grid and 3D Navigator, which test a model's capacity to retain 2D spatial positions and predict trajectories in 3D space. Similarly, VSI-bench (Yang et al., 2024b) focuses on skills essential for navigation, such as egocentric-to-allocentric transformation and perspective-shifting.

While these efforts have built a strong foundation, they predominantly assess reasoning based on explicit visual cues. They largely neglect the more advanced capability of spatial visualization—the mental manipulation of shapes and inference of implicit spatial information—leaving a significant gap in the current evaluation landscape.

### A.2   THE INADEQUATE EVALUATION OF SPATIAL VISUALIZATION

Despite its importance, the evaluation of spatial visualization is fraught with challenges, including obscured categorization in general benchmarks, high risk of data contamination, and a lack of diagnostic depth.

**Obscured Categorization** Spatial visualization is often not recognized as a distinct spatial skill. Instead, it is frequently subsumed under broader domains like mathematical or logical reasoning within general-purpose MLLM benchmarks. Examples are widespread: it appears as the 3D-Geometry category in MM-IQ (Cai et al., 2025) and MARVEL (Jiang et al., 2024), the 3D Spatial Simulation category in EMMA (Hao et al., 2025), 3D Shapes in LogicVista (Xiao et al., 2024), IQ-Test in Blink (Fu et al., 2024), and Descriptive/Transformation Geometry in Math-Vision (Wang et al., 2024). While VisualPuzzles (Song et al., 2025) correctly situates it under spatial reasoning, this is an exception. This common miscategorization diverts focus from developing and evaluating spatial visualization as a core ability, treating it merely as a type of puzzle.

**Risk of Data Contamination** The difficulty of designing novel spatial visualization tasks means that existing benchmarks often source questions from public materials like IQ tests, administrative exams, and math contests. This practice creates a high risk of data contamination, as these materials are likely part of the massive web-scraped datasets used for pretraining MLLMs. For example, work by Xu et al. (2025b) collects data entirely from online psychological tests. Consequently, a model's high performance on such benchmarks may not reflect true reasoning capabilities but rather memorization from the training data, compromising evaluation validity.

**Non-Diagnostic Evaluation** Current evaluations are often caught between two non-diagnostic extremes. On one hand, the heterogeneous, mixed-format questions in general benchmarks make it difficult to isolate and diagnose errors in spatial visualization specifically. On the other hand, specialized datasets are often too narrowly focused on a single sub-skill. For example, SPARE3D (Han et al., 2020) and CLEVR-MRT (Beckham et al., 2023) concentrate on mental rotation, while SRBench (Sto-

giannidis et al., 2025) uses only paper folding tasks to assess the entire ability. This narrow scope fails to provide a comprehensive assessment of a model's overall spatial visualization proficiency.

In contrast to these prior works, our benchmark is designed to be systematic and diagnostic. It is structured around *4* core sub-skills of spatial visualization identified in cognitive psychology, with curated tasks targeting each ability. By employing procedural generation for most tasks, our benchmark ensures greater reliability, reduces the risk of training-set overlap, and enables scalable data creation for both evaluation and future training. Furthermore, by summarizing the essential phases of spatial visualization, our framework allows for a more granular analysis to identify the root causes of reasoning errors.

# B    Data Curation Details

## B.1    Task Construction

### 1. Mental Rotation

**2D Rotation Task.** A colored grid pattern with a red corner marker is rotated by 90°/180°/270° to generate positive samples. Negative samples involve horizontal/vertical mirroring. We further replace symmetric color fills with non-centrally symmetric patterns. Negatives include mirror flips and internal rotations of pattern components, increasing spatial reasoning difficulty. As shown in Algorithm 1.

**3D Rotation Task.** A connected cube stack is rotated along x/y/z axis to form positives. Negatives are created by removing one cube or mirroring the isometric view, ensuring no simple rotation can reproduce them. Spatial complexity is increased by enlarging assembly dimensions, requiring enhanced 3D rotational reasoning. As shown in Algorithm 2 and Algorithm 3.

**Three-View Projection Task.** This task has two categories. Firstly, given isometric, front, and top views of a connected cube stack with marked reference cubes, the task is to select the correct left view. Negatives involve altering reference cube positions or substituting the right view. We further introduce real engineering parts from the DeepCAD dataset (Wu et al., 2021), rendered into standard projections via FreeCAD. Negatives are crafted through random internal lines deletion, view flipping/rotation, or transformations on unseen views. As shown in Algorithm 4 and Algorithm 5.

### 2. Mental Folding

**Paper Folding Task.** A Python-based pipeline generates $m \times n$ grid patterns undergoing sequential folds (vertical/horizontal/diagonal), followed by hole-punching and unfolding. The task requires identifying the correct unfolded hole distribution. Negative samples are generated by mirroring, deleting, adding, or relocating holes to violate fold-induced symmetry. Task difficulty increases with more folds, larger grids, and denser hole placements. As shown in Algorithm 6 and Algorithm 7.

**Cube Unfolding Task.** Given a cube with six uniquely colored faces and a view from a corner (three visible faces), the task is to select the correct 2D net (11 possibilities as shown in Figure 7). Positives can be crafted either by using different cube nets of the same cube or by fixing the mapping of visible faces while randomly shuffling the remaining faces. Negatives are crafted by swapping visible face colors or flipping visible-opposite face pairs. We further replace solid colors with non-centrally symmetric patterns. View angles prioritize faces with asymmetric patterns. Internal rotations of pattern components are introduced to further increase the reasoning difficulty. To push the difficulty even further, all six faces feature random colored-dot patterns on a 3×3 grid. As shown in Algorithm 8, Algorithm 9 and Algorithm 10.

**Cube Reconstruction Task.** Cubes have six uniquely colored faces. Two task variants exist: (1) select the correct vertex view of a cube when given its net pattern, with negative samples created by mirroring the correct view; (2) identify the color of a face opposite to a given colored face. Difficulty progression follows the cube unfolding tasks. As shown in Algorithm 8 and Algorithm 11.

### 3. Visual Penetration

**Cross-Section Task.** Nine basic geometric solids (e.g., triangular/rectangular/circular prisms/pyramids/frustums) are combined in pairs with conical shapes on top. Cross-sections are generated by slicing the composite shapes using planes parallel to the XY/YZ/XZ planes. Negative samples are

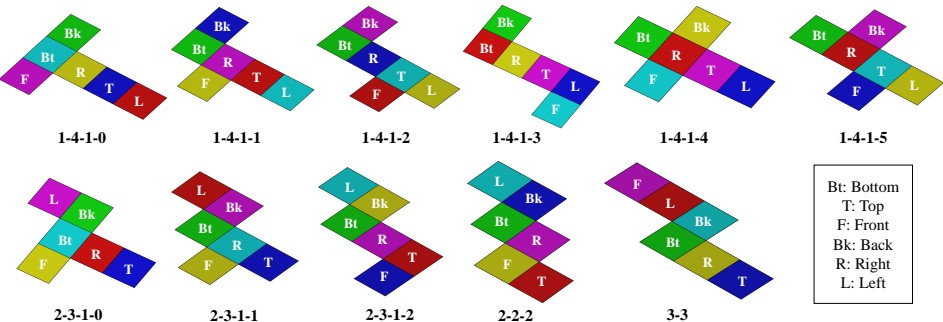

Figure 7: The eleven unfolded patterns of a cube with their corresponding numbered names. Assuming the square in row 1, position 0 represents the bottom face, and position 1 represents the right face, the corresponding arrangement of the remaining faces can be determined, facilitating the rotation of the cube.

constructed by adjusting the relative geometric proportions within the composite. Task complexity is increased by introducing composites with three solids, which often produce disconnected cross-sections that demand enhanced visual reasoning. Additional complexity is introduced by generating oblique cross-sections at $45°/135°$. As shown in Algorithm 12.

**Cube Counting Task.** The task requires inferring the total cube count of a connected cube stack based on two orthogonal projection views. The minimum and maximum counts are mathematically derived to guide the construction of answer options. Constraints increase to three orthogonal projection views, reducing the number of possible solutions while increasing view integration complexity. Task difficulty further increases by expanding the spatial dimensions of the cubic assemblies. As shown in Algorithm 2 and Algorithm 13.

**Cube Assembly Task.** A pyramid-like cube stack is split into two connected parts. Tasks require identifying the complementary piece that fits the reference part. Negative samples are generated by modifying the correct piece through the addition or removal of cubic units. The difficulty is further increased by enlarging the spatial dimensions and dividing the structure into three parts instead of two. As shown in Algorithm 14 and Algorithm 15.

## 4. Mental Animation

**Arrow Moving Task.** For the easy version, an arrow with random initial position and orientation in a 3×3 grid operates by ego-centric rules: movement occurs in 4 directions (forward/backward/left/right), with "forward" always indicating the arrow's current orientation. The arrow reorients to the movement direction after each movement. Valid operation sequences are algorithmically generated; negative samples share the same initial state but yield incorrect endpoints. For the hard version, multiple colored arrows are introduced with extended rules: empty positions allow direct entry; occupied positions trigger object exchanges while maintaining Level 0 movement principles. Tasks include predicting final states from sequences, or inferring correct sequences from state pairs. As shown in Algorithm 16, Algorithm 17, Algorithm 18 and Algorithm 19.

**Block Moving Task.** Colored cube stack combines directional movement with gravity simulation. Cubes move along six directions with unsupported cubes falling until reaching support and swapping positions as same as Arrow Moving Task. Increased spatial complexity and longer sequences elevate reasoning difficulty. As shown in Algorithm 20 and Algorithm 21.

**Mechanical System Task.** We use open-source mechanical system simulations, classifying complexity by module quantity and designing appropriate questions. These tasks assess advanced mental animation abilities, particularly to understand how the motion of one component affects others.

### B.2 PROGRAMMATIC DATA GENERATION PIPELINE

FreeCAD, an open-source Computer-Aided Design (CAD) software, provides deep integration with Python programming language, enabling parametric model construction through programming. We leveraged the synergy between FreeCAD and Python to successfully automate the generation of

9 spatial visualization tasks: 2DRotation, 3DRotation, 3ViewProjection, CubeFolding, CubeReconstruction, CrossSection, CubeCounting, CubeAssembly, and BlockMoving. Additionally, two tasks—PaperFolding and ArrowMoving—were implemented solely using Python. For the MechanicalSystem task, due to its complexity and specific requirements, we employed precise manual design methods. To supplement the task overview presented in Section 3.3, the following sections provide detailed pseudocode for each programmatically generated task, offering more systematic and in-depth technical insights.

**Mental Rotation Tasks.** Algorithm 1 presents the pseudocode for the 2D Rotation Task. For the 3D Rotation Task, Three-View Projection Task, Cube Counting Task, and Block Moving Task, we need to construct connected cube stacks, with the core functions detailed in Algorithm 2. Algorithm 3 demonstrates the complete implementation process of the 3D Rotation Task. The method for generating three-view projections of marked cube stacks is elaborated in Algorithm 4. Algorithm 5 describes the process of importing models from the DeepCAD dataset and generating their three-view projections.

**Mental Folding Tasks.** Algorithm 6 implements a Paper class for simulating the dynamic processes of paper folding, holes punching, and unfolding. Based on this simulation framework, Algorithm 7 constructs the data for the Paper Folding Task. Algorithm 8 presents the core functions for transforming 11 standard cube nets (as shown in Figure 7) into three-dimensional cubes. Utilizing these transformation functions, while Algorithm 9 demonstrates how different unfolding patterns can produce the same cube.Algorithm 10 and Algorithm 11 provide the complete pseudocode implementations for the Cube Unfolding Task and Cube Reconstruction Task, respectively.

**Visual Penetration Tasks.** Algorithm 12 details the implementation pseudocode for the Cross-Section Task. Algorithm 13 comprehensively presents the data generation procedure as well as the mathematical calculation process to guide the construction of answer options in the Cube Counting Task. Algorithm 14 contains the core functions for decomposing a complete cube stack into multiple connected parts. Building upon these functions, Algorithm 15 provides the complete construction pseudocode for the Cube Assembly Task.

**Mental Animation Tasks.** Algorithm 16 implements an ArrowPath class for simulating the movement process of an arrow centered on itself. Algorithm 17 implements an ArrowMap class that inherits from the ArrowPath class, designed to simulate movement and exchange operations in multi-arrow environments. Based on the ArrowPath class, Algorithm 18 details the data construction process for the single-arrow version of the Arrow Moving Task. Correspondingly, using the ArrowMap class, Algorithm 19 elucidates the data construction process for the multi-arrow version of the Arrow Moving Task. Algorithm 20 implements a Block class for simulating the movement and exchange processes of blocks that follow gravitational rules. Building upon this Block class, Algorithm 21 presents the complete pseudocode implementation of the Block Moving Task.

### B.3 MANUL DESIGN FOR MECHANICAL SYSTEM TASK

To ensure the objectivity and quality of the Mechanical System task, we first collected simulation materials from open-source platforms. The question-answer pairs were designed by members of the author team, who strictly followed a standardized template based on the observable and deterministic animations (e.g., "If component A rotates clockwise, how does component B move?"). This structured process was designed to minimize subjectivity and focus the evaluation specifically on a model's ability to infer causal dynamics from visual input. To verify the accuracy of these question-answer pairs, we recruited two graduate student annotators from our research group, who received compensation for their contributions. They first performed independent reviews of each sample and then discussed their findings to resolve any discrepancies and reach a final consensus. This rigorous process ultimately produced 80 validated data samples.

## B.4 PSEUDOCODE

---

**Algorithm 1** 2D Rotation Task

---

1: **Input:** Color(Pattern) set $C$, grid size $(H, W)$, unit length $s$, marker length $s'$, task mode $m$
2: Initialize binary matrix $M \in \{0,1\}^{H \times W}$ with random values
3: Initialize empty lists $positive\_samples$, $negative\_samples$
4: **function** DRAWGRIDWITHMARKER($M$, $C$, $H$, $W$, $s$, $s'$, $record = list()$)
5:      **for** $i \leftarrow 0$ to $H-1$ **do**
6:          **for** $j \leftarrow 0$ to $W-1$ **do**
7:              $pos \leftarrow (j \cdot s,\ (H-1-i) \cdot s,\ 0)$
8:              $square \leftarrow$ FreeCAD.makePlane($s$, $s$, $(pos, 0°)$)
9:              **if** $M[i][j] = 1$ **then**
10:                  **if** $record$ is empty **then**:
11:                      Randomly select $c \in C$ and assign $c$ to $square$ at $pos$
12:                      Append $c$ to $record$
13:                  **else**
14:                      Assign rotate(Pop($record, 0$), $90°$) to $square$ at $pos$
15:                  **end if**
16:              **end if**
17:          **end for**
18:      **end for**
19:      Randomly select $corner \in \{$"top_left", "top_right", "bottom_left", "bottom_right"$\}$
20:      $pos_{\text{marker}} \leftarrow$ get_marker_pos($H$, $W$, $s$, $s'$, $corner$)
21:      FreeCAD.makePlane($s'$, $s'$, $(pos_{\text{marker}}, 0°)$) with red color
22:      $img \leftarrow$ FreeCAD.saveImage()
23:      **return** $img$, $record$
24: **end function**

25: $ref\_img, record \leftarrow$ DrawGridWithMarker($M, C, H, W, s, s'$)
26: **if** $m =$ "pattern" **then**
27:      $transform\_image, record \leftarrow$ DrawGridWithMarker($M, C, H, W, s, s', record$)
28:      Append $transform\_img$ to $negative\_samples$
29: **end if**
30: **for** $angle \in \{90°, 180°, 270°\}$ **do**
31:      $img \leftarrow$ rotate($ref\_img, angle$)
32:      Append $img$ to $positive\_samples$
33: **end for**
34: **for** $flip\_dir \in \{$"horizontal", "vertical"$\}$ **do**
35:      $img \leftarrow$ flip($ref\_img, flip\_dir$)
36:      Append $img$ to $negative\_samples$
37: **end for**
38: $samples \leftarrow (positive\_samples, negative\_samples)$
39: Shuffle $samples$ to assign $[A, B, C, D]$ and record $answer\_id$
40: $data \leftarrow$ create_data($ref\_img, samples, question, answer\_id$)

---

---

**Algorithm 2** Fucntions for Creating Cubes with None-isolated Regions

---

1: **Input:** Spatial size $(X, Y, Z)$, cube size $s$
2: Initialize zero value 3D tensors $placement \in \{0\}^{Z \times Y \times X}$, empty list $cubes$
3: **function** CREATECUBE($x$, $y$, $z$)
4: $\quad$ $cube \leftarrow$ FreeCAD.makebox$(s, s, s, (x, y, z))$ and append $cube$ to $cubes$
5: $\quad$ $placement[z][y][x] \leftarrow 1$
6: **end function**

7: **function** CREATECUBES($X$, $Y$, $Z$)
8: $\quad$ **for** $z \leftarrow 0$ to $Z-1$ **do**
9: $\quad\quad$ **for** $y \leftarrow 0$ to $Y-1$ **do**
10: $\quad\quad\quad$ **for** $x \leftarrow 0$ to $X-1$ **do**
11: $\quad\quad\quad\quad$ **if** $z = 0$ or $placement\_space[z-1][y][x] = 1$ **then**
12: $\quad\quad\quad\quad\quad$ With 50% probability CreateCube$(x, y, z)$
13: $\quad\quad\quad\quad$ **end if**
14: $\quad\quad\quad$ **end for**
15: $\quad\quad$ **end for**
16: $\quad$ **end for**
17: **end function**

18: **function** CONNECTISOLATEDCUBES($X$, $Y$)
19: $\quad$ $cubes_{xy} \leftarrow \{(x, y) \mid placement[0][y][x] = 1\}$
20: $\quad$ Initialize empty set $visited$, empty list $regions$
21: $\quad$ $directions \leftarrow$ [(-1,0),(1,0),(0,-1),(0,1),(-1,-1),(-1,1),(1,-1),(1,1)]
22: $\quad$ **for all** $(x, y) \in cubes_{xy}$ **do**
23: $\quad\quad$ **if** $(x, y) \notin visited$ **then**
24: $\quad\quad\quad$ Initialize empty list $region$, empty queue $queue$
25: $\quad\quad\quad$ Add $(x, y)$ to $visited$, add $(x, y)$ to $queue$
26: $\quad\quad\quad$ **while** $queue$ is not empty **do**
27: $\quad\quad\quad\quad$ $(cx, cy) \leftarrow$ popLeft$(queue)$
28: $\quad\quad\quad\quad$ Append $(cx, cy)$ to $region$
29: $\quad\quad\quad\quad$ **for all** $(dx, dy) \in directions$ **do**
30: $\quad\quad\quad\quad\quad$ $(nx, ny) \leftarrow (cx + dx, cy + dy)$
31: $\quad\quad\quad\quad\quad$ **if** $0 \leq nx < X$ and $0 \leq ny < Y$ and $(nx, ny) \notin visited$
$\quad\quad\quad\quad\quad\quad$ and $placement[0][ny][nx] = 1$ **then**
32: $\quad\quad\quad\quad\quad\quad$ Add $(nx, ny)$ to $visited$, add $(nx, ny)$ to $queue$
33: $\quad\quad\quad\quad\quad$ **end if**
34: $\quad\quad\quad\quad$ **end for**
35: $\quad\quad\quad$ **end while**
36: $\quad\quad\quad$ Append $region$ to $regions$
37: $\quad\quad$ **end if**
38: $\quad$ **end for**
39: $\quad$ **if** $|regions| > 1$ **then**
40: $\quad\quad$ **for** $i \leftarrow 0$ to $|regions| - 2$ **do**
41: $\quad\quad\quad$ Find $(x_1, y_1), (x_2, y_2)$ with min $L_1$ distance between $regions[i]$ and $regions[i+1]$
42: $\quad\quad\quad$ $x \leftarrow x_1, y \leftarrow y_1$
43: $\quad\quad\quad$ **while** $(x \neq x_2)$ or $(y \neq y_2)$ **do**
44: $\quad\quad\quad\quad$ **if** $x \neq x_2$ and $y \neq y_2$ **then**
45: $\quad\quad\quad\quad\quad$ $x \leftarrow x \pm 1, y \leftarrow y \pm 1$
46: $\quad\quad\quad\quad$ **else if** $x \neq x_2$ **then**
47: $\quad\quad\quad\quad\quad$ $x \leftarrow x \pm 1$
48: $\quad\quad\quad\quad$ **else if** $y \neq y_2$ **then**
49: $\quad\quad\quad\quad\quad$ $y \leftarrow y \pm 1$
50: $\quad\quad\quad\quad$ **end if**
51: $\quad\quad\quad\quad$ **if** $placement\_space[0][y][x] = 0$ **then**
52: $\quad\quad\quad\quad\quad$ CreateCube$(placement, x, y, 0)$
53: $\quad\quad\quad\quad$ **end if**
54: $\quad\quad\quad$ **end while**
55: $\quad\quad$ **end for**
56: $\quad$ **end if**
57: **end function**

---

---

**Algorithm 3** 3D Rotation Task

---

1: **Input:** Spatial size $(X, Y, Z)$, cube size $s$
2: Initialize zero value 3D tensors $placement \in \{0\}^{Z \times Y \times X}$, empty list $cubes$
3: Initialize empty lists $positive\_samples$, $negative\_samples$
4: Update $placement$, $cubes$ with $\texttt{CreateCubes}(X, Y, Z)$
5: Update $placement$, $cubes$ with $\texttt{ConnectIsolatedCubes}(X, Y)$
6: $ref\_img \leftarrow \texttt{FreeCAD.saveImage}(cubes)$

7: **for** $i \leftarrow 1$ to $4$ **do**
8:     Randomly select $axis \in \{x, y, z\}$ and $angle \in \{90°, 180°, 270°\}$
9:     $rotated\_cubes \leftarrow \texttt{rotate}(cubes,\ axis,\ angle)$
10:     $rotated\_img \leftarrow \texttt{FreeCAD.saveImage}(rotated\_cubes)$
11:     Append $rotated\_img$ to $positive\_samples$
12: **end for**

13: $cubes' \leftarrow$ Randomly remove a cube from $cubes$ and rotate the left cubes as above
14: $rotated\_removed\_img \leftarrow \texttt{FreeCAD.saveImage}(cubes')$
15: Append $rotated\_removed\_img$ to $negative\_samples$

16: **for** $flip\_dir \in \{\text{"horizontal", "vertical"}\}$ **do**
17:     Randomly choose $sample$ from $positive\_samples$
18:     $img \leftarrow \texttt{flip}(sample, flip\_dir)$
19:     Append $img$ to $negative\_samples$
20: **end for**

21: $samples \leftarrow (positive\_samples, negative\_samples)$
22: Shuffle $samples$ to assign $[A, B, C, D]$ and record $answer\_id$
23: $data \leftarrow \texttt{create\_data}(ref\_img, samples, question, answer\_id)$

---

---

**Algorithm 4** Three-View Projection Task with Marked Cube Stack

---

1: **Input:** Spatial size $(X, Y, Z)$, cube size $s$
2: Initialize zero value 3D tensors $placement \in \{0\}^{Z \times Y \times X}$, empty list $cubes$
3: Initialize empty lists $positive\_samples$, $negative\_samples$
4: Update $placement, cubes$ with $\texttt{CreateCubes}(X, Y, Z)$
5: Update $placement, cubes$ with $\texttt{ConnectIsolatedCubes}(X, Y)$

6: **function** COLORVISIBLEFACES($X$, $Y$, $Z$, $colored\_num$)
7:     $cubes \leftarrow$ Find cubes that can be seen from front or top or left view
8:     Randomly color $\min(colored\_num, |cubes|)$ cubes in red
9: **end function**

10: **function** SAVEVIEWS($cubes$)
11:     Initialize empty list $views$
12:     **for all** $view \in \{$"Isometric", "Top", "Front", "Left"$\}$ **do**
13:         $img \leftarrow \texttt{FreeCAD.saveView}(view)$ and append $img$ to $views$
14:     **end for**
15:     **return** $views$
16: **end function**

17: Update $cubes$ with $\texttt{ColorVisibleFaces}(X, Y, Z, colored\_num)$
18: $views \leftarrow \texttt{SaveViews}(cubes)$
19: Select $left\_view$ from $views$ to $positive\_samples$
20: Select $right\_view$ from $views$ to $negative\_samples$

21: Cleaer all colors and update $cubus$ with $\texttt{ColorVisibleFaces}(X, Y, Z, colored\_num)$ as above
22: $new\_views \leftarrow \texttt{SaveViews}(cubes)$
23: Select $left\_view$ and $right\_view$ from $new\_views$ to $negative\_samples$

24: $samples \leftarrow (positive\_samples, negative\_samples)$
25: Shuffle $samples$ to assign $[A, B, C, D]$ and record $answer\_id$
26: $ref\_img \leftarrow (isometric\_view, top\_view, front\_view)$
27: $data \leftarrow \texttt{create\_data}(ref\_img, samples, question, answer\_id)$

---

---

**Algorithm 5** Three-View Projection Task with Models from DeepCAD Datasets

---

1: **Input:** step file path $pth$
2: Initialize empty lists $positive\_samples$, $negative\_samples$
3: $shape \leftarrow$ Open($pth$)
4: $views \leftarrow$ SaveViews($shape$)
5: **function** CREATEINCORRECTVIEW($view$, $mode$)
6:     **if** $mode = 0$ **then**
7:         $img' \leftarrow$ Extract all internal lines and randomly delete 1 line
8:     **else if** $mode = 1$ **then**
9:         $img' \leftarrow$ rotate($view, 90°$)
10:     **else if** $mode = 2$ **then**
11:         $img' \leftarrow$ flip($view$, "horizontal" or "vertical")
12:     **end if**
13:     **return** img'
14: **end function**

15: $ref\_view \leftarrow$ Choose view from $views$ with max area
16: $(questioned\_view, other\_view) \leftarrow$ Randomly assign $views$ except for $ref\_view$
17: Append $questioned\_view$ to $positive\_samples$
18: **for** $mode \leftarrow 0$ to $2$ **do**
19:     $incorrect\_view \leftarrow$ CreateIncorrectView($questioned\_view$ or $other\_view$, $mode$)
20:     Append $incorrect\_view$ to $negative\_samples$
21: **end for**

22: $samples \leftarrow (positive\_samples, negative\_samples)$
23: Shuffle $samples$ to assign $[A, B, C, D]$ and record $answer\_id$
24: $ref\_img \leftarrow (isometric\_view, top\_view, front\_view)$
25: $data \leftarrow$ create_data($ref\_img, samples, question, answer\_id$)

---

---

**Algorithm 6** Simulation for Paper Folding, Punching and Unfolding

---

1: **Class** Paper
2: **Attributes:**
3:    $grid$, $complete\_grid$: 2D arrays representing current and complete paper states
4:    $original\_rows$, $original\_cols$: initial dimensions
5:    $current\_rows$, $current\_cols$: current dimensions after folding
6:    $folds$: list of fold operations
7: **function** FOLD($direction$, $line$ or $diagonal\_points$)
8:    **if** $direction$ is horizontal **then**
9:       Calculate folded area
10:       Update $complete\_grid$ by marking folded area as -1
11:       Create new grid with updated dimensions
12:    **else if** $direction$ is vertical **then**
13:       Similar to horizontal but for columns
14:    **else if** $direction$ is diagonal **then**
15:       Calculate diagonal line equation
16:       Mark appropriate triangular area as -1
17:    **end if**
18:    Record fold operation in $folds$
19: **end function**

20: **function** PUNCH($points$)
21:    **for** each $(x, y)$ in $points$ **do**
22:       Set $grid[x][y] \leftarrow 1$
23:       Set corresponding $complete\_grid$ position to 1
24:    **end for**
25:    Record punch operation in $folds$
26: **end function**

27: **function** UNFOLD
28:    **for** each $fold$ in reverse $folds$ **do**
29:       **if** $fold$ is horizontal **then**
30:          Mirror grid about fold line
31:       **else if** $fold$ is vertical **then**
32:          Mirror grid about fold line
33:       **else if** $fold$ is diagonal **then**
34:          Mirror grid about diagonal line
35:       **end if**
36:       Update current dimensions of paper
37:    **end for**
38:    Clear $folds$ list
39: **end function**

40: **function** CREATEINCORRECTVIEW($mode$)
41:    Create incorrect variant by:
42:    **if** $mode = $ "row" **then**
43:       Either remove a row of holes, add extra row, or swap rows
44:    **else if** $mode = $ "col" **then**
45:       Either remove a column of holes, add extra column, or swap columns
46:    **else**
47:       Combine row and column errors
48:    **end if**
49:    Update $paper$ with above changes
50: **end function**

---

---

**Algorithm 7** Paper Folding Task

---

1: **Input:** Dimensions of paper $(rows, cols)$, number of folds $steps$, number of holes $punches$
2: Initialize $paper$ with dimensions $rows \times cols$
3: Initialize empty lists $ref\_imgs, positive\_samples, negative\_samples$
4: **for** $step \leftarrow 1$ to $steps$ **do**
5:     **if** $step = steps$ **then**
6:         $direction \leftarrow$ "diagonal"
7:     **else**
8:         $direction \leftarrow$ Randomly select $direction \in$ ["horizontal","vertical"]
9:     **end if**
10:     **if** $direction =$ "horizontal" **then**
11:         $line \leftarrow$ randomInt$(1, paper.current\_rows - 1)$
12:         $paper$.Fold$(direction, line)$
13:     **else if** $direction =$ "vertical" **then**
14:         $line \leftarrow$ randomInt$(1, paper.current\_cols - 1)$
15:         $paper$.Fold$(direction, line)$
16:     **else if** $direction =$ "diagonal" **then**
17:         $diagonal\_points \leftarrow$ Randomly select one set of 45-degree line endpoints
18:         $paper$.Fold$(direction, diagonal\_points)$
19:     **end if**
20:     $img \leftarrow$ draw_paper$(paper)$ and append $img$ to $ref\_imgs$
21: **end for**

22: $points \leftarrow$ Randomly select $punches$ zero positions
23: $paper$.Punch$(points)$
24: $img \leftarrow$ draw_paper$(paper)$ and append $img$ to $ref\_imgs$

25: $paper$.Unfold$()$
26: $img \leftarrow$ draw_paper$(paper)$ and append $img$ to $positive\_samples$

27: Initialize $paper'$ with same dimensions as $paper$
28: $paper'.grid \leftarrow paper.grid$ to copy the state of unfolded paper
29: Determine the incorrect view $mode$
30: **for** $i \leftarrow 1$ to $3$ **do**
31:     Update $paper'$ with $paper'$.CreateIncorrectView$(mode)$
32:     $img \leftarrow$ draw_paper$(paper')$ and append $img$ to $negative\_samples$
33: **end for**

34: $samples \leftarrow (positive\_samples, negative\_samples)$
35: Shuffle $samples$ to assign $[A, B, C, D]$ and record $answer\_id$
36: $data \leftarrow$ create_data$(ref\_imgs, samples, question, answer\_id)$

---

---

**Algorithm 8** Functions for Reconstruting Cube from 11 Kinds of Cube Nets

---

1: **Input:** cube size $s$
2: Define rotation operators:
3:    $R_x(\theta)$: Rotation about X-axis by $\theta$ degrees
4:    $R_y(\theta)$: Rotation about Y-axis by $\theta$ degrees
5:    $R_z(\theta)$: Rotation about Z-axis by $\theta$ degrees
6: **function** NET2CUBE($plane\_name$, $map$, $view$, $rot$)
7:    Initialize placement dictionary $planes$
8:    $planes[$"Top"$] \leftarrow ((s/2, s/2, s),\ R_y(180°))$
9:    $planes[$"Bottom"$] \leftarrow ((s/2, s/2, 0),\ Rx(0))$
10:    $planes[$"Right"$] \leftarrow ((s, s/2, s/2),\ R_y(-90°))$
11:    $planes[$"Left"$] \leftarrow ((0, s/2, s/2),\ R_y(90°) \circ R_z(90°))$
12:    $planes[$"Back"$] \leftarrow ((s/2, s, s/2),\ R_x(90°))$
13:    **if** $plane\_name$ is "2-2-2" **then**
14:       $planes[$"Top"$] \leftarrow (s/2, s/2, s),\ R_x(180°) \circ R_z(-90°)$
15:    **else if** $plane\_name$ is "1-4-1" **then**
16:       $planes[$"Left"$] \leftarrow (0, s/2, s/2),\ R_y(90°)\circ$
17:    **end if**
18:    **if** $plane\_name \in [$"1-4-1-0", "2-3-1-0"$]$ **then**
19:       $planes[$"Front"$] \leftarrow ((s/2, 0, s/2),\ R_x(-90°))$
20:    **else if** $plane\_name \in [$"1-4-1-1", "1-4-1-4", "2-3-1-1", "2-2-2"$]$ **then**
21:       $planes[$"Front"$] \leftarrow ((s/2, 0, s/2),\ R_x(-90°) \circ R_z(-90°))$
22:    **else if** $plane\_name \in [$"1-4-1-2", "1-4-1-5", "2-3-1-2", "3-3"$]$ **then**
23:       $planes[$"Front"$] \leftarrow ((s/2, 0, s/2),\ R_x(-90°) \circ R_z(180°))$
24:    **else if** $plane\_name$ is "1-4-1-3" **then**
25:       $planes[$"Front"$] \leftarrow ((s/2, 0, s/2),\ R_x(-90°) \circ R_z(90°))$
26:    **end if**
27:    **if** $plane\_name \in [$"1-4-1-4", "1-4-1-5"$]$ **then**
28:       $planes[$"Back"$] \leftarrow ((s/2, s, s/2),\ R_x(90°) \circ R_z(90°))$
29:    **end if**

30:    **Form a cube by:**
31:    **for all** $face\_name \in planes$ **do**
32:       $placement \leftarrow planes[face\_name]$
33:       $square \leftarrow$ FreeCAD.makePlane($s$, $s$, $placement$)
34:       $c \leftarrow map[face\_name]$
35:       **if** $rot$ is true **then**
36:          Assign rotate($c$, $90°$) to $square$ at $placement$
37:       **else**
38:          Assign $c$ to $square$ at $placement$
39:       **end if**
40:    **end for**
41:    $img \leftarrow$ FreeCAD.saveView($view$)
42:    **return** $img$
43: **end function**

44: **function** DRAWNET($net$, $map$, $s$, $rot$)
45:    **for** $face\_name \in net$ **do**
46:       $i,\ j \leftarrow net[face\_name]$
47:       $pos \leftarrow (j \cdot s,\ (H - 1 - i) \cdot s,\ 0)$
48:       $square \leftarrow$ FreeCAD.makePlane($s$, $s$, $(pos, 0°)$)
49:       $c \leftarrow map[face\_name]$
50:       **if** $rot$ is true **then**
51:          Assign rotate($c$, $90°$) to $square$ at $pos$
52:       **else**
53:          Assign $c$ to $square$ at $pos$
54:       **end if**
55:    **end for**
56:    $img \leftarrow$ FreeCAD.saveImage()
57:    **return** $img$
58: **end function**

---

---

**Algorithm 9** Functions for Unfolding Cube to 11 kinds of Cube Nets

---

1: Using the same parameter definitions as those in Algorithm 8
2: **function** DRAWNETWIPIVOT($plane\_name, net, map, s, rot$)
3:     $pivot\_plane\_name \leftarrow$ "1-4-1-0"
4:     Initialize rotation dictionary $planes$
5:     **if** $plane\_name \in$ ["1-4-1-1", "1-4-1-4", "2-3-1-1", "2-2-2"] **then**
6:         $planes$["Front"] $\leftarrow R_z(90°))$
7:     **else if** $plane\_name \in$ ["1-4-1-2", "1-4-1-5", "2-3-1-2", "3-3"] **then**
8:         $planes$["Front"] $\leftarrow R_z(-180°))$
9:     **else if** $plane\_name$ is "1-4-1-3" **then**
10:        $planes$["Front"] $\leftarrow R_z(-90°))$
11:     **end if**
12:     **if** $plane\_name \in$ ["1-4-1-4", "1-4-1-5"] **then**
13:        $planes$["Back"] $\leftarrow R_z(-90°))$
14:     **end if**
15:     **if** $plane\_name \in$ ["2-3-1-0", "2-3-1-1", "2-3-1-2", "3-3", "2-2-2"] **then**
16:        $planes$["Left"] $\leftarrow R_z(-90°))$
17:     **end if**
18:     **if** $plane\_name$ is "2-2-2" **then**
19:        $planes$["Top"] $\leftarrow R_z(-90°))$
20:     **end if**
21:     **Create a net which can form the same cube with pivot plane:**
22:     **for** $face\_name \in net$ **do**
23:        $i, j \leftarrow net[face\_name]$
24:        $pos \leftarrow (j \cdot s, (H - 1 - i) \cdot s, 0)$
25:        $square \leftarrow$ `FreeCAD.makePlane`$(s, s, (pos, 0°))$
26:        **if** $rot$ is true **then**
27:           Assign `rotate`$(c, 90°)$ to square at $pos$
28:        **else**
29:           Assign $c$ to $square$ at $pos$
30:        **end if**
31:        **if** $plane\_name \neq$ "1-4-1-0" **then**
32:           **if** $face\_name \in planes$ **then**
33:              $rotation \leftarrow planes[face\_name]$
34:              $square.Placement.Rotation \leftarrow rotation$
35:           **end if**
36:        **end if**
37:     **end for**
38:     $img \leftarrow$ `FreeCAD.saveImage()`
39: **end function**

---

---

**Algorithm 10** Cube Unfolding Task

---

1: **Input:** Color(Pattern) set $C$, unit length $s$, task mode $m$
2: Initialize 11 cube nets
   $nets : \{face\_name : (i, j)|face\_name \in \{$"Top", "Bottom", "Right", "Left", "Back", "Front"$\}\}$
3: Initialize empty lists $positive\_samples, negative\_samples$
4: $map : \{face\_name : c|c \in C\} \leftarrow$ Randomly shuffle set $C$ and assign it to six faces
5: Randomly select a $view \in 8$ corner views of a cube
6: $pivot\_net\_name \leftarrow$ "1-4-1-0"
7: $ref\_img \leftarrow$ Net2Cube($pivot\_net\_name, map, view, rot =$ false)
8: **for** $i \leftarrow 1$ to 2 **do**
9:    $plane\_name, net \leftarrow$ Randomly select net from $nets$
10:    $img \leftarrow$ DrawNetWiPivot($plane\_name, net, map, s, rot =$ false)
11:    Append $img$ to $positive\_samples$
12:    **if** $m =$ "pattern" **then**
13:       $img' \leftarrow$ DrawNetWiPivot($plane\_name, net, map, s, rot =$ true)
14:       Append $img'$ to $negative\_samples$
15:    **end if**
16: **end for**

17: $map' \leftarrow$ Fix the mapping of $face\_name \in view$, and random shuffle the others
18: **for** $i \leftarrow 1$ to 2 **do**
19:    $plane\_name, net \leftarrow$ Randomly select net from $nets$
20:    $img \leftarrow$ DrawNetWiPivot($plane\_name, net, map, s, rot =$ false)
21:    Append $img$ to $positive\_samples$
22: **end for**

23: $map' \leftarrow$ Swap the colors(patterns) of a randomly selected $face \in view$ with its opposite face
24: $plane\_name, net \leftarrow$ Randomly select net from $nets$
25: $img \leftarrow$ DrawNetWiPivot($plane\_name, net, map', s, rot =$ false)
26: Append $img$ to $negative\_samples$

27: $samples \leftarrow (positive\_samples, negative\_samples)$
28: Shuffle $samples$ to assign $[A, B, C, D]$ and record $answer\_id$
29: $data \leftarrow$ create\_data($ref\_img, samples, question, answer\_id$)

---

**Algorithm 11** Cube Reconstruction Task

---

1: **Input:** Color(Pattern) set $C$, unit length $s$, task mode $m$
2: Initialize 11 cube nets
   $nets : \{face\_name : (i, j)|face\_name \in \{$"Top", "Bottom", "Right", "Left", "Back", "Front"$\}\}$
3: Initialize empty lists $positive\_samples, negative\_samples$
4: $map : \{face\_name : c|c \in C\} \leftarrow$ Randomly shuffle set $C$ and assign it to six faces
5: $net \in \{0, 1\}^{3 \times 5} \leftarrow$ Randomly select net from $nets$
6: $ref\_img \leftarrow$ DrawNet($net, map, s, rot =$ false) and append $img$ to $positive\_samples$
7: **for** $i \leftarrow 1$ to 3 **do**
8:    $view \leftarrow$ Randomly select a view from 8 corner views of a cube
9:    $img \leftarrow$ Net2Cube($net, map, view, rot =$ false)
10:    Append $img$ to $positive\_samples$
11: **end for**
12: **for** $flip\_dir \in \{$"horizontal", "vertical"$\}$ **do**
13:    Randomly choose $sample$ from $positive\_samples$
14:    $img \leftarrow$ flip($sample, flip\_dir$)
15:    Append $img$ to $negative\_samples$
16: **end for**

17: $samples \leftarrow (positive\_samples, negative\_samples)$
18: Shuffle $samples$ to assign $[A, B, C, D]$ and record $answer\_id$
19: $data \leftarrow$ create\_data($ref\_img, samples, question, answer\_id$)

---

---

**Algorithm 12** Cross-Section Task

---

1: **Input:** Number of objects $num$, number of sections per mode $k$, whether rotate the slicing plane $rot$
2: Initialize candidate objects list $objects$, empty list $selected\_objects$
3: Initialize empty lists $positive\_samples$, $negative\_samples$
4: **function** GETSECTIONS($compound$, $k$, $plane$)
5:     Initialize empty list $imgs$
6:     Determine $coord_{min}$ and $coord_{max}$ from bounding box
7:     $step \leftarrow (coord_{max} - coord_{min})/(k+1)$
8:     **for** $i \leftarrow 1$ to $k$ **do**
9:         $offset \leftarrow coord_{min} + i \times step$
10:         $normal\_vector \leftarrow$ unit vector normal to $plane$
11:         $section \leftarrow$ `FreeCAD.slice`($compound$, $normal\_vector$, $offset$)
12:         Rotate $section$ for better visualization
13:         $img \leftarrow$ `FreeCAD.savaImage`($section$) and append $img$ to $imgs$
14:     **end for**
15:     **return** $imgs$
16: **end function**

17: **function** GETROTATEDSECTIONS($compound$, $axis$, $center$)
18:     $axis\_vector \leftarrow$ Corresponding unit vector of $axis$
19:     $plane \leftarrow$ Parallel to $axis$
20:     **for** $angle \in \{45°, 135°\}$ **do**
21:         $axix\_vector' \leftarrow$ `rotate`($axis\_vector, angle, plane$)
22:         $offset \leftarrow axix\_vector \cdot center$
23:         $section \leftarrow$ `FreeCAD.slice`($compound$, $axis\_vector$, $offset$)
24:         Rotate $section$ for better visualization
25:         $img \leftarrow$ `FreeCAD.savaImage`($section$) and append $img$ to $imgs$
26:     **end for**
27:     **return** $imgs$
28: **end function**

29: $selected\_objects \leftarrow$ Randomly select $num$ objects from $objects$
30: Randomly assign sizes to objects in $selected\_objects$
31: $compound \leftarrow$ Create objects in FreeCAD and compound objects
32: $center \leftarrow$ Obtain the center of compound object
33: **for** $plane \in \{$"XY","XZ", "YZ"$\}$ **do**
34:     $imgs \leftarrow$ `GetSections`($compound$, $k$, $plane$)
35:     Append $imgs$ to $positive\_samples$
36: **end for**
37: **if** $rot$ is true **then**
38:     **for** $axis \in \{$"x", "y", "z"$\}$ **do**
39:         **for** $angle \in \{45°, 135°\}$ **do**
40:             $imgs \leftarrow$ `GetRotatedSections`($compound$, $axis$, $center$)
41:             Append $imgs$ to $positive\_samples$
42:         **end for**
43:     **end for**
44: **end if**
45: $compound' \leftarrow$ Randomly alter the relative ratios of objects in $compound$
46: $imgs \leftarrow$ Use any of the above approaches to obtain cross-sections of $compound'$
47:
48: Append $imgs$ to $negative\_samples$

49: $samples \leftarrow (positive\_samples, negative\_samples)$
50: Shuffle $samples$ to assign $[A, B, C, D]$ and record $answer\_id$
51: $data \leftarrow$ `create_data`($ref\_img, samples, question, answer\_id$)

---

---

**Algorithm 13** Cube Counting Task

---

1: **Input:** Spatial size $(X, Y, Z)$, cube size $s$, number of constraint views $num$
2: Initialize zero value 3D tensors $placement \in \{0\}^{Z \times Y \times X}$, empty list $cubes$
3: Initialize empty list $samples$
4: **function** DETECTGRID($view, row\_num\ col\_num$)
5:     $contours \leftarrow$ Find contours in $view$
6:     Initialize $grid$ matrix of size $row\_num \times col\_num$
7:     **for** $contour \in contours$ **do**
8:         $(x, y, w, h) \leftarrow$ Bounding rectangle of $contour$
9:         $row \leftarrow y/h, col \leftarrow x/w$
10:         **if** $row$ and $col$ within bounds **then**
11:             $grid[row][col] \leftarrow 1$
12:         **end if**
13:     **end for**
14:     **return** $grid$
15: **end function**

16: **function** GETCUBEANSWER($front, top, left, num$)
17:     $sum\_front\_col \leftarrow$ Column sums of $front$
18:     $sum\_top\_col \leftarrow$ Column sums of $top$
19:     $max\_2view \leftarrow sum\_front\_col \cdot sum\_top\_col$
20:     $min\_2view \leftarrow \text{sum}(sum\_top\_col - 1 + sum\_front\_col)$
21:     **if** $num = 2$ **then**
22:         **return** $(max\_2view, min\_2view)$
23:     **end if**

24:     $sum\_left\_col \leftarrow$ Column sums of $left$
25:     Initialize answer matrix with the same dimension as $top \in \{0\}^{H \times W}$
26:     **for** $row \leftarrow 0$ to $H - 1$ **do**
27:         **for** $col \leftarrow 0$ to $W - 1$ **do**
28:             **if** $top[row][col] = 1$ **then**
29:                 $ans[row][col] \leftarrow \text{min}(sum\_front\_col[col], sum\_left\_col[row])$
30:             **end if**
31:         **end for**
32:     **end for**
33:     $max\_3view \leftarrow \text{sum}(ans)$
34:     $sum\_top\_row \leftarrow$ Row sums of $top$
35:     $min\_3view \leftarrow \text{max}(\text{sum}(sum\_top\_row - 1 + sum\_left\_col), min\_2view)$
36:     **return** $(max\_3view, min\_3view)$
37: **end function**

38: Update $placement, cubes$ with `CreateCubes`$(X, Y, Z)$
39: Update $placement, cubes$ with `ConnectIsolatedCubes`$(X, Y)$
40: $(front\_view, top\_view, left\_view) \leftarrow$ `SaveViews`($cubes$)
41: $front\_mat, top\_mat, left\_mat \leftarrow$
    `DetectGrid`($front\_view$), `DetectGrid`($top\_view$), `DetectGrid`($left\_view$)
42: **if** $num = 2$ **then**
43:     $ref\_img \leftarrow (top\_view, front\_view)$
44:     $(max\_view, min\_view) \leftarrow$ `GetCubeAnswer`($front\_mat, top\_mat, left\_mat, 2$)
45: **else if** $num = 3$ **then**
46:     $ref\_img \leftarrow (top\_view, front\_view, left_view)$
47:     $(max\_view, min\_view) \leftarrow$ `GetCubeAnswer`($front\_mat, top\_mat, left\_mat, 3$)
48: **end if**

49: $samples \leftarrow$ Generate correct and incorrect nums based on the $min\_view$ to $max\_view$ range
50: Shuffle $samples$ to assign $[A, B, C, D]$ and record $answer\_id$
51: $data \leftarrow$ `create_data`($ref\_img, samples, question, answer\_id$)

---

---

**Algorithm 14** Functions for Splitting Cube Stack into Several Connected Parts

---

1: **function** GETNEIGHBORS($cube\_pos$, $cubes$)
2:     $(x, y, z) \leftarrow cube\_pos$
3:     Initialize empty list $neighbours$
4:     **for** $dx \in \{-1, 0, 1\}$ **do**
5:         **for** $dy \in \{-1, 0, 1\}$ **do**
6:             **for** $dz \in \{-1, 0, 1\}$ **do**
7:                 **if** $|dx| + |dy| + |dz| = 1$ **then**              ▷ 6-connected neighborhood
8:                     $neighbor\_pos \leftarrow (x + dx, y + dy, z + dz)$
9:                     **if** $neighbor\_pos \in cubes$ **then**
10:                         Append $neighbor\_pos$ to $neighbours$
11:                     **end if**
12:                 **end if**
13:             **end for**
14:         **end for**
15:     **end for**
16:     **return** $neighbors$
17: **end function**

18: **function** REGIONGROWING($cubes$, $max\_cubes$)
19:     Initialize empty set $part$, empty list $queue$
20:     $start\_pos \leftarrow$ Randomly select a position from $cubes$ and append $start\_pos$ to $queue$
21:     **while** $queue$ not empty and $|part| < max\_cubes$ **do**
22:         $current\_pos \leftarrow \text{pop}(queue, 0)$
23:         **if** $current\_pos \notin part$ **then**
24:             Add $current\_pos$ to $part$
25:             $neighbors \leftarrow \text{GetNeighbors}(current\_pos, cubes)$
26:             Extend $[n \in neighbors \mid n \notin part]$ to $queue$
27:         **end if**
28:     **end while**
29:     **return** $part$
30: **end function**

31: **function** ISCONTINUOUS($part$)
32:     Initialize empty set $part$, empty list $queue$
33:     $start\_pos \leftarrow part[0]$ and append $start\_pos$ to $queue$
34:     **while** $queue$ not empty **do**
35:         $current\_pos \leftarrow \text{pop}(queue, 0)$
36:         **if** $current\_pos \notin visited$ **then**
37:             Add $current\_pos$ to $visited$
38:             $neighbors \leftarrow \text{GetNeighbors}(current\_pos, part)$
39:             Extend $[n \in neighbors \mid n \in part \text{ and } n \notin visited]$ to $queue$
40:         **end if**
41:     **end while**
42:     **return** Whether $|visited| = |part|$
43: **end function**

44: **function** SPLITCUBES($cubes$, $max\_cubes$, $num\_parts$)
45:     $part1 \leftarrow \text{RegionGrowing}(cubes, max\_cubes)$
46:     **if** IsContinuous($part1$) **then**
47:         $remaining \leftarrow$ Remove $part1$ from $cubes$
48:     **end if**
49:     **if** IsContinuous($remaining$) **then**
50:         **if** $num\_parts = 2$ **then**
51:             **return** $\text{sort}([part1, remaining])$ by size
52:         **else if** $num\_parts = 3$ **then**
53:             Similarly find $part2$ from remaining cubes as above
54:             $part3 \leftarrow$ Remove $part2$ from $remaining$
55:             **return** $\text{sort}([part1, part2, part3])$ by size
56:         **end if**
57:     **end if**
58: **end function**

---

**Algorithm 15** Cube Assembly Task

1: **Input:** Spatial size $(X, Y, Z)$, cube size $s$, number of splitting parts $k$
2: Initialize zero value 3D tensors $placement \in \{0\}^{Z \times Y \times X}$, empty list $cubes$
3: Initialize empty lists $ref\_imgs, positive\_samples, negative\_samples$
4: **function** CREATECUBESPYRAMID($X$, $Y$, $Z$)
5:     Initialize $num = 1$
6:     **for** $y \leftarrow 0$ to $Y - 1$ **do**
7:         $num = \texttt{randomInt}(num, \min(y + 2, X))$
8:         **for** $x \leftarrow 0$ to $num - 1$ **do**
9:             $\texttt{CreateCube}(x, y, 0)$
10:         **end for**
11:     **end for**
12:     **for** $z \leftarrow 1$ to $Z - 2$ **do**
13:         Initialize $num = 0$
14:         **for** $y \leftarrow 0$ to $Y - 1$ **do**
15:             $num = \texttt{randomInt}(num, \max(num, \text{sum}(placement[z - 1][y])))$
16:             **for** $x \leftarrow 0$ to $num - 1$ **do**
17:                 $\texttt{CreateCube}(x, y, z)$
18:             **end for**
19:         **end for**
20:     **end for**
21:     **for** $y \leftarrow 0$ to $Y - 1$ **do**
22:         **for** $x \leftarrow 0$ to $X - 1$ **do**
23:             With 50% probability $\texttt{CreateCube}(x, y, Z - 1)$
24:         **end for**
25:     **end for**
26: **end function**

27: Update $placement, cubes$ with $\texttt{CreateCubesPyramid}(X, Y, Z)$
28: $cubes\_img \leftarrow \texttt{FreeCAD.saveImage}(cubes)$ and append $cubes\_img$ to $ref\_imgs$
29: $parts \leftarrow \texttt{SplitCubes}(cubes, max\_cubes, num\_parts)$
30: **for** $part \in parts[:-1]$ **do**
31:     $part\_img \leftarrow \texttt{FreeCAD.saveImage}(part)$ and append $part\_img$ to $ref\_imgs$
32: **end for**
33: $part\_img \leftarrow \texttt{FreeCAD.saveImage}(parts[-1])$ and append $part\_img$ to $positive\_samples$
34: **for** $i \leftarrow 1$ to 2 **do**
35:     $part' \leftarrow$ Randomly remove 1 cube from part[-1]
36:     $part'\_img \leftarrow \texttt{FreeCAD.saveImage}(part')$ and append $part\_img$ to $negative\_samples$
37: **end for**

38: $samples \leftarrow (positive\_samples, negative\_samples)$
39: Shuffle $samples$ to assign $[A, B, C, D]$ and record $answer\_id$
40: $data \leftarrow \texttt{create\_data}(ref\_img, samples, question, answer\_id)$

---

**Algorithm 16** Simulation for Arrow Moving

---

1: **Class** ArrowPath
2: **Attributes:**
3:    $W, H, k$: Map width, height, and step count
4:    $max\_step \leftarrow \min(x, y)$
5:    $directions \leftarrow \{(0,1),(1,0),(0,-1),(-1,0)\}$                        ▷ up, right, down, left
6:    $path$: Initialize with empty list to record relative moving direction and steps
7:    $states$: Initialize with empty list to record pos and orientation during transformation
8: **function** INITIALIZESTATE
9:    Reset $path$, $states$
10:    $orient\_id \leftarrow \texttt{randomInt}(0, 3)$
11:    $pos \in \{(x, y)\} \leftarrow$ Randomly select a position in the map
12:    Append $(orient\_id, pos)$ to $states$
13: **end function**

14: **function** GETRELATIVEDIRECTION($orient\_id$)
15:    $forward \leftarrow directions[orient\_id]$
16:    $backward \leftarrow (-forward[0], -forward[1])$
17:    $left \leftarrow directions[(orient\_id - 1) \mod 4]$
18:    $right \leftarrow directions[(orient\_id + 1) \mod 4]$
19:    **return** {"forward":forward, "backward":backward, "left":left, "right":right}
20: **end function**

21: **function** UPDATEORIENTID($rel\_dir$, $orient\_id$)
22:    **if** $rel\_dir$ is "backward" **then**
23:       $orient\_id \leftarrow (orient\_id + 2) \mod 4$
24:    **else if** $rel\_dir$ is "left" **then**
25:       $orient\_id \leftarrow (orient\_id - 1) \mod 4$
26:    **else if** $rel\_dir$ is "right" **then**
27:       $orient\_id \leftarrow (orient\_id + 1) \mod 4$
28:    **end if**
29:    **return** $orient\_id$
30: **end function**

31: **function** MOVE($state$, $rel\_dir$, $steps$)
32:    $pos, orient\_id \leftarrow state$
33:    $move\_dir \leftarrow \texttt{GetRelativeDirection}(orient\_id)[rel\_dir]$
34:    $new\_pos \leftarrow [pos[0] + move\_dir[0] \times steps, pos[1] + move\_dir[1] \times steps]$
35:    **if** $new\_pos$ is invalid **then**
36:       **return** false
37:    **end if**
38:    Append $(rel\_dir, steps)$ to $path$
39:    Append $(\texttt{UpdateOrientId}(rel\_dir, orient\_id), new\_pos)$ to $states$
40:    **return** true
41: **end function**

42: **function** GENERATEPATH($k$, $end\_state$=None)
43:    **for** $i \leftarrow 1$ to $k$ **do**
44:       **repeat**
45:          Randomly select $rel\_dir \in \{$"forward", "backward", "left", "right"$\}$
46:          $steps \leftarrow \texttt{randomInt}(1, max\_step)$
47:          $valid\_flag \leftarrow \texttt{Move}(states[-1], rel\_dir, steps)$
48:          **if** $end\_state$ is not None and $i = k$ **then**
49:             $valid\_flag \leftarrow valid\_flag \;\&\; state[-1] \neq end\_state$
50:          **end if**
51:       **until** $valid\_flag$ is true
52:    **end for**
53: **end function**

---

---

**Algorithm 17** Simulation for Arrows Moving

---

1: **Class** ArrowMap(Inherit from **Class** ArrowPath)
2: **Attributes:**
3:    $colors$: Color set
4:    $path$: Initialize with empty list to record arrow position, relative moving direction and steps
5:    $states$: Initialize with empty list to record map during transformation
6: **function** INITIALIZESTATE
7:    Initialize empty matrix $state$
8:    **for** $y \leftarrow 1$ to $H$ **do**
9:       **for** $x \leftarrow 1$ to $W$ **do**
10:          With 50% probability:
11:          Randomly select $color \in colors$
12:          Randomly get $orient\_id \leftarrow \texttt{randomInt}(0, 3)$
13:          $state[pos] \leftarrow$ Record $color$ and $orient\_id$ at $pos(x, y)$
14:       **end for**
15:    **end for**
16:    Append $state$ to $states$
17: **end function**

18: **function** MOVE($state, arrow\_pos, rel\_dir, steps$)
19:    $curr\_pos \leftarrow arrow\_pos$
20:    $curr\_orient\_id, curr\_color \leftarrow state[x][y]$
21:    $move\_dir \leftarrow \texttt{GetRelativeDirection}(curr\_orient\_id)[rel\_dir]$
22:    $new\_pos \leftarrow [pos[0] + move\_dir[0] \times steps, pos[1] + move\_dir[1] \times steps]$
23:    **if** $new\_pos$ is invalid **then**
24:       **return** false
25:    **end if**
26:    $new\_orient\_id \leftarrow \texttt{UpdateOrientId}(rel\_dir, orient\_id)$
27:    **if** $new\_pos = curr\_pos$ and $new\_orient\_id = curr\_orient\_id$ **then**
28:       **return** false
29:    **end if**
30:    Append $arrow\_pos, rel\_dir, steps$ to $path$
31:    **if** $state[new\_pos]$ is None **then**
32:       $state[curr\_pos] \leftarrow$ None
33:    **else**
34:       $target\_color, target\_orient\_id \leftarrow state[new\_pos]$
35:       $target\_move\_dir \leftarrow -move\_dir$
36:       $target\_rel\_directions \leftarrow \texttt{GetRelativeDirection}(target\_orient\_id)$
37:       $taget\_rel\_dir \leftarrow$ Find $\{key \in target\_rel\_directions \mid value = target\_move\_dir\}$
38:       $new\_target\_orient\_id \leftarrow \texttt{UpdateOrientId}(taget\_rel\_dir, target\_orient\_id)$
39:       $state[curr\_pos] \leftarrow target\_color$ and $new\_target\_orient\_id$
40:    **end if**
41:    $state[new\_pos] \leftarrow curr\_color$ and $curr\_orient\_id$
42:    **return** true
43: **end function**

44: **function** GENERATEPATH($k, end\_state$=None)
45:    **for** $i \leftarrow 1$ to $k$ **do**
46:       **repeat**
47:          Randomly select $arrow\_pos \in \{pos \mid state[pos]$ is not None$\}$
48:          Randomly select $rel\_dir \in \{$"forward", "backward", "left", "right"$\}$
49:          $steps \leftarrow \texttt{randomInt}(1, max\_step)$
50:          $valid\_flag \leftarrow \texttt{Move}(state, arrow\_pos, rel\_dir, steps)$
51:          **if** $end\_state$ is not None and $i = k$ **then**
52:             $valid\_flag \leftarrow valid\_flag$ & $state[-1] \neq end\_state$
53:          **end if**
54:       **until** $valid\_flag$ is true
55:    **end for**
56: **end function**

---

---

**Algorithm 18** Arrow Moving Task in Easy Version

---

1: **Input:** Dimension of map $(W, H)$, step count $k$
2: Initialize empty lists $positive\_samples$, $negative\_samples$
3: Initialize $arrow\_path$ with dimension $W \times H$
4: Initialize state with $arrow\_path.\texttt{InitializeState}()$ and record as $initial\_state$
5: Update $path$, $states$ with $arrow\_path.\texttt{GeneratePath}(k)$
6: Append $path$ to $positive\_samples$
7: $ref\_img \leftarrow \texttt{draw\_map}(states[0], states[-1])$
8: Record $end\_state \leftarrow states[-1]$

9: From the same $initial\_state$
10: **for** $i \leftarrow 1$ to 3 **do**
11:     Update $path'$ with $arrow\_path.\texttt{GeneratePath}(k, end\_state)$
12:     Append $path'$ to $negative\_samples$
13: **end for**

14: $samples \leftarrow (positive\_samples, negative\_samples)$
15: Shuffle $samples$ to assign $[A, B, C, D]$ and record $answer\_id$
16: $data \leftarrow \texttt{create\_data}(ref\_img, samples, question, answer\_id)$

---

**Algorithm 19** Arrow Moving Task in Hard Version

---

1: **Input:** Dimension of map $(W, H)$, step count $k$, task mode $m$
2: Initialize empty lists $positive\_samples$, $negative\_samples$
3: Initialize $arrow\_map$ with dimension $W \times H$
4: Initialize state with $arrow\_map.\texttt{InitializeState}()$ and record as $initial\_state$
5: Update $path$, $states$ with $arrow\_map.\texttt{GeneratePath}(k)$
6: Append $path$ to $positive\_samples$
7: **if** $m = $ "state" **then**
8:     $ref\_img \leftarrow \texttt{draw\_map}(states[0])$
9:     Append $states[-1]$ to $positive\_samples$
10: **else if** $m = $ "path" **then**
11:     $ref\_img \leftarrow \texttt{draw\_map}(states[0], state[-1])$
12:     Append $path$ to $positive\_samples$
13: **end if**
14: Record $end\_state \leftarrow states[-1]$

15: From the same $initial\_state$
16: **for** $i \leftarrow 1$ to 3 **do**
17:     Update $path'$, $states'$ with $arrow\_map.\texttt{GeneratePath}(k, end\_state)$
18:     **if** $m = $ "state" **then**
19:         Append $states'[-1]$ to $negative\_samples$
20:     **else if** $m = $ "path" **then**
21:         Append $path'$ to $negative\_samples$
22:     **end if**
23: **end for**

24: $samples \leftarrow (positive\_samples, negative\_samples)$
25: Shuffle $samples$ to assign $[A, B, C, D]$ and record $answer\_id$
26: $data \leftarrow \texttt{create\_data}(ref\_img, samples, question, answer\_id)$

---

---

**Algorithm 20** Simulation for Block Moving

---

1: **Class** Block
2: **Attributes:**
3:     $X, Y, Z, k$: Spatial size and step count
4:     $directions$: 6 directions
5:     $colors$: Color set
6:     $cubes\_info$: Initialize with empty list to record positions and colors of cube objects
7:     $transformation$: Initialize with empty list to record transformations
8: **function** INITIALIZESTATE
9:     Update $cubes$ with `CreateCubes`$(X, Y, Z)$
10:     Assign randomly selected colors to $cubes$ and record their colors and positions in $cubes\_info$
11: **end function**

12: **function** HASSUPPORT($x$, $y$, $z$)
13:     **if** $z = 0$ or there is cube at $(x, y, z - 1)$ **then**
14:         **return** true
15:     **end if**
16:     **return** flase
17: **end function**

18: **function** DROPCUBES
19:     Sort $cubes\_info$ by $z$ of $pos$ in ascending order
20:     **for** $cube \in cubes\_info$ **do**
21:         $(x, y, z) \leftarrow$ Acquire position of $cube$ from $cubes\_info$
22:         **while** `HasSupport`$(x, y, z)$ is flase **do**
23:             Change the position of $cube$ to $(x, y, z - 1)$ and update $z \leftarrow z - 1$
24:         **end while**
25:     **end for**
26: **end function**

27: **function** CHECKMOVE($from\_pos$, $to\_pos$)
28:     **if** ($to\_pos$ is invalid) or (`HasSupport`($to\_pos$) is false) or (there is no cube at $from\_pos$) or (there is no cube at $to\_pos$ and $to\_pos$ is on top of $from\_pos$) **then**
29:         **return** false
30:     **end if**
31:     **return** true
32: **end function**

33: **function** MOVECUBE($from\_pos$, $to\_pos$)
34:     **if** there is no cube at $to\_pos$ **then**
35:         Update $cubes\_info$ with changing the position of $cube$ at $from\_pos$ to $to\_pos$
36:     **else**
37:         Update $cubes\_info$ with swapping the cube at $from\_pos$ and $to\_pos$
38:     **end if**
39:     `DropCubes()`
40:     Append $(from\_pos, to\_pos - from\_pos)$ to $transformation$
41: **end function**

42: **function** GENERATETRANSFORMATION($k$)
43:     **for** $i \leftarrow 1$ to $k$ **do**
44:         Initialize empty list $possible\_moves$
45:         **for all** $cube \in cubes\_info$ **do**
46:             **for all** $direction \in directions$ **do**
47:                 $to\_pos \leftarrow$ The position of cube $from\_pos + direction$
48:                 **if** `CheckMove`($from\_pos$, $to\_pos$) is true **then**
49:                     Append $(from\_pos, direction, to\_pos)$ to $possible\_moves$
50:                 **end if**
51:             **end for**
52:         **end for**
53:         Randomly select $(from\_pos, direction, to\_pos) \in possible\_moves$
54:         `MoveCube`($from\_pos$, $to\_pos$)
55:     **end for**
56: **end function**

---

---

**Algorithm 21** Block Moving Task

---

1: **Input:** Spatial size $(X, Y, Z)$, step count $k$
2: Initialize empty lists $ref\_imgs$, $positive\_samples$, $negative\_samples$
3: Initialize $block$ with size $(X, Y, Z)$
4: Initialize with $block$.`InitializeState()` and record as $initial\_cubes\_info$
5: $img \leftarrow$ `FreeCAD.saveImage`($initial\_cubes$) and append $img$ to $ref\_imgs$

6: Update $transformation$, $cubes\_info$ with $block$.`GenerateTransformation`($k$)
7: Append $transformation$ to $positive\_samples$
8: Record $final\_cubes\_info$ after transformation
9: $img \leftarrow$ `FreeCAD.saveImage`($final\_cubes$) and append $img$ to $ref\_imgs$

10: From the same $initial\_cubes\_info$
11: **for** $i \leftarrow 1$ to 3 **do**
12:     **repeat**
13:         Update $transformation'$, $cubes\_info'$ with $block$.`GenerateTransformation`($k$)
14:     **until** $cubes\_info \neq final\_cubes\_info$
15:     Append $transformation$ to $negative\_samples$
16: **end for**

17: $samples \leftarrow (positive\_samples, negative\_samples)$
18: Shuffle $samples$ to assign $[A, B, C, D]$ and record $answer\_id$
19: $data \leftarrow$ `create_data`($ref\_imgs$, $samples$, $question$, $answer\_id$)

---

## C  DATASET CHARACTERISTIC

**Option Modality & Format** A significant majority of questions (818) feature image-based options to emphasize visual reasoning. The choice formats are intentionally varied, including standard A/B/C/D choices (508 questions), options with A/B/C/'All three other options are incorrect' (310 questions), and unique text (242 questions) or numeric (120 questions) answers to prevent models from overfitting to a single question style. For the numeric answers, we additionally provide direct numerical responses, and in F.3 we present a comparative analysis of model performance across different question format.

**Answer Distribution** The answer distribution is well-balanced across options A (26.5%), B (27.5%), and C (28.5%). The lower frequency of option D (17.5%) is a deliberate design choice to enhance the rigor of the evaluation. For many complex tasks, option D serves the distinct role of "All three other options are incorrect". This asymmetrical design is critical for two reasons. First, it acknowledges the difficulty of generating multiple high-quality distractors for complex 3D tasks, ensuring all visual options remain challenging. Second, it compels models to move beyond simple heuristics like "pick the most similar". Instead, this approach demands eliminative reasoning, requiring the model to rule out every other option to prove a genuine understanding of the spatial rules being tested.

# D    DATA EXAMPLES

We present exemplars of varying difficulty levels for all tasks, with each sample containing an image, question, options, answer, and explanation.

**Mental Rotation** 2DRotation: Figure 8, 3DRotation: Figure 9, 3ViewProjection: Figure 10;

**Mental Folding** PaperFolding: Figure 11, CubeUnfolding: Figure 12, CubeReconstruction: Figure 13;

**Visual Penetration** CrossSection: Figure 14, CubeCounting: Figure 15, CubeAssembly: Figure 16;

**Mental Animation** ArrowMoving: Figure 17, BlockMoving: Figure 18, MechanicalSystem: Figure 19.

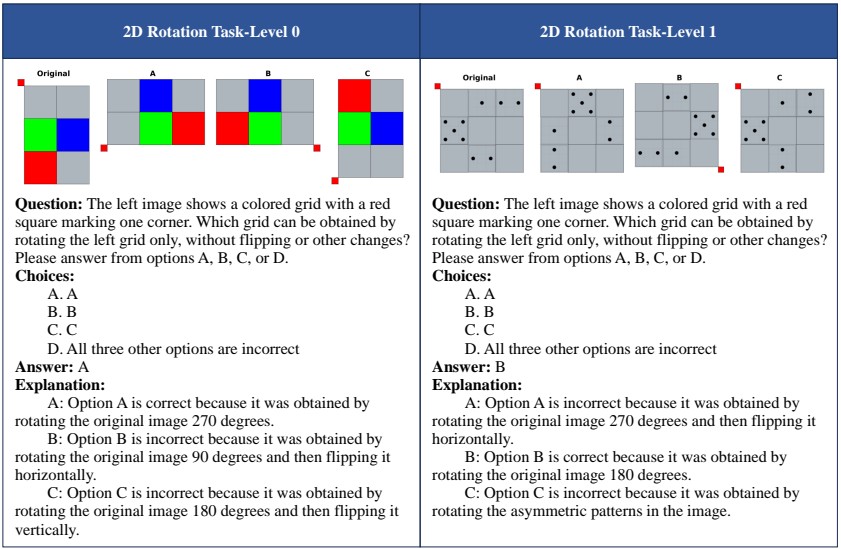

Figure 8: 2D Rotation Task.

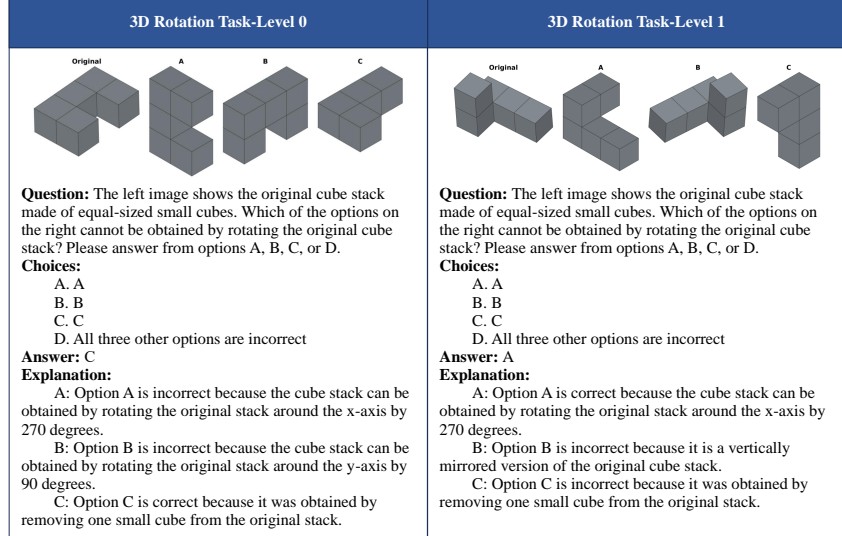

Figure 9: 3D Rotation Task.

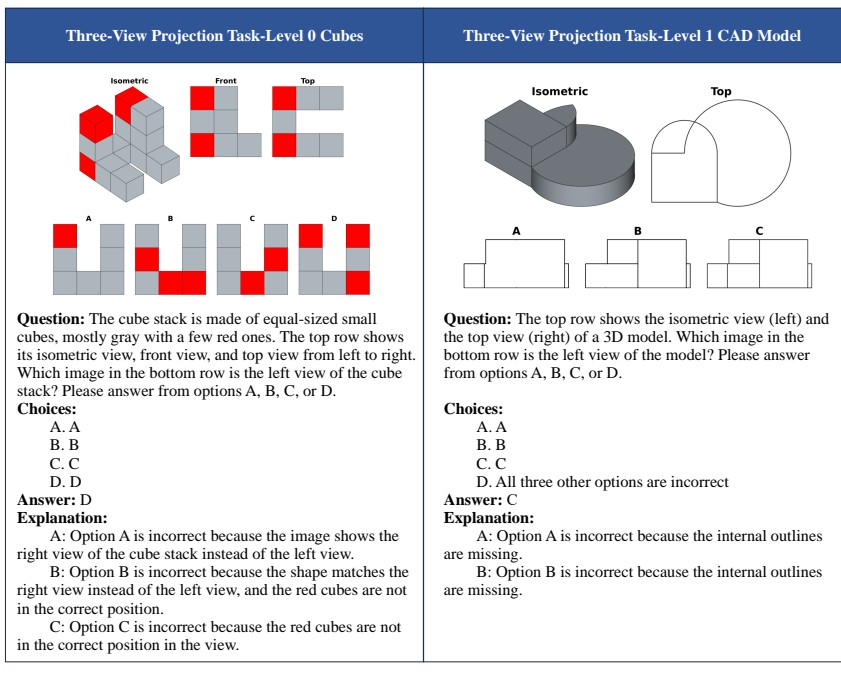

Figure 10: Three-view Projection Task.

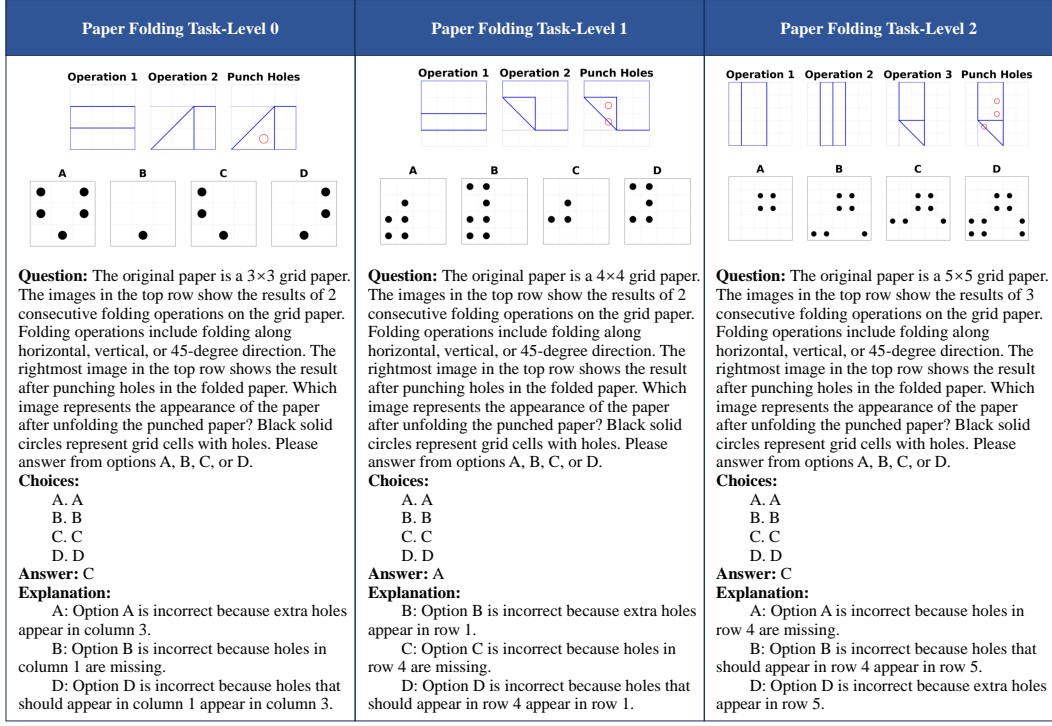

Figure 11: Paper Folding Task.

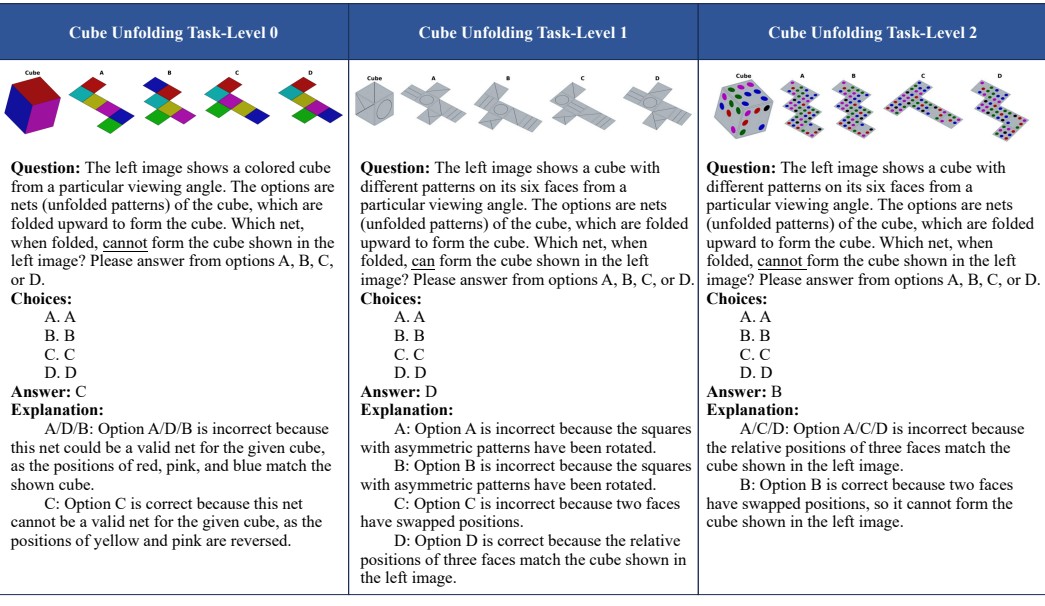

| Cube Unfolding Task-Level 0 | Cube Unfolding Task-Level 1 | Cube Unfolding Task-Level 2 |
|---|---|---|
| **Question:** The left image shows a colored cube from a particular viewing angle. The options are nets (unfolded patterns) of the cube, which are folded upward to form the cube. Which net, when folded, cannot form the cube shown in the left image? Please answer from options A, B, C, or D.
**Choices:**
    A. A
    B. B
    C. C
    D. D
**Answer:** C
**Explanation:**
    A/D/B: Option A/D/B is incorrect because this net could be a valid net for the given cube, as the positions of red, pink, and blue match the shown cube.
    C: Option C is correct because this net cannot be a valid net for the given cube, as the positions of yellow and pink are reversed. | **Question:** The left image shows a cube with different patterns on its six faces from a particular viewing angle. The options are nets (unfolded patterns) of the cube, which are folded upward to form the cube. Which net, when folded, can form the cube shown in the left image? Please answer from options A, B, C, or D.
**Choices:**
    A. A
    B. B
    C. C
    D. D
**Answer:** D
**Explanation:**
    A: Option A is incorrect because the squares with asymmetric patterns have been rotated.
    B: Option B is incorrect because the squares with asymmetric patterns have been rotated.
    C: Option C is incorrect because two faces have swapped positions.
    D: Option D is correct because the relative positions of three faces match the cube shown in the left image. | **Question:** The left image shows a cube with different patterns on its six faces from a particular viewing angle. The options are nets (unfolded patterns) of the cube, which are folded upward to form the cube. Which net, when folded, cannot form the cube shown in the left image? Please answer from options A, B, C, or D.
**Choices:**
    A. A
    B. B
    C. C
    D. D
**Answer:** B
**Explanation:**
    A/C/D: Option A/C/D is incorrect because the relative positions of three faces match the cube shown in the left image.
    B: Option B is correct because two faces have swapped positions, so it cannot form the cube shown in the left image. |

Figure 12: Cube Unfolding Task.

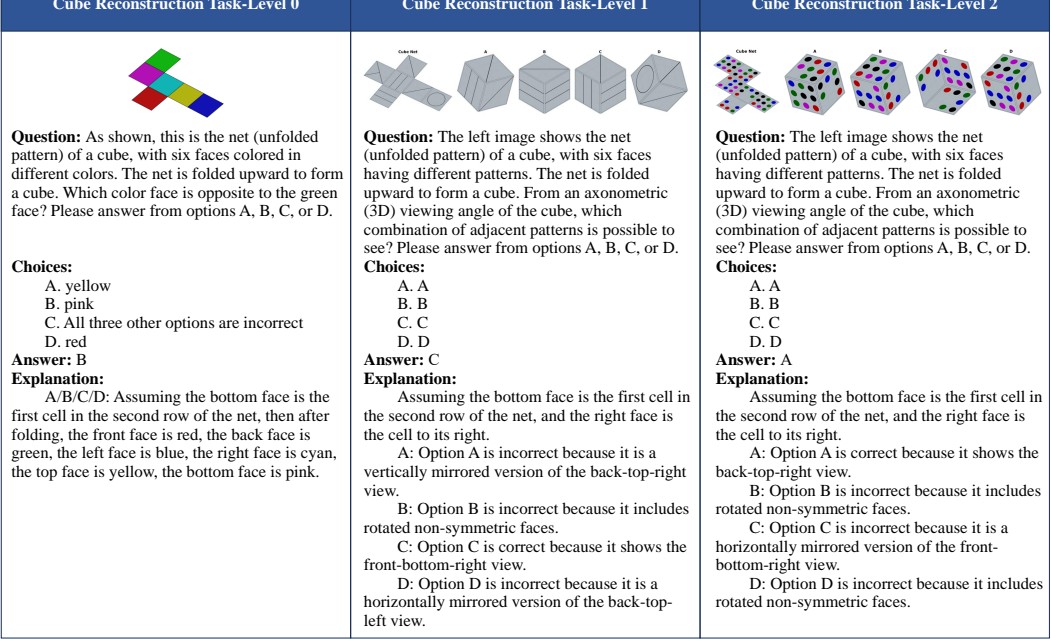

| Cube Reconstruction Task-Level 0 | Cube Reconstruction Task-Level 1 | Cube Reconstruction Task-Level 2 |
|---|---|---|
| **Question:** As shown, this is the net (unfolded pattern) of a cube, with six faces colored in different colors. The net is folded upward to form a cube. Which color face is opposite to the green face? Please answer from options A, B, C, or D.

**Choices:**
    A. yellow
    B. pink
    C. All three other options are incorrect
    D. red
**Answer:** B
**Explanation:**
    A/B/C/D: Assuming the bottom face is the first cell in the second row of the net, then after folding, the front face is red, the back face is green, the left face is blue, the right face is cyan, the top face is yellow, the bottom face is pink. | **Question:** The left image shows the net (unfolded pattern) of a cube, with six faces having different patterns. The net is folded upward to form a cube. From an axonometric (3D) viewing angle of the cube, which combination of adjacent patterns is possible to see? Please answer from options A, B, C, or D.
**Choices:**
    A. A
    B. B
    C. C
    D. D
**Answer:** C
**Explanation:**
    Assuming the bottom face is the first cell in the second row of the net, and the right face is the cell to its right.
    A: Option A is incorrect because it is a vertically mirrored version of the back-top-right view.
    B: Option B is incorrect because it includes rotated non-symmetric faces.
    C: Option C is correct because it shows the front-bottom-right view.
    D: Option D is incorrect because it is a horizontally mirrored version of the back-top-left view. | **Question:** The left image shows the net (unfolded pattern) of a cube, with six faces having different patterns. The net is folded upward to form a cube. From an axonometric (3D) viewing angle of the cube, which combination of adjacent patterns is possible to see? Please answer from options A, B, C, or D.
**Choices:**
    A. A
    B. B
    C. C
    D. D
**Answer:** A
**Explanation:**
    Assuming the bottom face is the first cell in the second row of the net, and the right face is the cell to its right.
    A: Option A is correct because it shows the back-top-right view.
    B: Option B is incorrect because it includes rotated non-symmetric faces.
    C: Option C is incorrect because it is a horizontally mirrored version of the front-bottom-right view.
    D: Option D is incorrect because it includes rotated non-symmetric faces. |

Figure 13: Cube Reconstruction Task.

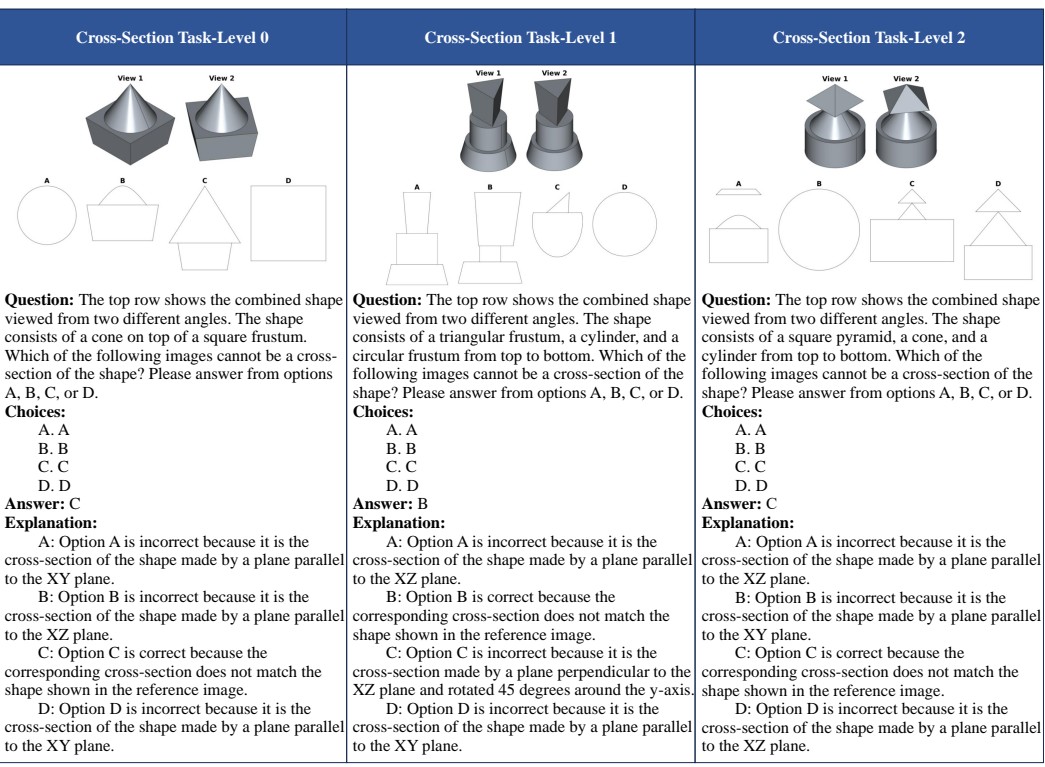

| Cross-Section Task-Level 0 | Cross-Section Task-Level 1 | Cross-Section Task-Level 2 |
|---|---|---|
| **Question:** The top row shows the combined shape viewed from two different angles. The shape consists of a cone on top of a square frustum. Which of the following images cannot be a cross-section of the shape? Please answer from options A, B, C, or D. | **Question:** The top row shows the combined shape viewed from two different angles. The shape consists of a triangular frustum, a cylinder, and a circular frustum from top to bottom. Which of the following images cannot be a cross-section of the shape? Please answer from options A, B, C, or D. | **Question:** The top row shows the combined shape viewed from two different angles. The shape consists of a square pyramid, a cone, and a cylinder from top to bottom. Which of the following images cannot be a cross-section of the shape? Please answer from options A, B, C, or D. |
| **Choices:** A. A  B. B  C. C  D. D | **Choices:** A. A  B. B  C. C  D. D | **Choices:** A. A  B. B  C. C  D. D |
| **Answer:** C | **Answer:** B | **Answer:** C |
| **Explanation:** A: Option A is incorrect because it is the cross-section of the shape made by a plane parallel to the XY plane. B: Option B is incorrect because it is the cross-section of the shape made by a plane parallel to the XZ plane. C: Option C is correct because the corresponding cross-section does not match the shape shown in the reference image. D: Option D is incorrect because it is the cross-section of the shape made by a plane parallel to the XY plane. | **Explanation:** A: Option A is incorrect because it is the cross-section of the shape made by a plane parallel to the XZ plane. B: Option B is correct because the corresponding cross-section does not match the shape shown in the reference image. C: Option C is incorrect because it is the cross-section made by a plane perpendicular to the XZ plane and rotated 45 degrees around the y-axis. D: Option D is incorrect because it is the cross-section of the shape made by a plane parallel to the XY plane. | **Explanation:** A: Option A is incorrect because it is the cross-section of the shape made by a plane parallel to the XZ plane. B: Option B is incorrect because it is the cross-section of the shape made by a plane parallel to the XY plane. C: Option C is correct because the corresponding cross-section does not match the shape shown in the reference image. D: Option D is incorrect because it is the cross-section of the shape made by a plane parallel to the XZ plane. |

Figure 14: Cross-sectionn Task.

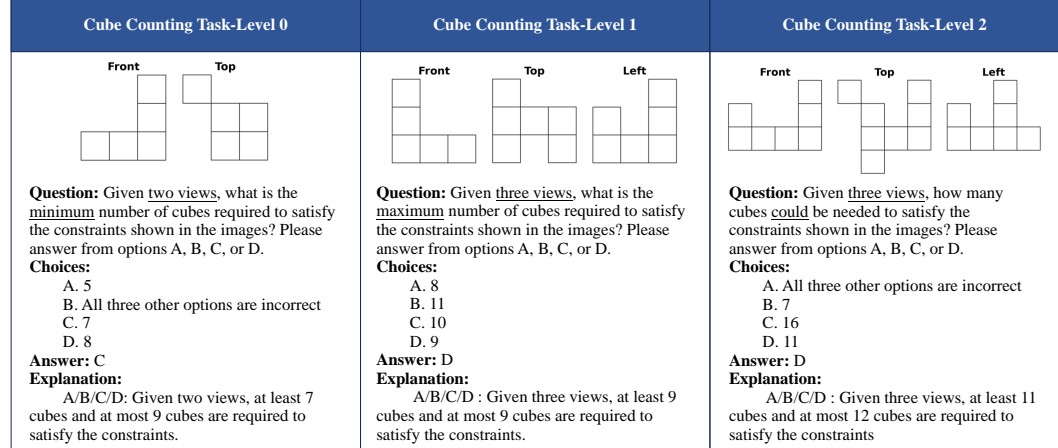

| Cube Counting Task-Level 0 | Cube Counting Task-Level 1 | Cube Counting Task-Level 2 |
|---|---|---|
| **Question:** Given two views, what is the minimum number of cubes required to satisfy the constraints shown in the images? Please answer from options A, B, C, or D. | **Question:** Given three views, what is the maximum number of cubes required to satisfy the constraints shown in the images? Please answer from options A, B, C, or D. | **Question:** Given three views, how many cubes could be needed to satisfy the constraints shown in the images? Please answer from options A, B, C, or D. |
| **Choices:** A. 5  B. All three other options are incorrect  C. 7  D. 8 | **Choices:** A. 8  B. 11  C. 10  D. 9 | **Choices:** A. All three other options are incorrect  B. 7  C. 16  D. 11 |
| **Answer:** C | **Answer:** D | **Answer:** D |
| **Explanation:** A/B/C/D: Given two views, at least 7 cubes and at most 9 cubes are required to satisfy the constraints. | **Explanation:** A/B/C/D : Given three views, at least 9 cubes and at most 9 cubes are required to satisfy the constraints. | **Explanation:** A/B/C/D : Given three views, at least 11 cubes and at most 12 cubes are required to satisfy the constraints |

Figure 15: Cube Counting Task.

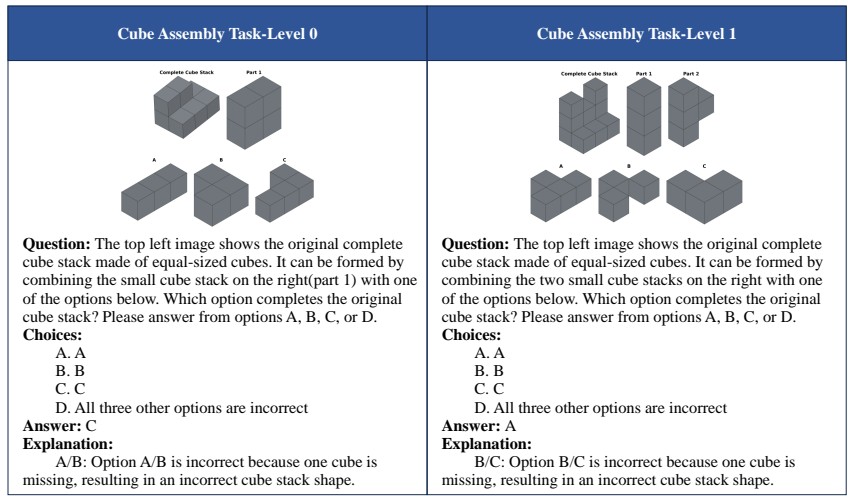

Figure 16: Cube Assembly Task.

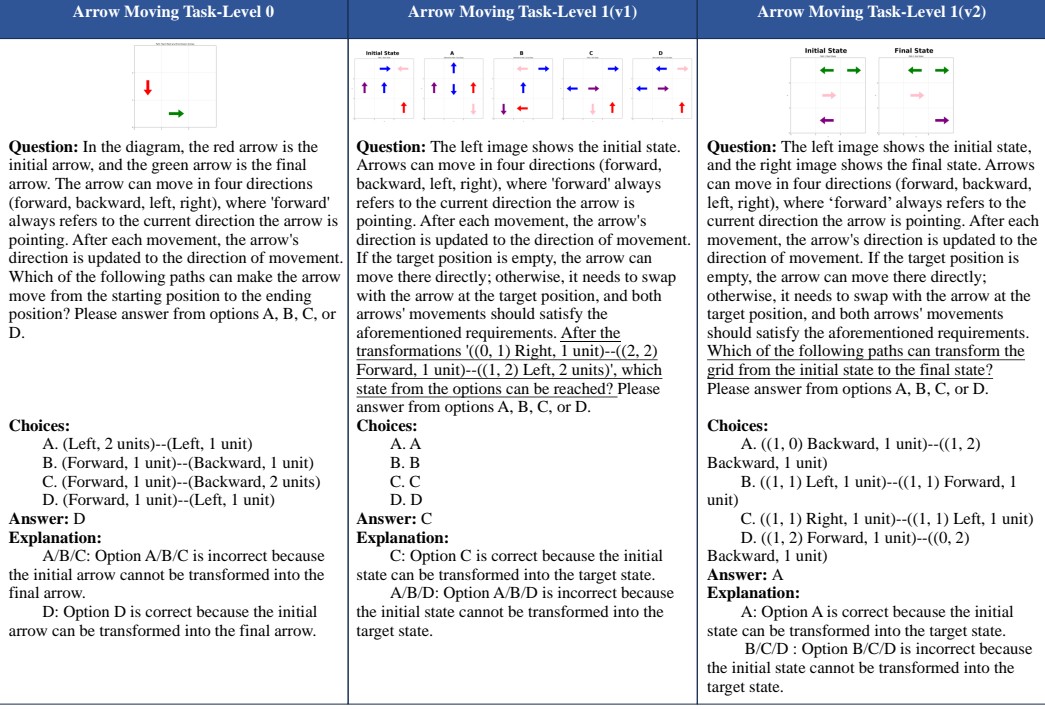

Figure 17: Arrow Moving Task.

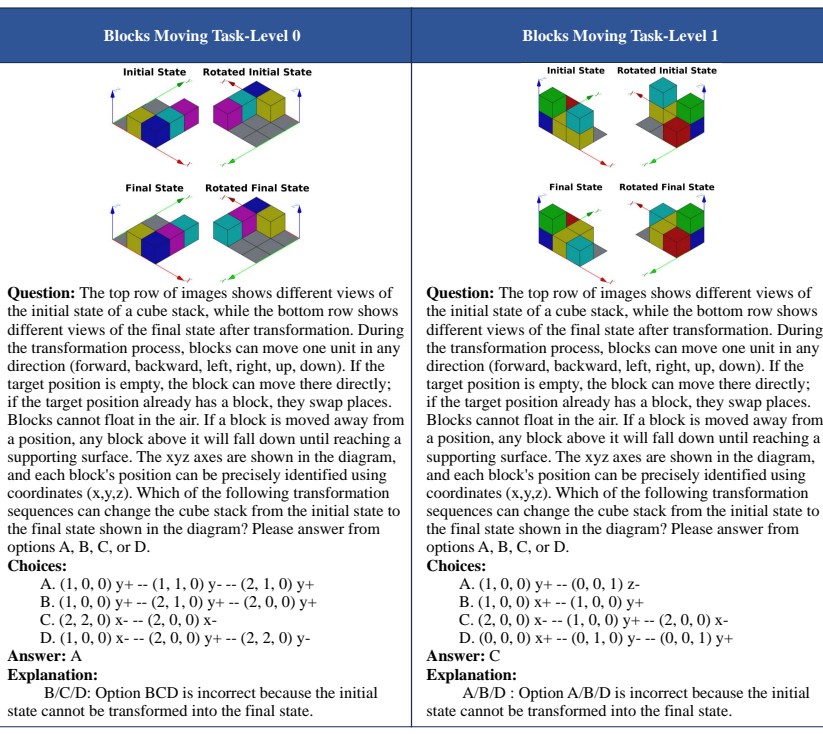

Figure 18: Block Moving Task.

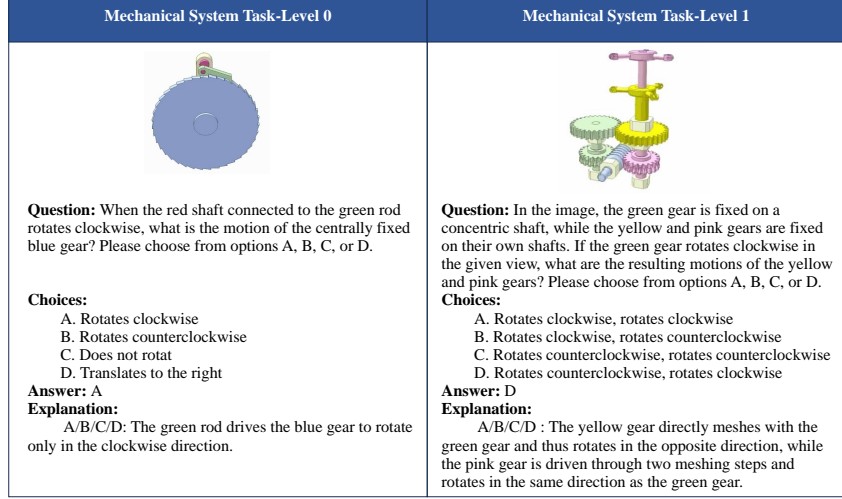

Figure 19: Mechanical System Task.

# E    EVALUATION DETAILS

## E.1    MODELS

For the DeepseekVL2 series, InternVL2.5 series, InternVL3 series and SAIL-VL series, we deployed these models on H100 servers and used the officially provided code to load the pre-trained models for inference. For all other models, we employed API calls through OpenAI's client service for inference. All closed-source models accessed via API in this study were used with specific, identifiable versions to ensure consistency and reproducibility. Specifically, we used the following model versions:

- `gpt-4o-2024-08-06` for GPT-4o
- `o1-2024-12-17` for o1
- `claude-3-5-sonnet-20240620` for Claude-3.5-Sonnet
- `claude-3-7-sonnet-20250219` for Claude-3.7-Sonnet
- `Gemini-2.5-flash-preview-04-17` for Gemini-2.5-flash
- `Gemini-2.5-pro-preview-03-25` for Gemini-2.5-pro
- `Doubao-1-5-vision-pro-32k-250115` for Doubao-1-5-vision-pro
- `qwen-vl-max-0408` for Qwen-VL-max

## E.2    PROMPTS FOR RESPONSE GENERATION

We use the prompt template as follows:

1) **Original CoT Prompt A** from DeepSeek-R1(DeepSeek-AI et al., 2025): "You should first provide a reasoning process, then provide a single option (A, B, C or D) as the final answer. The reasoning process and the answer are enclosed within <think></think> and <answer></answer> tags, respectively, i.e., <think>reasoning process</think>, <answer>answer</answer>.\nQuestion: **<question here>**\nA.**<option A here>**\nB.**<option B here>**\nC.**<option C here>**\nD.**<option D here>**\n"

2) **Variant CoT Prompt B** from EMMA(Hao et al., 2025): "Answer with the option's letter from the given choices and put the letter in one '\boxed'. Please solve the problem step by step.\nQuestion: <question here>\nA.<option A here>\nB.<option B here>\nC.<option C here>\nD.<option D here>\n"

3) **Non-CoT Prompt:**   "Answer with a single option letter (A, B, C, or D), enclosed within the <answer></answer> tag. For example: <answer>A</answer>. Ensure that your output contains only the final answer, without any intermediate reasoning or additional content.\nQuestion: **<question here>**\nA.**<option A here>**\nB.**<option B here>**\nC.**<option C here>**\nD.**<option D here>**\n"

## E.3    ZERO-SHOT SETTING

Our decision to focus exclusively on the zero-shot evaluation setting is grounded in both methodological precedent and practical considerations. This approach aligns with the standards set by many recent, high-impact benchmark papers, such as Math-Vision (Wang et al., 2024), MM-IQ (Cai et al., 2025), and EMMA (Hao et al., 2025), all of which centered their evaluations on the zero-shot setting to assess novel reasoning capabilities. While we considered few-shot prompting, we concluded its utility is limited in our context of complex spatial reasoning. For these intricate visualization tasks, providing examples with only the final answer offers minimal effective guidance. On the other hand, creating effective chain-of-thought examples that include complete, multi-step reasoning would be prohibitively expensive for comprehensive benchmarking.

## E.4    METHODS FOR ANSWER EXTRACTION

To ensure robust evaluation and minimize parsing errors, we employ a hierarchical, two-stage rule-based approach for answer extraction.

**Stage 1: Coarse Extraction with Boundary Enforcement.**
Adopting the strategy from MME-CoT (Jiang et al., 2025), we first attempt to locate the answer

segment by scanning for a comprehensive set of standard identifiers, including XML-style tags (e.g., `<answer></answer>`) and natural language markers (e.g., "<answer>", "Answer:", "Final answer", "final answer", "Final Answer", "the answer is", "The answer is", "correct answer", "Correct answer", "Correct Answer", and "correct path"). The text following these markers is isolated and truncated at the first subsequent period delimiter. Critically, to prevent false positives where common words starting with option letters (e.g., "All", "Backward") are mistakenly identified as answers, we enforce strict word boundary constraints. We utilize the regular expression `\b([A-D])\b` to accept only standalone option letters.

**Stage 2: Prioritized Fine-Grained Matching.**
In instances where the coarse extraction fails to yield a valid option, we trigger a secondary, high-precision extraction routine. This process iterates through a prioritized list of compiled regular expression patterns designed to handle specific formatting variations (e.g., tagged encapsulated outputs, boxed answers) and semantic fallback structures. The patterns are applied in the following order:

- **CoT Prompt A with tags:**
  ```
  r"<answer>\s*(?P<value>.*?)\s*</answer>"
  ```

- **CoT Prompt B with boxes:**
  ```
  r"\\{1,2}boxed{(?:(?:\\text|rm){)?(?P<value>[A-D])"
  ```

- **Other common answer formats:**
  ```
  r"<answer>\s*option\s+(?P<value>[A-D])(?=</answer>")
  r"(?:final|correct)\s+answer\s*(?:is:)\s*(?:option\s*)?(?P<value>[A-D])\b"
  r"option\s+(?P<value>[A-D])\b"
  r"choose\s+(?P<value>[A-D])\b"
  ```

This dual-layer approach ensures high recall for compliant responses while maintaining precision against hallucinated or verbose outputs. Even with these rules, 100% parsing success isn't guaranteed, as models can still flexibly produce outputs in non-standard formats. For the purpose of our comparative analysis, we designate the baseline coarse extraction method (excluding strict boundary enforcement) as **Extract Rule A**, and the comprehensive dual-stage strategy described herein as **Extract Rule B**.

For multiple-choice questions, a response is considered correct if and only if the extracted result contains exactly one uppercase option letter (A, B, C, or D) matching the standard answer. For non-choice questions, we perform direct string matching between the extracted result and the reference answer. This hybrid rule-based evaluation ensures consistent and fair judgment across both option-based and open-form tasks.

### E.5    HUMAN PERFORMANCE

To establish a robust human baseline analogous to the tested MLLMs, we recruited 8 graduate students (4 Ph.D., 4 M.S.; aged 22-27) from mechanical engineering and computer science. All participants possessed strong backgrounds in geometry and physics, confirmed through their academic curriculum, and reported familiarity with spatial reasoning tasks. This selection criterion was chosen because it mirrors the specialized knowledge domains inherent in the models' training data. Participants were compensated at the standard rate for graduate research assistants.

To ensure data quality and minimize the impact of cognitive fatigue and time constraints, we curated a representative subset of the benchmark for the evaluation. Specifically, we randomly sampled 6 problems from each of the 12 task categories, resulting in a total of 72 problems per participant. Before commencing each task type, participants were briefed on the rules and completed several practice trials for familiarization. The evaluation protocol required participants to solve problems without the use of external aids (e.g., scratch paper, calculators), and they were allowed unlimited time per question. This approach was designed to emphasize and assess their intrinsic spatial visualization and mental manipulation capabilities, creating an evaluation condition comparable to assessing a model's internal reasoning processes without external memory aids. The reported human performance is the mean accuracy across all participants.

### E.6 ERROR ANALYSIS

#### E.6.1 MODEL SELECTION FOR DIRECT ANSWER (NON-COT) EVALUATION

Our Direct Answer evaluation tests model accuracy without induced reasoning chains. We excluded specific models based on 2 criteria:

1. **Reasoning-Centric Architectures:** Models explicitly designed for extended reasoning (e.g., o1, Gemini-2.5, Kimi-thinking, Llama-4 series) were excluded, as inhibiting CoT contradicts their core design principles.

2. **Instruction Adherence:** Models unable to suppress reasoning traces despite strict formatting prompts (specifically InternVL3-2B) were excluded. This failure reflects a limitation in instruction following rather than reasoning capability.

Consequently, we retained only models capable of strictly adhering to the single-letter answer format. This exclusion criteria—based on format compliance rather than performance—ensures the baseline remains representative and uninflated.

#### E.6.2 ERROR TYPES

1. **Perceptual Error:** Failure to perceive fundamental visual properties, such as color, shape, or pattern structures.

2. **Spatial Transformation Error:** Failure to deduce correct spatial states after a transformation. This includes:
   (a) Rotation/Flipping: Errors in angle or axis; confusing rotation with flipping.
   (b) Folding/Unfolding: Incorrect mapping between 2D nets and 3D cubes; confusing adjacent or opposite faces.
   (c) Spatial Relationships: Misjudging object composition, internal structure, or occlusion.

3. **Spatial Memorization Error:** Forgetting or misremembering object positions or relationships across a sequence of operations.

4. **Instruction Following Error:** Misunderstanding textual instructions, such as task rules (e.g., negation) or required output formats.

5. **Methodological Error:** Adopting a flawed or suboptimal problem-solving strategy, such as using a rigid or unnecessarily complex reasoning path.

6. **Calculation and Reasoning Error:** Errors in non-spatial logic or mathematical calculations.

#### E.6.3 INTER-ANNOTATOR AGREEMENT ANALYSIS

To ensure the reliability and reproducibility of our error taxonomy (detailed in Appendix E.6.2), we conducted a rigorous inter-annotator agreement study.

Table 4: **Inter-Annotator Agreement.** Cohen's $\kappa$ calculated via binary decomposition for multi-label error classification.

| Category | Perc. | Trans. | Meth. | Instr. | Memo. | Calc. | Avg. |
|---|---|---|---|---|---|---|---|
| **Cohen's $\kappa$** | 0.90 | 0.81 | 0.75 | 0.96 | 0.89 | 1.00 | **0.88** |

**Methodology** Since our error analysis involves a multi-label classification task (i.e., a single failure case may stem from multiple error sources simultaneously), the traditional global Cohen's $\kappa$ is not directly applicable. Instead, we adopted a standard binary decomposition approach for multi-label agreement. Specifically, we decomposed the multi-label task into 6 independent binary classification tasks, treating each error category as a "Yes/No" decision.

**Calculation** We randomly sampled 100 failure cases from the evaluation set. Two authors independently annotated these cases based on the defined taxonomy. We then calculated Cohen's $\kappa$ separately for each error category. The results, presented in Table 4, demonstrate high reliability. The Methodological category showed substantial agreement ($\kappa = 0.75$), while all other categories

achieved almost perfect agreement ($\kappa > 0.81$), with Calculation & Reasoning reaching perfect consensus ($\kappa = 1.00$). The macroscopic average Cohen's $\kappa$ across all categories is 0.8847, indicating an almost perfect level of inter-annotator consistency.

# F    DETAILED RESULTS

In this section, we provide more evaluation results and test cases from Gemini-2.5-pro for each task.

## F.1    INTRA-CATEGORY COMPARISONS ACROSS LEVELS

To provide deeper insight into the spatial visualization reasoning capabilities of Multi-modal Large Language Models (MLLMs), this section presents comprehensive experimental results that complement the aggregate performance assessment in Section 4.2. This analysis details the accuracy of each evaluated model across the four core sub-abilities—*mental rotation, mental folding, visual penetration, and mental animation*—defined in the SpatialViz-Bench benchmark, with results stratified by task type and difficulty level. This granular performance breakdown reveals specific strengths and weaknesses of the models when confronting various spatial reasoning challenges, offering targeted insights to guide future model improvements.

### F.1.1    MENTAL ROTATION

Table 7 documents model performance on 3 sub-tasks within the mental rotation category—2D Rotation (2DR), 3D Rotation (3DR), and 3-View Projection (3VP)—across different difficulty levels.

In the 2D Rotation (2DR) task, several models demonstrate foundational capabilities at Level 0, with o1 (72.5%) and Gemini-2.5-pro (62.5%) achieving notable results. As difficulty increases to Level 1, most models show performance decline, though leading models maintain relatively high accuracy (o1: 52.5%, Gemini-2.5-pro: 42.5%).

For 3D Rotation (3DR), performance degradation with increased difficulty is more pronounced. At Level 0, o1 (42.5%) and Gemini-2.5-pro (45.0%) perform adequately, but their accuracies decrease substantially to 15.0% and 20.0%, respectively, at Level 1. Many open-source models perform at or below random chance (25%-30%) at this higher difficulty level, highlighting the challenge of mental rotation in complex 3D space.

Interestingly, the 3-View Projection (3VP) task reveals a different pattern: when transitioning from Level 0 (cube stacks) to Level 1 (DeepCAD engineering models), some top-tier models like o1 (improving from 40.0% to 58.0%) and Gemini-2.5-pro (increasing from 28.0% to 66.0%) demonstrate enhanced performance. This suggests certain Level 1 image features may be more amenable to these models' processing mechanisms, despite the presumed increase in complexity. Nevertheless, many other models show decreased performance from Level 0 to Level 1 in this sub-task. Overall, mental rotation tasks reveal a clear performance gradient across dimensions and geometric complexity while highlighting significant capability variations among model families.

### F.1.2    MENTAL FOLDING

Table 8 documents model performance on 3 sub-tasks within the mental folding category—Paper Folding (PF), Cube Unfolding (CU), and Cube Reconstruction (CR)—at varying difficulty levels. These tasks assess models' capacity for continuous reasoning and dynamic visualization of 3D information throughout transformation processes.

In the Paper Folding (PF) task, as folding steps and hole-punching complexity increase (Level 0 to Level 2), most models perform near random chance, indicating significant challenges in tracking multi-step geometric operations and performing subsequent spatial reasoning.

The more complex Cube Unfolding (CU) and Cube Reconstruction (CR) tasks proved challenging for all models. These tasks require understanding the correspondence between 2D nets and 3D cubes, while also assessing the ability to mentally execute folding operations and continuously reason about transforming 3D structures. Even at Level 0, most models demonstrate low accuracy, often below random chance. In the CU task, Gemini-2.5-pro scored 37.5% (L0), 27.5% (L1), and 30.0% (L2), while o1 achieved 37.5% (L0), 37.5% (L1), and 27.5% (L2).

For CR, Gemini-2.5-pro performed at 45.0% (L0), 10.0% (L1), and 35.0% (L2), and o1 at 42.5% (L0), 12.5% (L1), and 25.0% (L2), both experiencing significant performance drops at Level 1. However, the surprising performance improvement at Level 2 contradicts human intuition, as Level 2 patterns are objectively more complex for humans. Analysis of sample solutions reveals that models approached these tasks by employing clear textual descriptions to define patterns composed of differently colored dots, representing their positions in matrix form. Conversely, line patterns proved more challenging for models to describe, and internal rotations could not be easily represented through matrix transposition operations, which . This insight provides valuable direction for designing more challenging tests that effectively evaluate model limitations. The overall results reveal a severe deficiency in reasoning and visualization capabilities when finer-grained correspondence and transformation tracking are required. The introduction of asymmetric patterns further challenges models' ability to maintain precise visual perception and spatial-topological understanding. These results highlight current MLLMs' core weaknesses in handling spatial tasks involving geometric correspondence, topological transformations, and dynamic 3D reasoning.

### F.1.3  VISUAL PENETRATION

Table 9 documents model performance on 3 sub-tasks within the Visual Penetration category—Cross-Section (CS), Cube Counting (CC), and Cube Assembly (CA)—at varying difficulty levels. This ability requires models to infer internal object structures from visible external features.

In the Cross-Section (CS) task, which requires models to visualize sectional shapes produced by cutting composite geometric solids with various planes, Gemini-2.5-pro and o1 maintained relatively stable performance across Levels 0, 1, and 2, while most other models performed near random chance.

For the Cube Counting (CC) task, increasing constraints from two-view (Level 0) to three-view (Level 1), and subsequently expanding spatial dimensions (Level 2), progressively challenged models' view integration and counting inference capabilities. Gemini-2.5-pro's accuracy declined sharply from 80.0% (L0) to 52.5% (L1) and 32.5% (L2). Interestingly, o1's performance followed a pattern of 45.0% (L0), 32.5% (L1), and 45.0% (L2), recovering at Level 2 to match its Level 0 score. Most models struggled to effectively integrate multi-view information in this task.

The Cube Assembly (CA) task, which assesses the ability to identify complementary parts forming a complete structure, showed increasing difficulty as structures enlarged and constituent parts increased (Level 0 to Level 1). For example, Gemini-2.5-pro's accuracy dropped from 45.0% (L0) to 27.5% (L1), and o1's from 35.0% (L0) to 32.5% (L1). Collectively, these results reveal current models' limitations in inferring global internal structures and spatial occupancy from local surface information.

### F.1.4  MENTAL ANIMATION

Table 10 documents model performance on 3 sub-tasks within the Mental Animation category—Arrow Moving (AM), Block Moving (BM), and Mechanical System (MS)—at varying difficulty levels. These tasks assess understanding of dynamic state changes and causal propagation among system components.

In the Arrow Moving (AM) task, which requires understanding ego-centric movement rules and tracking state changes, the transition from simple single-arrow movements (Level 0) to multi-arrow environments involving swaps (Level 1) increasingly challenges models' rule comprehension and state tracking. A notable performance disparity exists between closed-source models (e.g., Gemini-2.5-pro and o1) and open-source counterparts: the former maintain high accuracy across both difficulty levels (almost 100% accuracy by Gemini-2.5-pro), while most open-source models perform significantly worse (near random), particularly in complex multi-arrow Level 1 scenarios. This suggests a capability gap, potentially stemming from differences in architecture or training data, when precise instruction following and multi-step dynamic spatial reasoning are required.

The Block Moving (BM) task combines directional movement with gravity simulation, increasing spatial complexity and operational sequence length, thereby challenging models' intuitive physics and 3D dynamic spatial reasoning. Gemini-2.5-pro's accuracy declined sharply from 95% to 35%, showing the difficulty in dealing with 3D scene.

For the Mechanical System (MS) task, which evaluates understanding of motion transmission and component linkage in complex mechanical systems, questions were designed to minimize reliance on formal physics formulas while emphasizing comprehension through observation and spatial imagination. Interestingly, some open-source models performed better than expected based on their performance in other 3D imagination tasks. This suggests these models may transform such problems into more formalized reasoning processes similar to physical rule application, rather than relying solely on intuitive 3D mental simulation. While this strategy may yield relatively good scores in certain instances, it potentially deviates from the primary goal of assessing pure spatial visualization capabilities. Overall, mental animation tasks—especially those involving complex dynamic interactions and implicit physical laws—continue to pose significant challenges for current MLLMs, with models exhibiting considerable diversity in performance strategies and capabilities.

## F.2 STATISTICAL SIGNIFICANCE ANALYSIS OF CoT PROMPTING

The impact is significantly positive for claude-3.5-sonnet but significantly negative for several leading open-source models.

Table 5: Statistical significance analysis of CoT prompting impact ($p < 0.05$).

| Model | Source | CoT Impact | Significant ($p < 0.05$) | p-value |
|---|---|---|---|---|
| Kimi-VL-A3B-Instruct | Open | Negative | Yes | 0.0192 |
| Deepseek-VL2-tiny | Open | Negative | Yes | 0.0463 |
| Internvl2.5-78B | Open | Negative | Yes | 0.0368 |
| Qwen2.5-Omni-7B | Open | Negative | Yes | 0.0216 |
| Sail-VL-1.6-8B | Open | Negative | Yes | 0.0479 |
| Claude-3.5-sonnet | Closed | Positive | Yes | 0.0007 |

## F.3 PERFORMANCE COMPARISON BETWEEN DIFFERENT QUESTION FORMAT

This benchmark primarily uses MCAs, a deliberate and justified design choice. MCAs are particularly effective for tasks with complex answers (e.g., 3D Rotation Task) that are difficult to express textually or match automatically. Moreover, well-crafted distractors can increase task difficulty and test a model's fine-grained discrimination.

Our rationale for using the MCA format is threefold:

- MCAs align with human qualitative intuition. Humans often rely on estimation rather than precise calculation in spatial reasoning. This format assesses a model's grasp of core transformation logic ("qualitatively correct" reasoning) without penalizing minor deviations.

- Converting some tasks to a direct-answer format is technically challenging. For instance, in 3D Rotation and Paper Folding, the answers are complex images. Requiring models to generate these images is a frontier research problem beyond the scope of current multimodal evaluation.

- We quantitatively measured the difficulty gap. When the Cube Counting task was converted to a fill-in-the-blank format, all models showed a significant performance drop. As shown in Table 6, GPT-4o's accuracy dropped by 32.50%, while even the top-performing Gemini-2.5-pro's declined by 14.17%. This indicates the direct-answer format is more demanding of a model's independent reasoning, even with options like "All three other options are incorrect" to reduce guessing. Consequently, for a comprehensive assessment, we provide both formats for the Cube Counting and parts of the Cube Reconstruction tasks. This performance gap demonstrates that MCA options provide clues or "error-correction" opportunities, helping models select a best-fit answer. In contrast, the direct-answer format more authentically exposes deficits in precise reasoning.

Table 6: Performance Drop on Cube Counting: Multiple-Choice vs. Fill-in-the-Blank. The "Performance Drop" column quantifies the accuracy degradation when switching from the discriminative (Multiple-Choice) to the more challenging generative (Fill-in-the-Blank) task format.

| Model | Multiple-Choice Acc. (%) | | | Fill-in-the-Blank Acc. (%) | | | Avg Performance |
|---|---|---|---|---|---|---|---|
| | L0 | L1 | L2 | L0 | L1 | L2 | Drop (%) |
| *Open Source Models* | | | | | | | |
| Qwen2.5-VL-7B-Instruct | 32.50 | 50.00 | 27.50 | 15.00 | 2.50 | 0.00 | -30.83 |
| Qwen2.5-VL-72B-Instruct | 32.50 | 50.00 | 42.50 | 25.00 | 32.50 | 5.00 | -20.83 |
| *Closed Source Models* | | | | | | | |
| GPT-4o | 40.00 | 45.00 | 37.50 | 10.00 | 12.50 | 2.50 | -32.50 |
| o1 | 45.00 | 32.50 | 45.00 | 20.51 | 22.50 | 10.00 | -23.16 |
| Gemini-2.5-pro | 80.00 | 52.50 | 32.50 | 55.00 | 52.50 | 15.00 | **-14.17** |

Table 7: Comparison of model performances on Mental Rotation tasks. The first and second highest accuracy of MLLMs are marked in red and blue, with open-source and closed-source models marked separately.

| Model | Overall | 2DRotation | | | 3DRotation | | | 3ViewProjection | | |
|---|---|---|---|---|---|---|---|---|---|---|
| | | L0 | L1 | Avg | L0 | L1 | Avg | L0 | L1 | Avg |
| Human | 85.56 | 92.50 | 87.50 | 90.00 | 83.33 | 75.00 | 79.17 | 91.67 | 83.33 | 87.50 |
| Random | 27.69 | 25.00 | 22.50 | 23.75 | 25.00 | 30.00 | 27.50 | 30.00 | 32.00 | 31.00 |
| **Open Source MLLMs** | | | | | | | | | | |
| *3B* | | | | | | | | | | |
| SAIL-VL-1.5-2B | 22.31 | 20.00 | 25.00 | 22.50 | 17.50 | 27.50 | 22.50 | 20.00 | 24.00 | 22.00 |
| InternVL3-2B | 27.31 | 12.50 | 20.00 | 16.25 | 32.50 | 35.00 | 33.75 | 24.00 | 38.00 | 31.00 |
| Deepseek-VL2-tiny(3B) | 22.69 | 10.00 | 25.00 | 17.50 | 20.00 | 25.00 | 22.50 | 22.00 | 32.00 | 27.00 |
| Qwen2.5-VL-3B-Instruct | 20.00 | 25.00 | 15.00 | 20.00 | 15.00 | 22.50 | 18.75 | 16.00 | 26.00 | 21.00 |
| *7B* | | | | | | | | | | |
| Qwen2.5-VL-7B-Instruct | 23.85 | 25.00 | 25.00 | 25.00 | 20.00 | 12.50 | 16.25 | 14.00 | 44.00 | 29.00 |
| Qwen2.5-Omni-7B | 24.23 | 32.50 | 12.50 | 22.50 | 25.00 | 15.00 | 20.00 | 22.00 | 36.00 | 29.00 |
| SAIL-VL-1.6-8B | 21.92 | 25.00 | 12.50 | 18.75 | 27.50 | 15.00 | 21.25 | 24.00 | 26.00 | 25.00 |
| InternVL3-8B | 28.85 | 22.50 | 17.50 | 20.00 | 35.00 | 42.50 | 38.75 | 18.00 | 38.00 | 28.00 |
| *16B* | | | | | | | | | | |
| Kimi-VL-A3B-Instruct(16B) | 28.08 | 15.00 | 17.50 | 16.25 | 32.50 | 27.50 | 30.00 | 24.00 | 48.00 | 36.00 |
| Kimi-VL-A3B-thinking(16B) | 20.00 | 10.00 | 17.50 | 13.75 | 17.50 | 22.50 | 20.00 | 20.00 | 30.00 | 25.00 |
| Deepseek-VL2-small(16B) | 24.62 | 40.00 | 22.50 | 31.25 | 10.00 | 22.50 | 16.25 | 22.00 | 30.00 | 26.00 |
| *32B* | | | | | | | | | | |
| Deepseek-VL2(27B) | 29.62 | 20.00 | 30.00 | 25.00 | 35.00 | 32.50 | 33.75 | 20.00 | 40.00 | 30.00 |
| Qwen2.5-VL-32B-Instruct | 35.00 | 35.00 | 27.50 | 31.25 | 32.50 | 37.50 | 35.00 | 22.00 | 54.00 | 38.00 |
| InternVL3-38B | 28.46 | 25.00 | 20.00 | 22.50 | 32.50 | 35.00 | 33.75 | 22.00 | 36.00 | 29.00 |
| *72B* | | | | | | | | | | |
| Qwen2.5-VL-72B-Instruct | 29.23 | 25.00 | 32.50 | 28.75 | 40.00 | 22.50 | 31.25 | 22.00 | 34.00 | 28.00 |
| QvQ-72B-preview | 27.69 | 15.00 | 27.50 | 21.25 | 27.50 | 32.50 | 30.00 | 32.00 | 30.00 | 31.00 |
| InternVL3-78B | 28.46 | 20.00 | 30.00 | 25.00 | 25.00 | 25.00 | 25.00 | 20.00 | 48.00 | 34.00 |
| *108B* | | | | | | | | | | |
| Llama-4-Maverick-17B-128E-Instruct | 33.85 | 25.00 | 15.00 | 20.00 | 45.00 | 35.00 | 40.00 | 26.00 | 54.00 | 40.00 |
| LLama-4-Scout-17B-16E-Instruct | 37.31 | 32.50 | 32.50 | 32.50 | 32.50 | 37.50 | 35.00 | 28.00 | 58.00 | 43.00 |
| **Closed Source MLLMs** | | | | | | | | | | |
| GPT-4o | 31.15 | 20.00 | 45.00 | 32.50 | 30.00 | 25.00 | 27.50 | 20.00 | 46.00 | 33.00 |
| o1 | 46.92 | 72.50 | 52.50 | 62.50 | 42.50 | 15.00 | 28.75 | 40.00 | 58.00 | 49.00 |
| Claude-3.5-sonnet | 34.62 | 27.50 | 35.00 | 31.25 | 32.50 | 17.50 | 25.00 | 36.00 | 54.00 | 45.00 |
| Claude-3.7-sonnet | 38.08 | 40.00 | 25.00 | 32.50 | 40.00 | 32.50 | 36.25 | 34.00 | 54.00 | 44.00 |
| Gemini-2.5-flash | 35.77 | 55.00 | 30.00 | 42.50 | 40.00 | 20.00 | 30.00 | 18.00 | 52.00 | 35.00 |
| Gemini-2.5-pro | 44.23 | 62.50 | 42.50 | 52.50 | 45.00 | 20.00 | 32.50 | 28.00 | 66.00 | 47.00 |
| Doubao-1-5-vision-pro | 30.38 | 7.50 | 7.50 | 7.50 | 42.50 | 27.50 | 35.00 | 28.00 | 62.00 | 45.00 |
| Qwen-VL-max | 28.08 | 12.50 | 35.00 | 23.75 | 30.00 | 22.50 | 26.25 | 22.00 | 44.00 | 33.00 |

Table 8: Comparison of model performances on Mental Folding tasks.

| Model | Overall | PaperFolding | | | | CubeUnfolding | | | | CubeReconstruction | | | |
|---|---|---|---|---|---|---|---|---|---|---|---|---|---|
| | | L0 | L1 | L2 | Avg | L0 | L1 | L2 | Avg | L0 | L1 | L2 | Avg |
| Human | 80.56 | 100.00 | 93.75 | 87.50 | 93.75 | 87.50 | 75.00 | 62.50 | 75.00 | 81.25 | 75.00 | 62.50 | 72.92 |
| Random | 21.67 | 17.50 | 20.00 | 20.00 | 19.17 | 15.00 | 27.50 | 17.50 | 20.00 | 30.00 | 25.00 | 22.50 | 25.83 |
| *Open Source* | | | | | | | | | | | | | |
| *3B* | | | | | | | | | | | | | |
| SAIL-VL-1.5-2B | 22.50 | 12.50 | 25.00 | 22.50 | 20.00 | 30.00 | 27.50 | 25.00 | 27.50 | 22.50 | 20.00 | 17.50 | 20.00 |
| InternVL3-2B | 24.44 | 25.00 | 27.50 | 15.00 | 22.50 | 35.00 | 12.50 | 30.00 | 25.83 | 35.00 | 22.50 | 17.50 | 25.00 |
| Deepseek-VL2-tiny(3B) | 20.56 | 27.50 | 17.50 | 20.00 | 21.67 | 20.00 | 25.00 | 17.50 | 20.83 | 15.00 | 20.00 | 22.50 | 19.17 |
| Qwen2.5-VL-3B-Instruct | 24.17 | 20.00 | 37.50 | 17.50 | 25.00 | 25.00 | 25.00 | 27.50 | 25.83 | 25.00 | 32.50 | 7.50 | 21.67 |
| *7B* | | | | | | | | | | | | | |
| Qwen2.5-VL-7B-Instruct | 28.61 | 35.00 | 35.00 | 32.50 | 34.17 | 17.50 | 30.00 | 17.50 | 21.67 | 27.50 | 30.00 | 32.50 | 30.00 |
| Qwen2.5-Omni-7B | 24.17 | 27.50 | 30.00 | 17.50 | 25.00 | 32.50 | 37.50 | 12.50 | 27.50 | 17.50 | 27.50 | 15.00 | 20.00 |
| SAIL-VL-1.6-8B | 23.89 | 35.00 | 17.50 | 32.50 | 28.33 | 25.00 | 30.00 | 20.00 | 25.00 | 17.50 | 25.00 | 12.50 | 18.33 |
| InternVL3-8B | 25.56 | 25.00 | 20.00 | 40.00 | 28.33 | 25.00 | 20.00 | 25.00 | 23.33 | 25.00 | 27.50 | 22.50 | 25.00 |
| *16B* | | | | | | | | | | | | | |
| Kimi-VL-A3B-Instruct(16B) | 24.17 | 27.50 | 22.50 | 27.50 | 25.83 | 22.50 | 15.00 | 22.50 | 20.00 | 15.00 | 27.50 | 37.50 | 26.67 |
| Kimi-VL-A3B-thinking(16B) | 24.72 | 10.00 | 25.00 | 35.00 | 23.33 | 20.00 | 20.00 | 32.50 | 24.17 | 35.00 | 17.50 | 27.50 | 26.67 |
| Deepseek-VL2-small(16B) | 24.72 | 25.00 | 22.50 | 20.00 | 22.50 | 27.50 | 25.00 | 22.50 | 25.00 | 22.50 | 25.00 | 32.50 | 26.67 |
| *32B* | | | | | | | | | | | | | |
| Deepseek-VL2(27B) | 26.39 | 22.50 | 35.00 | 37.50 | 31.67 | 32.50 | 15.00 | 27.50 | 25.00 | 17.50 | 30.00 | 20.00 | 22.50 |
| Qwen2.5-VL-32B-Instruct | 24.72 | 15.00 | 37.50 | 12.50 | 21.67 | 17.50 | 35.00 | 22.50 | 25.00 | 30.00 | 10.00 | 42.50 | 27.50 |
| InternVL3-38B | 26.94 | 22.50 | 20.00 | 20.00 | 20.83 | 25.00 | 35.00 | 27.50 | 29.17 | 22.50 | 32.50 | 37.50 | 30.83 |
| *72B* | | | | | | | | | | | | | |
| Qwen2.5-VL-72B-Instruct | 24.17 | 12.50 | 27.50 | 27.50 | 22.50 | 15.00 | 17.50 | 27.50 | 20.00 | 30.00 | 25.00 | 35.00 | 30.00 |
| QvQ-72B-preview | 21.11 | 15.00 | 12.50 | 22.50 | 16.67 | 22.50 | 15.00 | 20.00 | 19.17 | 30.00 | 25.00 | 27.50 | 27.50 |
| InternVL3-78B | 22.22 | 15.00 | 30.00 | 12.50 | 19.17 | 35.00 | 22.50 | 17.50 | 25.00 | 30.00 | 20.00 | 17.50 | 22.50 |
| *108B* | | | | | | | | | | | | | |
| Llama-4-Maverick-17B-128E-Instruct | 25.00 | 15.00 | 17.50 | 17.50 | 16.67 | 30.00 | 25.00 | 32.50 | 29.17 | 30.00 | 32.50 | 25.00 | 29.17 |
| LLama-4-Scout-17B-16E-Instruct | 28.61 | 15.00 | 17.50 | 17.50 | 16.67 | 35.00 | 32.50 | 30.00 | 32.50 | 42.50 | 32.50 | 35.00 | 36.67 |
| *Closed Source* | | | | | | | | | | | | | |
| GPT-4o | 25.00 | 25.00 | 35.00 | 27.50 | 29.17 | 25.00 | 12.50 | 10.00 | 15.83 | 30.00 | 17.50 | 42.50 | 30.00 |
| o1 | 29.72 | 27.50 | 30.00 | 27.50 | 28.33 | 37.50 | 37.50 | 27.50 | 34.17 | 42.50 | 12.50 | 25.00 | 26.67 |
| Claude-3.5-sonnet | 25.00 | 7.50 | 35.00 | 20.00 | 20.83 | 25.00 | 17.50 | 25.00 | 22.50 | 32.50 | 20.00 | 42.50 | 31.67 |
| Claude-3.7-sonnet | 24.72 | 20.00 | 20.00 | 15.00 | 18.33 | 32.50 | 25.00 | 22.50 | 26.67 | 32.50 | 17.50 | 37.50 | 29.17 |
| Gemini-2.5-flash | 32.50 | 15.00 | 37.50 | 27.50 | 26.67 | 32.50 | 30.00 | 27.50 | 30.00 | 55.00 | 27.50 | 40.00 | 40.83 |
| Gemini-2.5-pro | 35.00 | 57.50 | 40.00 | 32.50 | 43.33 | 37.50 | 27.50 | 30.00 | 31.67 | 45.00 | 10.00 | 35.00 | 30.00 |
| Doubao-1-5-vision-pro | 28.06 | 25.00 | 37.50 | 32.50 | 31.67 | 22.50 | 22.50 | 25.00 | 23.33 | 45.00 | 17.50 | 25.00 | 29.17 |
| Qwen-VL-max | 24.44 | 27.50 | 25.00 | 20.00 | 24.17 | 12.50 | 15.00 | 25.00 | 17.50 | 42.50 | 22.50 | 30.00 | 31.67 |

Table 9: Comparison of model performances on Visual Penetration tasks.

| Model | Overall | CrossSection | | | | CubeCounting | | | | CubeAssembly | | |
|---|---|---|---|---|---|---|---|---|---|---|---|---|
| | | L0 | L1 | L2 | Avg | L0 | L1 | L2 | Avg | L0 | L1 | Avg |
| Human | 75.42 | 75.00 | 75.00 | 68.75 | 72.92 | 81.25 | 75.00 | 56.25 | 70.83 | 87.50 | 75.00 | 82.50 |
| Random | 28.12 | 32.50 | 27.50 | 30.00 | 30.00 | 30.00 | 20.00 | 25.00 | 25.00 | 22.50 | 37.50 | 30.00 |
| **Open Source** | | | | | | | | | | | | |
| *3B* | | | | | | | | | | | | |
| SAIL-VL-1.5-2B | 27.19 | 37.50 | 20.00 | 15.00 | 24.17 | 40.00 | 20.00 | 20.00 | 26.67 | 32.50 | 32.50 | 32.50 |
| InternVL3-2B | 26.56 | 22.50 | 22.50 | 15.00 | 20.00 | 22.50 | 32.50 | 37.50 | 30.83 | 27.50 | 32.50 | 30.00 |
| Deepseek-VL2-tiny(3B) | 20.94 | 17.50 | 25.00 | 20.00 | 20.83 | 25.00 | 25.00 | 17.50 | 22.50 | 17.50 | 20.00 | 18.75 |
| Qwen2.5-VL-3B-Instruct | 25.94 | 25.00 | 25.00 | 27.50 | 25.83 | 17.50 | 35.00 | 17.50 | 23.33 | 30.00 | 30.00 | 30.00 |
| *7B* | | | | | | | | | | | | |
| Qwen2.5-VL-7B-Instruct | 27.19 | 12.50 | 12.50 | 25.00 | 16.67 | 32.50 | 50.00 | 27.50 | 36.67 | 35.00 | 22.50 | 28.75 |
| Qwen2.5-Omni-7B | 27.19 | 15.00 | 22.50 | 25.00 | 20.83 | 37.50 | 27.50 | 35.00 | 33.33 | 25.00 | 30.00 | 27.50 |
| SAIL-VL-1.6-8B | 21.25 | 17.50 | 22.50 | 25.00 | 21.67 | 22.50 | 17.50 | 17.50 | 19.17 | 30.00 | 17.50 | 23.75 |
| InternVL3-8B | 30.94 | 17.50 | 15.00 | 15.00 | 15.83 | 25.00 | 45.00 | 52.50 | 40.83 | 45.00 | 32.50 | 38.75 |
| *16B* | | | | | | | | | | | | |
| Kimi-VL-A3B-Instruct(16B) | 17.19 | 17.50 | 25.00 | 22.50 | 21.67 | 7.50 | 2.50 | 5.00 | 5.00 | 27.50 | 30.00 | 28.75 |
| Kimi-VL-A3B-thinking(16B) | 29.38 | 27.50 | 17.50 | 30.00 | 25.00 | 45.00 | 40.00 | 25.00 | 36.67 | 20.00 | 30.00 | 25.00 |
| Deepseek-VL2-small(16B) | 25.31 | 7.50 | 12.50 | 7.50 | 9.17 | 30.00 | 32.50 | 42.50 | 35.00 | 30.00 | 40.00 | 35.00 |
| *32B* | | | | | | | | | | | | |
| Kimi-VL-A3B-Instruct(16B) | 17.19 | 17.50 | 25.00 | 22.50 | 21.67 | 7.50 | 2.50 | 5.00 | 5.00 | 27.50 | 30.00 | 28.75 |
| Kimi-VL-A3B-thinking(16B) | 29.38 | 27.50 | 17.50 | 30.00 | 25.00 | 45.00 | 40.00 | 25.00 | 36.67 | 20.00 | 30.00 | 25.00 |
| Deepseek-VL2-small(16B) | 25.31 | 7.50 | 12.50 | 7.50 | 9.17 | 30.00 | 32.50 | 42.50 | 35.00 | 30.00 | 40.00 | 35.00 |
| *72B* | | | | | | | | | | | | |
| Qwen2.5-VL-72B-Instruct | 39.06 | 27.50 | 40.00 | 22.50 | 30.00 | 32.50 | 50.00 | 42.50 | 41.67 | 55.00 | 42.50 | 48.75 |
| QvQ-72B-preview | 27.81 | 32.50 | 30.00 | 27.50 | 30.00 | 35.00 | 25.00 | 7.50 | 22.50 | 40.00 | 25.00 | 32.50 |
| InternVL3-78B | 35.00 | 17.50 | 25.00 | 20.00 | 20.83 | 37.50 | 52.50 | 30.00 | 40.00 | 42.50 | 55.00 | 48.75 |
| *108B* | | | | | | | | | | | | |
| Llama-4-Maverick-17B-128E-Instruct | 32.19 | 27.50 | 15.00 | 15.00 | 19.17 | 27.50 | 47.50 | 30.00 | 35.00 | 52.50 | 42.50 | 47.50 |
| LLama-4-Scout-17B-16E-Instruct | 34.06 | 17.50 | 17.50 | 17.50 | 17.50 | 35.00 | 47.50 | 30.00 | 37.50 | 50.00 | 57.50 | 53.75 |
| **Closed Source** | | | | | | | | | | | | |
| GPT-4o | 32.50 | 25.00 | 25.00 | 7.50 | 19.17 | 40.00 | 45.00 | 37.50 | 40.83 | 52.50 | 27.50 | 40.00 |
| o1 | 37.81 | 40.00 | 42.50 | 30.00 | 37.50 | 45.00 | 32.50 | 45.00 | 40.83 | 35.00 | 32.50 | 33.75 |
| Claude-3.5-sonnet | 33.44 | 35.00 | 20.00 | 12.50 | 22.50 | 35.00 | 45.00 | 27.50 | 35.83 | 47.50 | 45.00 | 46.25 |
| Claude-3.7-sonnet | 31.56 | 20.00 | 35.00 | 17.50 | 24.17 | 30.00 | 32.50 | 30.00 | 30.83 | 40.00 | 47.50 | 43.75 |
| Gemini-2.5-flash | 32.81 | 32.50 | 35.00 | 22.50 | 30.00 | 52.50 | 32.50 | 30.00 | 38.33 | 30.00 | 27.50 | 28.75 |
| Gemini-2.5-pro | 42.19 | 32.50 | 35.00 | 32.50 | 33.33 | 80.00 | 52.50 | 32.50 | 55.00 | 45.00 | 27.50 | 36.25 |
| Doubao-1-5-vision-pro | 39.69 | 35.00 | 30.00 | 25.00 | 30.00 | 62.50 | 65.00 | 40.00 | 55.83 | 42.50 | 17.50 | 30.00 |
| Qwen-VL-max | 38.44 | 32.50 | 20.00 | 27.50 | 26.67 | 57.50 | 62.50 | 22.50 | 47.50 | 50.00 | 35.00 | 42.50 |

Table 10: Comparison of model performances on Mental Animation tasks.

| Model | Overall | ArrowMoving | | | BlockMoving | | | MechanicalSystem | | |
|---|---|---|---|---|---|---|---|---|---|---|
| | | L0 | L1 | Avg | L0 | L1 | Avg | L0 | L1 | Avg |
| Human | 88.33 | 92.50 | 87.5 | 90.00 | 95.83 | 79.16 | 87.5 | 87.50 | 87.50 | 87.50 |
| Random | 23.33 | 32.50 | 25.00 | 28.75 | 10.00 | 22.50 | 16.25 | 30.00 | 20.00 | 25.00 |
| **Open Source** | | | | | | | | | | |
| **3B** | | | | | | | | | | |
| SAIL-VL-1.5-2B | 24.58 | 15.00 | 27.50 | 21.25 | 22.50 | 27.50 | 25.00 | 35.00 | 20.00 | 27.50 |
| InternVL3-2B | 27.08 | 22.50 | 15.00 | 18.75 | 37.50 | 27.50 | 32.50 | 25.00 | 35.00 | 30.00 |
| Deepseek-VL2-tiny(3B) | 21.67 | 25.00 | 12.50 | 18.75 | 25.00 | 17.50 | 21.25 | 25.00 | 25.00 | 25.00 |
| Qwen2.5-VL-3B-Instruct | 35.83 | 35.00 | 35.00 | 35.00 | 32.50 | 27.50 | 30.00 | 57.50 | 27.50 | 42.50 |
| **7B** | | | | | | | | | | |
| Qwen2.5-VL-7B-Instruct | 32.50 | 22.50 | 22.50 | 22.50 | 22.50 | 25.00 | 23.75 | 67.50 | 35.00 | 51.25 |
| Qwen2.5-Omni-7B | 35.42 | 27.50 | 35.00 | 31.25 | 32.50 | 27.50 | 30.00 | 67.50 | 22.50 | 45.00 |
| SAIL-VL-1.6-8B | 35.00 | 12.50 | 37.50 | 25.00 | 37.50 | 32.50 | 35.00 | 52.50 | 37.50 | 45.00 |
| InternVL3-8B | 37.08 | 30.00 | 30.00 | 30.00 | 30.00 | 30.00 | 30.00 | 62.50 | 40.00 | 51.25 |
| **16B** | | | | | | | | | | |
| Kimi-VL-A3B-Instruct(16B) | 27.92 | 17.50 | 12.50 | 15.00 | 27.50 | 35.00 | 31.25 | 57.50 | 17.50 | 37.50 |
| Kimi-VL-A3B-thinking(16B) | 40.42 | 22.50 | 37.50 | 30.00 | 35.00 | 52.50 | 43.75 | 62.50 | 32.50 | 47.50 |
| Deepseek-VL2-small(16B) | 26.25 | 25.00 | 27.50 | 26.25 | 25.00 | 22.50 | 23.75 | 47.50 | 10.00 | 28.75 |
| **32B** | | | | | | | | | | |
| Deepseek-VL2(27B) | 29.17 | 20.00 | 32.50 | 26.25 | 35.00 | 25.00 | 30.00 | 40.00 | 22.50 | 31.25 |
| Qwen2.5-VL-32B-Instruct | 37.08 | 22.50 | 35.00 | 28.75 | 27.50 | 27.50 | 27.50 | 62.50 | 47.50 | 55.00 |
| InternVL3-38B | 37.08 | 25.00 | 25.00 | 25.00 | 25.00 | 35.00 | 30.00 | 65.00 | 47.50 | 56.25 |
| **72B** | | | | | | | | | | |
| Qwen2.5-VL-72B-Instruct | 43.75 | 27.50 | 27.50 | 27.50 | 45.00 | 35.00 | 40.00 | 67.50 | 60.00 | 63.75 |
| QvQ-72B-preview | 39.58 | 27.50 | 22.50 | 25.00 | 40.00 | 60.00 | 50.00 | 42.50 | 45.00 | 43.75 |
| InternVL3-78B | 35.42 | 25.00 | 22.50 | 23.75 | 35.00 | 47.50 | 41.25 | 55.00 | 27.50 | 41.25 |
| **108B** | | | | | | | | | | |
| Llama-4-Maverick-17B-128E-Instruct | 39.17 | 35.00 | 35.00 | 35.00 | 40.00 | 40.00 | 40.00 | 45.00 | 40.00 | 42.50 |
| LLama-4-Scout-17B-16E-Instruct | 39.58 | 15.00 | 42.50 | 28.75 | 47.50 | 32.50 | 40.00 | 57.50 | 42.50 | 50.00 |
| **Closed Source** | | | | | | | | | | |
| GPT-4o | 38.33 | 32.50 | 12.50 | 22.50 | 25.00 | 40.00 | 32.50 | 62.50 | 57.50 | 60.00 |
| o1 | 57.50 | 75.00 | 60.00 | 67.50 | 50.00 | 55.00 | 52.50 | 62.50 | 42.50 | 52.50 |
| Claude-3.5-sonnet | 40.42 | 42.50 | 32.50 | 37.50 | 25.00 | 37.50 | 31.25 | 57.50 | 47.50 | 52.50 |
| Claude-3.7-sonnet | 46.25 | 72.50 | 60.00 | 66.25 | 25.00 | 32.50 | 28.75 | 55.00 | 32.50 | 43.75 |
| Gemini-2.5-flash | 50.00 | 82.50 | 52.50 | 67.50 | 32.50 | 35.00 | 33.75 | 62.50 | 35.00 | 48.75 |
| Gemini-2.5-pro | 62.92 | 97.50 | 92.50 | 95.00 | 27.50 | 42.50 | 35.00 | 75.00 | 42.50 | 58.75 |
| Doubao-1-5-vision-pro | 35.83 | 22.50 | 22.50 | 22.50 | 32.50 | 42.50 | 37.50 | 57.50 | 37.50 | 47.50 |
| Qwen-VL-max | 39.17 | 25.00 | 27.50 | 26.25 | 35.00 | 37.50 | 36.25 | 65.00 | 45.00 | 55.00 |

## F.4 TEST CASES

This section presents test cases from Gemini-2.5-pro across various tasks, helping us identify error sources and provide direction for model improvement.

---

**Mental Rotation-2D Rotation-Level 1**

**Prompt：** You should first provide a reasoning process, then provide a single option(A, B, C or D) as the final answer. The reasoning process and the answer are enclosed within <think></think> and <answer></answer> tags, respectively, i.e., <think>reasoning process</think>, <answer>answer</answer>.

**Question:** The left image shows a patterned grid with a red square marking one corner. Which grid can be obtained by rotating the left grid only? Please answer from options A, B, C, or D.

**Choices:**
    A. A   B. B
    C. C   D. All three other options are incorrect

**Ground Truth Answer:** B

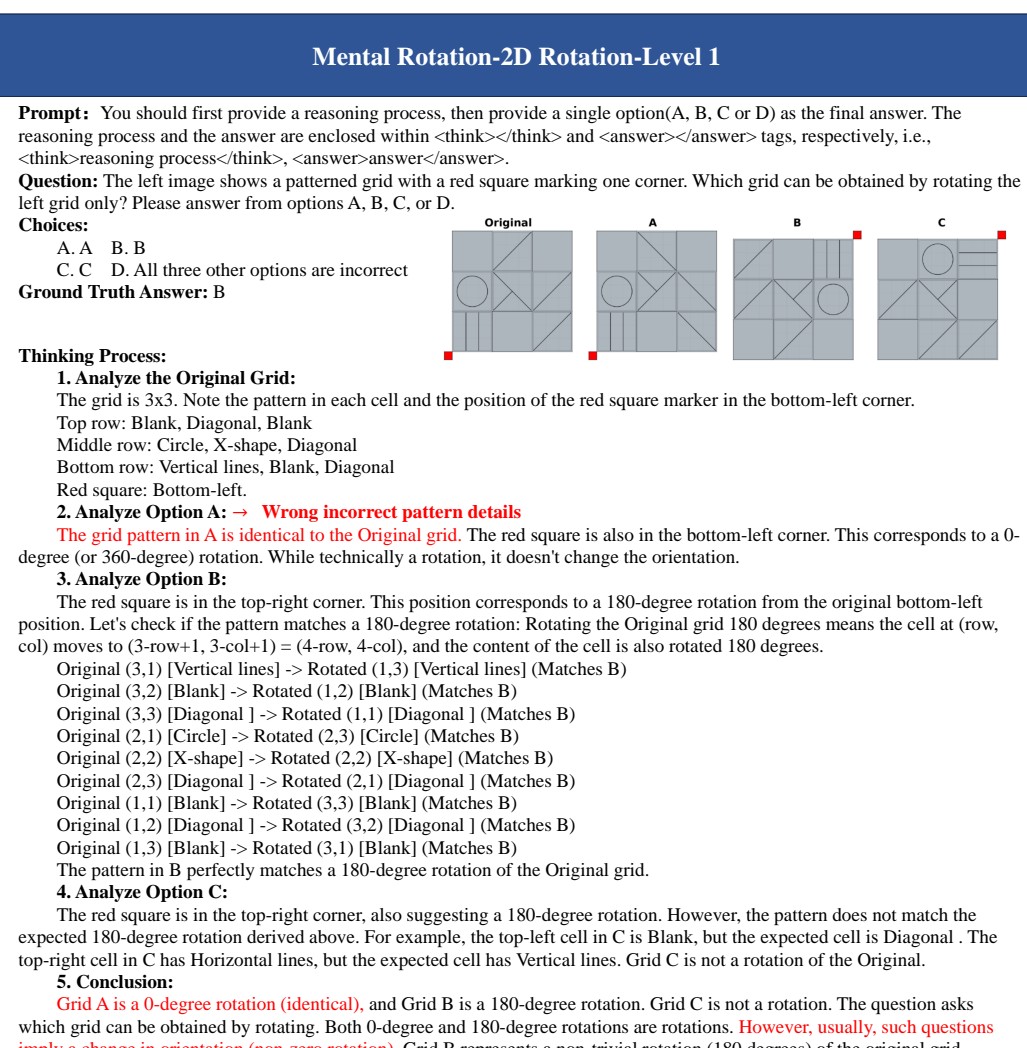

**Thinking Process:**
    **1. Analyze the Original Grid:**
The grid is 3x3. Note the pattern in each cell and the position of the red square marker in the bottom-left corner.
Top row: Blank, Diagonal, Blank
Middle row: Circle, X-shape, Diagonal
Bottom row: Vertical lines, Blank, Diagonal
Red square: Bottom-left.
    **2. Analyze Option A: → Wrong incorrect pattern details**
The grid pattern in A is identical to the Original grid. The red square is also in the bottom-left corner. This corresponds to a 0-degree (or 360-degree) rotation. While technically a rotation, it doesn't change the orientation.
    **3. Analyze Option B:**
The red square is in the top-right corner. This position corresponds to a 180-degree rotation from the original bottom-left position. Let's check if the pattern matches a 180-degree rotation: Rotating the Original grid 180 degrees means the cell at (row, col) moves to (3-row+1, 3-col+1) = (4-row, 4-col), and the content of the cell is also rotated 180 degrees.
Original (3,1) [Vertical lines] -> Rotated (1,3) [Vertical lines] (Matches B)
Original (3,2) [Blank] -> Rotated (1,2) [Blank] (Matches B)
Original (3,3) [Diagonal ] -> Rotated (1,1) [Diagonal ] (Matches B)
Original (2,1) [Circle] -> Rotated (2,3) [Circle] (Matches B)
Original (2,2) [X-shape] -> Rotated (2,2) [X-shape] (Matches B)
Original (2,3) [Diagonal ] -> Rotated (2,1) [Diagonal ] (Matches B)
Original (1,1) [Blank] -> Rotated (3,3) [Blank] (Matches B)
Original (1,2) [Diagonal ] -> Rotated (3,2) [Diagonal ] (Matches B)
Original (1,3) [Blank] -> Rotated (3,1) [Blank] (Matches B)
The pattern in B perfectly matches a 180-degree rotation of the Original grid.
    **4. Analyze Option C:**
The red square is in the top-right corner, also suggesting a 180-degree rotation. However, the pattern does not match the expected 180-degree rotation derived above. For example, the top-left cell in C is Blank, but the expected cell is Diagonal . The top-right cell in C has Horizontal lines, but the expected cell has Vertical lines. Grid C is not a rotation of the Original.
    **5. Conclusion:**
Grid A is a 0-degree rotation (identical), and Grid B is a 180-degree rotation. Grid C is not a rotation. The question asks which grid can be obtained by rotating. Both 0-degree and 180-degree rotations are rotations. However, usually, such questions imply a change in orientation (non-zero rotation). Grid B represents a non-trivial rotation (180 degrees) of the original grid.
**Final Answer: B**

Figure 20: Test case of 2D Rotation Task from Gemini-2.5-pro.

## Mental Rotation-3D Rotation-Level 0

**Prompt：** You should first provide a reasoning process, then provide a single option(A, B, C or D) as the final answer. The reasoning process and the answer are enclosed within <think></think> and <answer></answer> tags, respectively, i.e., <think>reasoning process</think>, <answer>answer</answer>.

**Question:** The left image shows the original cube stack made of equal-sized small cubes. Which of the options on the right **cannot** be obtained by rotating the original cube stack? Please answer from options A, B, C, or D.

**Choices:**

A. A   B. B

C. C   D. All three other options are incorrect

**Ground Truth Answer:** A

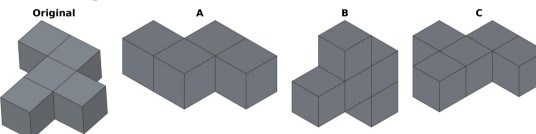

**Thinking Process: → Confusing description**

**1. Analyze the Original Shape:**

The original shape consists of 5 equal-sized cubes. Visual inspection suggests a central cube connected to four other cubes. Given the 3D isometric view, it's reasonable to assume the central cube (which might be hidden) has cubes attached to four of its faces. Let's assume the structure is a central cube C with neighbours North (N), South (S), East (E), and Top (T). This structure, {C, N, S, E, T}, is inherently 3-dimensional (non-planar). The central cube C has 4 connections (neighbours), and the other 4 cubes (N, S, E, T) each have only 1 connection (to C).

**2. Analyze Shape A:**

Shape A consists of 5 cubes. It appears non-planar. Let's determine its connectivity (number of neighbours for each cube). By visual inspection, it seems to have a 'backbone' of 3 cubes and two 'legs'. The central cube of the backbone appears connected to 3 other cubes. No cube appears connected to 4 others. The connectivity pattern seems to be (1 cube with 3 neighbours, 1 cube with 2 neighbours, 3 cubes with 1 neighbour). This differs from the Original's pattern (1 cube with 4 neighbours, 4 cubes with 1 neighbour). Therefore, A cannot be obtained by rotating the Original.

**3. Analyze Shape B:**

Shape B consists of 5 cubes. It appears non-planar. Let's determine its connectivity. It looks like an L-shape base of 3 cubes, with two cubes stacked vertically on two of the base cubes (one on the corner, one on an end). By visual inspection, the maximum number of neighbours any cube has is 2 (the corner cube of the L and the two cubes adjacent to it in the L). The connectivity pattern seems to be (3 cubes with 2 neighbours, 2 cubes with 1 neighbour). This differs from the Original's pattern (1x4, 4x1). Therefore, B cannot be obtained by rotating the Original.

**4. Analyze Shape C:**

Shape C consists of 5 cubes. This shape appears to be planar, meaning all cubes lie in a single layer. It forms a shape known as the 'P' pentomino (a 2x2 square with one cube attached to the side). A planar shape consists of cubes whose centers all lie on the same plane.

**5. Compare Shapes with Original:**

The Original shape {C, N, S, E, T} is non-planar because the Top cube (T) is in a different layer than the N, S, E cubes.

Shape C is planar. Rotation is a rigid transformation. It cannot change a non-planar object into a planar object.

Since the Original shape is non-planar and Shape C is planar, Shape C cannot be obtained by rotating the Original shape.

While analysis also suggested A and B cannot be obtained by rotating the Original (due to different connectivity patterns), the difference in planarity between the Original and C is a more fundamental geometric property preserved under rotation. Given that usually only one option is correct in such problems, the planarity difference provides the strongest reason.

**6. Conclusion:**

Shape C cannot be obtained by rotating the original cube stack because the original stack is non-planar and shape C is planar.

**Final Answer: C**

Figure 21: Test case of 3D Rotation Task from Gemini-2.5-pro.

## Mental Rotation-3View Projection-Level 0

**Prompt：** You should first provide a reasoning process, then provide a single option(A, B, C or D) as the final answer. The reasoning process and the answer are enclosed within <think></think> and <answer></answer> tags, respectively, i.e., <think>reasoning process</think>, <answer>answer</answer>.

**Question:** The cube stack is made of equal-sized small cubes, mostly gray with a few red ones. The top row shows its isometric view, front view, and top view from left to right. Which image in the bottom row is the left view of the cube stack? Please answer from options A, B, C, or D.

**Choices:**

A. A  B. B  C. C  D. D

**Ground Truth Answer:** A

**Thinking Process:**

**1. Analyze the Isometric View:**

The object is composed of stacks of cubes on a 3x3 base grid (viewed from top-down, let's label rows as Front, Middle, Back and columns as Left, Middle, Right).

Front-Left stack: 3 cubes high (Bottom Red, Middle Grey, Top Red).

Front-Middle stack: 3 cubes high (All Grey).

Front-Right stack: 1 cube high (Grey).

Middle-Left stack: 3 cubes high (All Grey).

Middle-Middle stack: 4 cubes high (3 Grey base + 1 Red on top).

Middle-Right stack: 1 cube high (Grey).

Back-Left stack: 3 cubes high (All Grey, implied by structure).

Back-Middle stack: 3 cubes high (All Grey, implied base for MM red cube).

Back-Right stack: 1 cube high (All Grey, implied by structure).

**2. Determine the Left View:**

The left view shows the object as seen from the left side. The 2D view grid will have columns representing the depth (Back, Middle, Front) and rows representing the height (Bottom, Middle, Top). We need to determine the color of the cube face visible at each position in this grid. Since the options are 3x3, we consider heights up to 3.

**3. Looking from the left:**

We see the leftmost faces of the stacks in each depth layer (Front, Middle, Back).

Front layer (Closest depth to the right in the left view): The leftmost stack is the Front-Left stack. It is 3 high (Bottom Red, Middle Grey, Top Red). Its left face is visible. So, the rightmost column of the left view is R (Top), G (Middle), R (Bottom).

Middle layer (Middle depth): The leftmost stack is the Middle-Left stack. It is 3 high (All Grey). Its left face is visible. So, the middle column of the left view is G (Top), G (Middle), G (Bottom).

Back layer (Furthest depth to the left in the left view): The leftmost stack is the Back-Left stack. It is 3 high (All Grey). Its left face is visible. So, the leftmost column of the left view is G (Top), G (Middle), G (Bottom).

**4. Construct the 3x3 Left View Grid:**

Row 3 (Top):   Grey (Back), Grey (Middle), Red (Front)

Row 2 (Middle): Grey (Back), Grey (Middle), Grey (Front)

Row 1 (Bottom): Grey (Back), Grey (Middle), Red (Front)

**5. Compare with Options:**

Option A: Top Row (G, G, R), Middle Row (G, G, G), Bottom Row (G, G, R). This matches the derived left view.

Option B: Top Row (R, G, G), Middle Row (G, G, G), Bottom Row (R, G, G). Mismatch.

Option C: Top Row (G, G, R), Middle Row (G, G, G), Bottom Row (G, G, G). Mismatch.

Option D: Top Row (R, G, G), Middle Row (G, G, G), Bottom Row (G, G, G). Mismatch.

Note on discrepancies: The provided Front and Top views in the question appear inconsistent with the Isometric view (e.g., the height of the Front-Right stack, the position of the topmost Red cube, the empty space in the Top view). However, the question asks for the left view of the cube stack shown in the isometric view. Therefore, we derive the left view from the isometric drawing. The derivation based on the isometric view matches Option A.

**Final Answer: A**

Figure 22: Test case of Three-View Projection (Level 0) Task from Gemini-2.5-pro.

## Mental Rotation-3View Projection-Level 1

**Prompt：** You should first provide a reasoning process, then provide a single option(A, B, C or D) as the final answer. The reasoning process and the answer are enclosed within <think></think> and <answer></answer> tags, respectively, i.e., <think>reasoning process</think>, <answer>answer</answer>.

**Question:** The top row shows the isometric view (left) and the top view (right) of a 3D model. Which image in the bottom row is the front view of the model? Please answer from options A, B, C, or D.

**Choices:**

A. A   B. B   C. C   D. All three other options are incorrect

**Ground Truth Answer:** A

**Thinking Process:**

1. **Understand the Task:**
The goal is to identify the correct front view of the 3D object, given its isometric view and top view.

2. **Analyze the Isometric View:**
The isometric view shows a 3D representation of the object. It consists of a main rectangular prism (or block) and a cylinder attached to one of the larger faces of the prism. The cylinder extends outwards from this face.

3. **Analyze the Top View:**
The top view shows the object as seen from directly above. It appears as a 'T' shape. The wider horizontal part of the 'T' corresponds to the top surface of the rectangular prism. The narrower vertical part of the 'T' corresponds to the top view of the cylinder. This confirms that the cylinder is centered horizontally on the face it's attached to. It also shows the relative width of the prism and the diameter of the cylinder, as well as the length the cylinder protrudes.

4. **Determine the Orientation for the Front View:**
The isometric view typically shows the front, top, and one side (usually right). The top view is given. The front view would be looking at the face from which the cylinder protrudes.

5. **Visualize the Front View:**
Imagine looking directly at the face of the rectangular prism where the cylinder is attached. You would see the rectangular outline of that face. You would also see the end of the cylinder, which is a circle, superimposed onto that rectangular face.
From the isometric view, the cylinder appears centered vertically on the face.
From the top view, the cylinder is centered horizontally on the face.
Therefore, the front view should be a rectangle with a circle located in its center.

6. **Compare with the Options:**
Option A: Shows a rectangle with a circle inside, centered horizontally and vertically. This matches the visualization of the front view derived from the isometric and top views.
Option B: Shows a 'T' shape lying on its side. This does not match the front view. It resembles the top view rotated.
Option C: Shows only a rectangle. This would be the back view of the object, or the front view if the cylinder were absent.

7. **Conclusion:**
Option A is the correct representation of the front view of the given 3D object.
**Final Answer: A**

Figure 23: Test case of Three-View Projection (Level 1) Task from Gemini-2.5-pro.

## Mental Folding-Paper Folding-Level 0

**Prompt**：You should first provide a reasoning process, then provide a single option(A, B, C or D) as the final answer. The reasoning process and the answer are enclosed within <think></think> and <answer></answer> tags, respectively, i.e., <think>reasoning process</think>, <answer>answer</answer>.

**Question:** The original paper is a 3×3 grid paper. The images in the top row show the results of 2 consecutive folding operations on the grid paper. Folding operations include folding along horizontal, vertical, or 45-degree direction. The rightmost image in the top row shows the result after punching holes in the folded paper. Which image represents the appearance of the paper after unfolding the punched paper? Black solid circles represent grid cells with holes. Please answer from options A, B, C, or D.

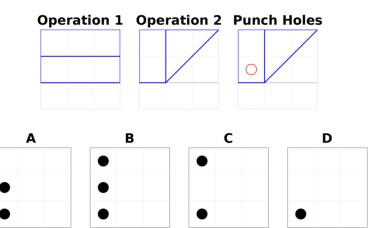

**Choices:**
 A. A   B. B   C. C   D. D

**Ground Truth Answer:** A

**Thinking Process:**
  **Step 1: Analyze the starting paper.**
  The paper is a 3x3 grid. Let's label the cells by (row, column) starting from the top-left as (1,1).
  (1,1) (1,2) (1,3); (2,1) (2,2) (2,3); (3,1) (3,2) (3,3)
  **Step 2: Analyze Operation 1.**
  A horizontal fold is made across the middle of the original 3x3 grid. The diagram shows the bottom third (row 3) being folded upwards, lying underneath the middle third (row 2). The fold line is between row 2 and row 3.
  The state after Operation 1 is a 3x2 grid shape (rows 1 and 2 are visible).
  - Row 1 contains cells (1,1), (1,2), (1,3) (single layer).
  - Row 2 (visible) contains cells (2,1), (2,2), (2,3) as the top layer, and cells (3,1), (3,2), (3,3) as the bottom layer, folded underneath.
  **Step 3: Analyze Operation 2.**
  Starting from the 3x2 shape after Operation 1. A diagonal fold is made within the right 2x2 section (columns 2 and 3). The fold line goes from the bottom-left corner of cell (2,2) to the top-right corner of cell (1,3). The top-right triangular part (involving cells (1,2), (1,3), (2,3)) is folded down/left onto the bottom-left triangular part (involving cells (1,2), (2,2), (2,3)).
  The state after Operation 2 is shown. Note the layers:
  - The leftmost column (original cells (1,1) and (2,1)/(3,1)) is unaffected by this fold.
  Cell (1,1) is single layer. Cell (2,1) is the top layer, (3,1) is the bottom layer.
  - The right part is now folded.
  **Step 4: Analyze Punch Holes.**
  A single hole is punched. Its location is in the bottom-left square of the shape shown after Operation 2. This corresponds to the cell location (2,1) in the configuration after Operation 1. Since this location (2,1) had two layers of paper from Operation 1 (original cell (2,1) on top and original cell (3,1) underneath), punching a hole here creates holes through both layers. This area was not affected by the fold in Operation 2.
  **Step 5: Unfold Operation 2.**
  We reverse Operation 2 by unfolding the diagonal fold. Since the punched hole was in the part not involved in this fold ((2,1) area), unfolding Operation 2 does not change the location of the holes or create new ones. The paper returns to the state after Operation 1 (the 3x2 shape). We have holes through both layers at position (2,1). That means original cell (2,1) has a hole, and original cell (3,1) has a hole.
  **Step 6: Unfold Operation 1.**
  We reverse Operation 1 by unfolding the bottom third (row 3) back down. The fold line was between row 2 and row 3. The holes are currently in cell (2,1) and cell (3,1). Unfolding separates these layers. The final unfolded 3x3 paper will have a hole in cell (2,1) (middle row, left column) and a hole in cell (3,1) (bottom row, left column).
  **Step 7: Compare with options.**
  - Option A shows holes at (2,1) and (3,1). This matches our result.
  - Option B shows holes at (1,1), (2,1), (3,1). Incorrect.
  - Option C shows holes at (1,1) and (3,1). Incorrect.
  - Option D shows a hole at (3,1). Incorrect.
  Therefore, the correct unfolded pattern is shown in Option A.
  **Final Answer: A**

Figure 24: Test case of Paper Folding Task from Gemini-2.5-pro.

## Mental Folding-Cube Unfolding-Level 0

**Prompt：** You should first provide a reasoning process, then provide a single option(A, B, C or D) as the final answer. The reasoning process and the answer are enclosed within <think></think> and <answer></answer> tags, respectively, i.e., <think>reasoning process</think>, <answer>answer</answer>.

**Question:** The left image shows a colored cube from a particular viewing angle. The options are nets (unfolded patterns) of the cube, which are folded upward to form the cube. Which net, when folded, **cannot** form the cube shown in the left image? Please answer from options A, B, C, or D.

**Choices:**

A. A  B. B  C. C  D. D

**Ground Truth Answer:** C

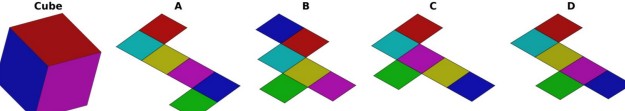

**Thinking Process:**

**1. Analyze the Target Cube:**

The image shows a cube with three visible faces meeting at a vertex: Red (Top), Blue (Front), and Green (Right). This means Red, Blue, and Green must be adjacent faces in any valid net representation. → **Wrong color detection**

**2. Analyze the Nets (A, B, C, D):**

Each net represents an unfolded cube. To determine if a net can form the target cube, we need to check if the required faces (Red, Blue, Green) are adjacent and if their spatial relationship matches the target cube when folded. A key property of cube nets is identifying opposite faces. Faces are opposite if they are separated by one face in a straight line in the net, or by other standard rules depending on the net's shape.

**3. Check Net A:**

Identify opposite faces: Trace paths or visualize folding. In Net A, Teal is opposite Blue, Red is opposite Yellow, and Green is opposite Purple. → **Wrong opposite faces**

Check required adjacencies: The target cube requires Red, Blue, and Green to be adjacent. Since none of these are opposite pairs in Net A (Red/Yellow, Blue/Teal, Green/Purple), they can be adjacent.

Visualize folding: It's possible to fold Net A such that Red is Top, Blue is Front, and Green is Right. (For example, place Red top, Yellow bottom. Arrange neighbours Teal, Purple, Green around Red. Blue is opposite Teal. One possible fold results in Top:Red, Bottom:Yellow, Front:Purple, Right:Green, Back:Blue, Left:Teal. Rotating this gives the target view.)

Thus, Net A can form the cube.

**4. Check Net B:**

Identify opposite faces: In Net B, Yellow is opposite Red, Teal is opposite Green, and Purple is opposite Blue.

Check required adjacencies: The target cube requires Red, Blue, and Green to be adjacent. None of these are opposite pairs in Net B (Red/Yellow, Blue/Purple, Green/Teal), so they can be adjacent.

Visualize folding: It's possible to fold Net B to match the target cube. (For example, one fold gives Top:Blue, Bottom:Purple, Front:Green, Back:Teal, Left:Yellow, Right:Red. This can be rotated to match the target view: Red-Top, Blue-Front, Green-Right).

Thus, Net B can form the cube.

**5. Check Net C:**

Identify opposite faces: In Net C, Yellow is opposite Blue, Red is opposite Purple, and Green is opposite Teal.

Check required adjacencies: The target cube requires Red, Blue, and Green to be adjacent. None of these are opposite pairs in Net C (Red/Purple, Blue/Yellow, Green/Teal), so they can be adjacent.

Visualize folding: It's possible to fold Net C to match the target cube. (For example, one fold gives Top:Teal, Bottom:Green, Front:Red, Back:Purple, Left:Yellow, Right:Blue. This configuration can be rotated into the target view Red-Top, Blue-Front, Green-Right).

Thus, Net C can form the cube.

**6. Check Net D:**

Identify opposite faces: In Net D, Blue is opposite Red, Purple is opposite Teal, and Green is opposite Yellow.

Check required adjacencies: The target cube shows Red, Blue, and Green meeting at a vertex, which requires Red and Blue to be adjacent. However, in Net D, Red and Blue are opposite faces.

Conclusion:

Opposite faces cannot meet at a vertex or share an edge. Since Net D places Red and Blue opposite each other, it is impossible to fold Net D into the cube shown on the left where Red and Blue are adjacent. Thus, Net D cannot form the cube.

**7. Final Answer:**

Based on the analysis, Nets A, B, and C can form the target cube, while Net D cannot because it places two adjacent faces from the target cube (Red and Blue) in opposite positions.

**Final Answer: D**

Figure 25: Test case of Cube Unfolding Task from Gemini-2.5-pro.

## Mental Folding-Cube Unfolding-Level 2

**Prompt:** You should first provide a reasoning process, then provide a single option(A, B, C or D) as the final answer. The reasoning process and the answer are enclosed within <think></think> and <answer></answer> tags, respectively, i.e., <think>reasoning process</think>, <answer>answer</answer>.

**Question:** The left image shows a cube with different patterns on its six faces from a particular viewing angle. The options are nets (unfolded patterns) of the cube, which are folded upward to form the cube. Which net, when folded, cannot form the cube shown in the left image? Please answer from options A, B, C, or D.

**Choices:**
   A. A  B. B  C. C  D. D

**Ground Truth Answer:** A

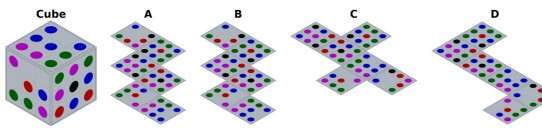

**Thinking Process:**

  Let's analyze the visible faces of the cube:

  Top face (T): Pattern of Blue, Purple, Green dots. Specifically, (B B B / P . P / G G G). The bottom-right corner dot is Green (G).

  Front face (F): Pattern of Purple, Green, Red, Blue dots. Specifically, (P G / G R / G B). The top-right corner dot is Green (G).

  Right face (R): Pattern of Green, Purple, Black, Blue, Red dots. Specifically, (G P . / G Bl B / R B R). The top-left corner dot is Green (G).

  The common vertex where these three faces (T, F, R) meet has a Green dot from each face. So, this vertex is (G, G, G).

  **Now let's examine each net:**

  **Net A:**

  Let's identify the patterns T, F, R in Net A if they exist.

  Face 2 of Net A matches T. (B B B / P . P / G G G)

  Face 3 of Net A matches F. (P G / G R / G B)

  Now, we need to find face R: (G P . / G Bl B / R B R).

  Let's list the patterns of the other faces in Net A:

  Face 1: (B G . / M R . / Bl B .) (M=Magenta/Purple). Face 4: (Bl B . / P R . / G P .) Face 5: (P Bl . / B R . / G M .) Face 6: (P G . / B M . / B R .) Comparing R with faces 1, 4, 5, 6: None of them match the pattern of R. Since face R is not present in Net A, Net A cannot form the cube shown.

  **Let's verify other nets,** assuming this is the primary criterion. If other nets contain T, F, R and can be arranged correctly, then A is the answer.

  **Net B:**

  Face 2 is T. Face 3 is F. Face 4 is R. These faces are arranged such that T is above F, and R is to the right of F. This is the standard T, F, R configuration.

  Let's check the common vertex: T (face 2): bottom-right dot is G. F (face 3): top-right dot is G. R (face 4): top-left dot is G. So, the common vertex is (G,G,G). This is consistent with the cube. Thus, Net B can form the cube.

  **Net C:**

  Face 1 is T. Face 3 is F. Face 4 is R. These faces are arranged such that T is above F, and R is to the right of F. This is the standard T, F, R configuration. Let's check the common vertex:

  T (face 1): bottom-right dot is G. F (face 3): top-right dot is G. R (face 4): top-left dot is G. So, the common vertex is (G,G,G). This is consistent with the cube. Thus, Net C can form the cube.

  **Net D:**

  The topmost face is R. The face below it is T. The face to the left of T is F. So, R, T, F are present. Arrangement when folded: R is Top, T is Front, F is Left.

  Let's find the common vertex for this configuration: R (Top): bottom-left dot is R (from R B R). T (Front): top-left dot is B (from B B B). F (Left): top-right dot is G (from P G). So, the common vertex is (R, B, G). For this to match the (G,G,G) vertex of the cube, we would need R=G and B=G, which is false as the colors are distinct. Therefore, Net D cannot form the cube shown with this specific corner.

  However, the primary reason for Net A being unable to form the cube is the absence of face R. This is a more fundamental issue than corner matching if all faces are present. If a face is missing, it absolutely cannot form the cube.

  If we assume that there's only one incorrect option, and the "missing face" criterion for A is decisive, then A is the answer. The issues with D (and potential edge matching issues with B and C) might be secondary or based on misinterpretations of "forming the cube". But missing a face is a definitive failure. Let's reconfirm the "missing face R in Net A" argument. R = (G P . / G Bl B / R B R)

  Net A, Face 1: (B G / M R / Bl B) - No match (e.g., G at top-left of R, B at top-left of Face 1)

  Net A, Face 4: (Bl B / P R / G P) - No match (e.g., G at top-left of R, Bl at top-left of Face 4)

  Net A, Face 5: (P Bl / B R / G M) - No match (e.g., G at top-left of R, P at top-left of Face 5)

  Net A, Face 6: (P G / B M / B R) - No match (e.g., G at top-left of R, P at top-left of Face 6)

  **The conclusion** that face R is not present in Net A seems correct. Therefore, Net A cannot form the cube. The final answer is $\boxed{A}$.

**Final Answer: A**

Figure 26: Test case of Cube Unfolding Task from Gemini-2.5-pro.

## Mental Folding-Cube Reconstruction-Level 1

**Prompt：** You should first provide a reasoning process, then provide a single option(A, B, C or D) as the final answer. The reasoning process and the answer are enclosed within <think></think> and <answer></answer> tags, respectively, i.e., <think>reasoning process</think>, <answer>answer</answer>.

**Question:** The left image shows the net (unfolded pattern) of a cube, with six faces having different patterns. The net is folded upward to form a cube. From an axonometric (3D) viewing angle of the cube, which combination of adjacent patterns is possible to see? Please answer from options A, B, C, or D.

**Choices:**
A. A   B. B   C. C   D. D

**Ground Truth Answer:** A

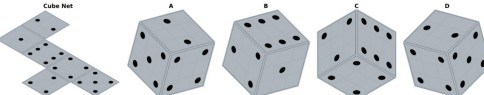

**Thinking Process:**

**1. Identify the faces and their opposite pairs:**
Label the faces in the net based on the number of dots: Top row: 4; Middle row: 3, 2, 6; Bottom row: 1, 5 (where 1 is below 3, and 5 is below 6) → **Wrong pattern detection**
Determine opposite faces by folding mentally or noting faces separated by one other face in a straight line (or around corners):
The face with 4 dots is opposite the face with 1 dot.
The face with 3 dots is opposite the face with 6 dots.
The face with 2 dots is opposite the face with 5 dots.

**2. Evaluate the options based on opposite faces:**
A cube view can only show faces that are adjacent to each other. Opposite faces can never be seen together.
Option A: Shows faces with 4 dots, 3 dots, and 1 dot. Since 4 and 1 are opposite faces, this view is impossible.
Option B: Shows faces with 5 dots, 3 dots, and 2 dots. Since 2 and 5 are opposite faces, this view is impossible.
Option D: Shows faces with 3 dots, 6 dots, and 5 dots. Since 3 and 6 are opposite faces, this view is impossible.
Option C: Shows faces with 2 dots, 4 dots, and 3 dots. None of these pairs (2-4, 2-3, 4-3) are opposite pairs (Opposites: 2-5, 4-1, 3-6). Therefore, this combination of adjacent faces is possible.

**3. Verify Option C with adjacency and orientation (Optional but recommended):**
Consider the face with 2 dots (let's call it F2) as the front face.
In the net, the face with 4 dots (F4) is attached to the top edge of F2 (relative to the net layout shown). When folded, F4 becomes the top face.
In the net, the face with 3 dots (F3) is attached to the left edge of F2. When folded, F3 becomes the left face.
So, the faces 2, 4, and 3 can indeed meet at the top-front-left vertex.
Check orientations shown in C:
Front (2 dots): The dots run diagonally from top-left to bottom-right. This matches the orientation on the net relative to the edges connected to F4 and F3.
Top (4 dots): The dots form a square. This matches the orientation on the net relative to the edge connected to F2.
Left (3 dots): The dots run vertically. The middle dot is closer to the edge shared with F2 (the front face). This matches the orientation on the net relative to the edge connected to F2.
The adjacencies and orientations in Option C are consistent with the net.

**Conclusion:**
Options A, B, and D are impossible because they show opposite faces.
Option C shows three faces that are adjacent in the net and whose orientations are consistent with folding.
**Final Answer: C**

Figure 27: Test case of Cube Reconstruction Task from Gemini-2.5-pro.

**Visual Penetration-Cross Section-Level 0**

**Prompt:** You should first provide a reasoning process, then provide a single option(A, B, C or D) as the final answer. The reasoning process and the answer are enclosed within <think></think> and <answer></answer> tags, respectively, i.e., <think>reasoning process</think>, <answer>answer</answer>.

**Question:** The top row shows the combined shape viewed from two different angles. The shape consists of a cone on top of a square frustum. Which of the following images cannot be a cross-section of the shape? Please answer from options A, B, C, or D.

**Choices:**
    A. A  B. B  C. C  D. D

**Ground Truth Answer:** C

**Thinking Process:**
    **1. Analyze the 3D Shape:**
    The object consists of a cone positioned inside a circular depression on the top face of a square frustum. The base of the cone is circular and lies on the plane of the top face of the frustum. The frustum has a square base and tapers upwards to a smaller square top face. The circular depression (and thus the cone's base) is contained entirely within this top square face.
    **2. Analyze Option A (Circle):**
    A horizontal cross-section plane passing through the cone part (above the frustum's top face) will intersect the cone to form a circle. This is a possible cross-section.
    **3. Analyze Option D (Square):**
    A horizontal cross-section plane passing through the frustum part below the level of the cone's base (and below the depression) will intersect the frustum to form a square. This is a possible cross-section.
    **4. Analyze Option C (Triangle on Trapezoid):**
    A vertical cross-section plane passing through the central axis of both the cone and the frustum, and parallel to two opposite side faces of the square frustum, will intersect the cone to form a triangle and the frustum to form an isosceles trapezoid. The triangle will sit atop the trapezoid. This is a possible cross-section.
    **5. Analyze Option B (Curve on Trapezoid):** → **Correct and detailed analysis**
    This shape represents a cross-section formed by a plane cutting through both the cone and the frustum. The curved top part represents the intersection with the cone. If the plane is vertical and off-center, this curve is typically a hyperbola segment. If the plane is tilted, it could be an ellipse or parabola segment. The trapezoidal bottom part represents the intersection with the frustum. A vertical plane cutting the frustum can result in a rectangle (if parallel to a side face) or a trapezoid (if cutting diagonally or at an angle relative to the side faces). A tilted plane could also result in a trapezoid.
    The key feature to examine is the junction between the curved top and the trapezoidal base. This junction line represents the intersection of the cutting plane with the plane containing the cone's base (the top surface of the frustum). The width of the curved section at its base corresponds to the length of the chord formed by the cutting plane intersecting the cone's circular base. The width of the top side of the trapezoid corresponds to the length of the line segment formed by the cutting plane intersecting the frustum's top square face. Since the cone's circular base is strictly smaller than and contained within the frustum's top square face (as seen in Views 1 and 2), any cutting plane that intersects both the circle and the square will create a chord across the circle that is strictly shorter than the segment across the square (unless the plane is merely tangent to the circle). Therefore, the base width of the curved (cone) section must be strictly narrower than the top width of the trapezoidal (frustum) section. However, in option B, the base width of the curve appears to be equal to the top width of the trapezoid. This configuration is geometrically impossible given that the cone's base circle lies inside the top square face of the frustum.
    **6. Conclusion:** → **Incorrect identification of the relative length relationship of options**
    Options A, C, and D represent possible cross-sections of the described 3D object.
    Option B depicts a configuration where the width of the cone section matches the width of the frustum section at the joining plane, which contradicts the geometry shown in Views 1 and 2 where the cone base is smaller than the top face of the frustum.
    Therefore, B cannot be a cross-section.
**Final Answer: B**

Figure 28: Test case of Cross-Section Task from Gemini-2.5-pro.

**Visual Penetration-Cube Counting-Level 1**

**Prompt:** You should first provide a reasoning process, then provide a single option(A, B, C or D) as the final answer. The reasoning process and the answer are enclosed within <think></think> and <answer></answer> tags, respectively, i.e., <think>reasoning process</think>, <answer>answer</answer>.

**Question:** Given three views, what is the **minimum** number of cubes required to satisfy the constraints shown in the images? Please answer from options A, B, C, or D.

**Choices:**

A. All three other options are incorrect   B. 9   C. 11   D. 10

**Ground Truth Answer:** B

**Thinking Process:** → **Incorrect details, but based on the wrong details, the analysis is correct**

**1. Understand the Goal:**

The objective is to find the minimum number of unit cubes required to build a 3D structure whose orthographic projections (Front, Top, Left views) match the given diagrams.

**2. Analyze the Top View:**

The Top view shows the layout of the cubes on the base level (or the projection of all cubes onto the base). It occupies a 2x3 grid area, with the cell in the top-right corner (row 1, column 3) and the bottom-right cell (row 2, column 3) being potentially different.

The cells occupied are: Top row (back): (1,1), (1,2), (1,3); Bottom row (front): (2,1), (2,2); Cell (2,3) is empty.

This means there must be at least one cube stacked at each of the locations (1,1), (1,2), (1,3), (2,1), and (2,2).

This establishes a minimum of 5 cubes based on the footprint.

**3. Analyze the Front View:** → **Wrong order**

The Front view shows the maximum height of the stacks in each column (left-to-right).

Column 1 (left): Height is 3. Corresponds to locations (1,1) and (2,1). So, max(Height(1,1), Height(2,1)) = 3.

Column 2 (middle): Height is 3. Corresponds to locations (1,2) and (2,2). So, max(Height(1,2), Height(2,2)) = 3.

Column 3 (right): Height is 1. Corresponds to location (1,3) (since (2,3) is empty according to the top view). So, Height(1,3) = 1.

**4. Analyze the Left View:**

The Left view shows the maximum height of the stacks in each row (front-to-back, viewed from the left).

Back Row (left side of the Left view): Height is 3. Corresponds to locations (1,1), (1,2), (1,3). So, max(Height(1,1), Height(1,2), Height(1,3)) = 3.

Front Row (right side of the Left view): Height is 1. Corresponds to locations (2,1), (2,2). So, max(Height(2,1), Height(2,2)) = 1.

**5. Combine Information and Minimize Cubes:**

Let H(r, c) denote the height of the stack at row 'r' and column 'c'.

From Top view: H(1,1)>=1, H(1,2)>=1, H(1,3)>=1, H(2,1)>=1, H(2,2)>=1.

From Front view: Height(1,3) = 1.

From Left view: max(H(2,1), H(2,2)) = 1.

Since we know H(2,1)>=1 and H(2,2)>=1, this forces H(2,1) = 1 and H(2,2) = 1.

Now use the Front view constraints again:

max(H(1,1), H(2,1)) = 3. Since H(2,1) = 1, we must have H(1,1) = 3 to satisfy the maximum height and minimize cubes.

max(H(1,2), H(2,2)) = 3. Since H(2,2) = 1, we must have H(1,2) = 3 to satisfy the maximum height and minimize cubes.

Check the remaining Left view constraint: max(H(1,1), H(1,2), H(1,3)) = max(3, 3, 1) = 3.

This is satisfied by our determined heights.

**6. Calculate Total Cubes:** → **The error does not affect the final numerical result**

We have determined the minimum required heights for each stack: H(1,1) = 3, H(1,2) = 3, H(1,3) = 1, H(2,1) = 1, H(2,2) = 1

Total number of cubes = H(1,1) + H(1,2) + H(1,3) + H(2,1) + H(2,2)

Total cubes = 3 + 3 + 1 + 1 + 1 = 9

**7. Compare with Options:**

The calculated minimum number of cubes is 9. This matches option B.

**Final Answer: B**

Figure 29: Test case of Cube Counting Task from Gemini-2.5-pro.

## Visual Penetration-Cube Assembly-Level 0

**Prompt:** You should first provide a reasoning process, then provide a single option(A, B, C or D) as the final answer. The reasoning process and the answer are enclosed within <think></think> and <answer></answer> tags, respectively, i.e., <think>reasoning process</think>, <answer>answer</answer>.

**Question:** The top left image shows the original complete cube stack made of equal-sized cubes. It can be formed by combining the small cube stack on the right(part 1) with one of the options below. Which option completes the original cube stack? Please answer from options A, B, C, or D.

**Choices:**

A. A   B. B   C. C   D. All three other options are incorrect

**Ground Truth Answer:** B

**Thinking Process: → Most of the identifications of the number of cube stacks are incorrect**

**1. Analyze the Complete Cube Stack:**

Visualize the stack layer by layer or column by column. Let's use columns from back-left to front-right (like reading order if viewed from slightly above).

Back-Left Column: 2 cubes high.
Back-Right Column: 2 cubes high.
Middle-Left Column: 2 cubes high.
Middle-Right Column: 1 cube high (only bottom layer).
Front-Left Column: 1 cube high (only bottom layer).
Front-Right Column: 1 cube high (only bottom layer).
Total Cubes = 2 + 2 + 2 + 1 + 1 + 1 = 9 cubes.

**2. Analyze Part 1:**

Using the same column description:
Back-Left Column: 2 cubes high.
Back-Right Column: 0 cubes.
Middle-Left Column: 1 cube high (only bottom layer).
Middle-Right Column: 1 cube high (only bottom layer).
Front-Left Column: 1 cube high (only bottom layer).
Front-Right Column: 0 cubes.
Total Cubes = 2 + 0 + 1 + 1 + 1 + 0 = 5 cubes.

**3. Determine the Missing Part (Complete Stack - Part 1): → Incorrect approach to calculation**

Subtract the cubes in Part 1 from the Complete Stack for each column position:
Back-Left Column: 2 (Complete) - 2 (Part 1) = 0 cubes.
Back-Right Column: 2 (Complete) - 0 (Part 1) = 2 cubes (column 2 high).
Middle-Left Column: 2 (Complete) - 1 (Part 1) = 1 cube (must be the top cube).
Middle-Right Column: 1 (Complete) - 1 (Part 1) = 0 cubes.
Front-Left Column: 1 (Complete) - 1 (Part 1) = 0 cubes.
Front-Right Column: 1 (Complete) - 0 (Part 1) = 1 cube (must be the bottom cube).
Total Cubes in Missing Part = 0 + 2 + 1 + 0 + 0 + 1 = 4 cubes.
Structure of Missing Part: A column of 2 cubes in the back-right position, a single cube in the middle-left top position, and a single cube in the front-right bottom position.

**4. Analyze the Options:**

Option A: Has a column of 2 in the back-left, 1 in middle-left (bottom), 1 in front-left (bottom). Total 4 cubes. Does not match the required structure.

Option B: Has a column of 2 in back-left, 1 in back-right (bottom), 1 in middle-left (bottom), 1 in middle-right (bottom). Total 5 cubes. Incorrect number of cubes and structure.

Option C: Has a column of 2 in the back-right position, a single cube that aligns with the middle-left top position, and a single cube that aligns with the front-right bottom position. Total 4 cubes. This matches the required structure derived in step 3.

**5. Conclusion:**

Option C is the missing part that combines with Part 1 to form the Complete Cube Stack.

**Final Answer: C**

Figure 30: Test case of Cube Assembly Task from Gemini-2.5-pro.

## Mental Animation-Arrow Moving-Level 0

**Prompt:** You should first provide a reasoning process, then provide a single option(A, B, C or D) as the final answer. The reasoning process and the answer are enclosed within <think></think> and <answer></answer> tags, respectively, i.e., <think>reasoning process</think>, <answer>answer</answer>.

**Question:** In the diagram, the red arrow is the initial arrow, and the green arrow is the final arrow. The arrow can move in four directions (forward, backward, left, right), where 'forward' always refers to the current direction the arrow is pointing. After each movement, the arrow's direction is updated to the direction of movement. Which of the following paths can make the arrow move from the starting position to the ending position? Please answer from options A, B, C, or D.

**Choices:**

    A. (Left, 2 units)--(Left, 1 unit)
    B. (Forward, 1 unit)--(Backward, 1 unit)
    C. (Forward, 1 unit)--(Backward, 2 units)
    D. (Forward, 1 unit)--(Left, 1 unit)\

**Ground Truth Answer:** D

**Thinking Process:**

    **1. Analyze Initial State:**
The red arrow (start) is located in the top-left quadrant. Its center appears to be at coordinates approximately (0.5, 1.5).
The direction of the red arrow is pointing downwards (South).
    **2. Analyze Final State:**
The green arrow (end) is located in the bottom-right quadrant. Its center appears to be at coordinates approximately (1.5, 0.5).
The direction of the green arrow is pointing to the right (East).
    **3. Define Movement Rules:**
The arrow can move Forward, Backward, Left, or Right relative to its current orientation. After each move, the arrow's orientation changes to the direction of the move. Cardinal Directions relative to current orientation:
    If facing North (Up): Forward=Up, Backward=Down, Left=West, Right=East.
    If facing South (Down): Forward=Down, Backward=Up, Left=East, Right=West.
    If facing East (Right): Forward=Right, Backward=Left, Left=Up, Right=Down.
    If facing West (Left): Forward=Left, Backward=Right, Left=Down, Right=Up.    **→ Fully comprehend the rules**
    **4. Evaluate Option A: (Left, 2 units) -- (Left, 1 unit)**
Start: Position (0.5, 1.5), Direction Down.
Move 1: Left, 2 units. Relative to Down, Left is East (Right). Move 2 units East.
-- New Position: (0.5 + 2, 1.5) = (2.5, 1.5). New Direction: East (Right).
Move 2: Left, 1 unit. Relative to East (Right), Left is North (Up). Move 1 unit North.
-- New Position: (2.5, 1.5 + 1) = (2.5, 2.5).
New Direction: North (Up).
Result: Position (2.5, 2.5), Direction Up. Does not match the final state.
    **5. Evaluate Option B: (Forward, 1 unit) -- (Backward, 1 unit)**
Start: Position (0.5, 1.5), Direction Down.
Move 1: Forward, 1 unit. Relative to Down, Forward is Down. Move 1 unit Down.
-- New Position: (0.5, 1.5 - 1) = (0.5, 0.5). New Direction: Down.
Move 2: Backward, 1 unit. Relative to Down, Backward is Up. Move 1 unit Up.
-- New Position: (0.5, 0.5 + 1) = (0.5, 1.5). New Direction: Up.
Result: Position (0.5, 1.5), Direction Up. Does not match the final state.
    **6. Evaluate Option C: (Forward, 1 unit) -- (Backward, 2 units)**
Start: Position (0.5, 1.5), Direction Down.
Move 1: Forward, 1 unit. Relative to Down, Forward is Down. Move 1 unit Down.
-- New Position: (0.5, 1.5 - 1) = (0.5, 0.5). New Direction: Down.
Move 2: Backward, 2 units. Relative to Down, Backward is Up. Move 2 units Up.
-- New Position: (0.5, 0.5 + 2) = (0.5, 2.5). New Direction: Up.
Result: Position (0.5, 2.5), Direction Up. Does not match the final state.
    **7. Evaluate Option D: (Forward, 1 unit) -- (Left, 1 unit)**
Start: Position (0.5, 1.5), Direction Down.
Move 1: Forward, 1 unit. Relative to Down, Forward is Down. Move 1 unit Down.
-- New Position: (0.5, 1.5 - 1) = (0.5, 0.5). New Direction: Down.
Move 2: Left, 1 unit. Relative to Down, Left is East (Right). Move 1 unit East.
-- New Position: (0.5 + 1, 0.5) = (1.5, 0.5). New Direction: East (Right).
Result: Position (1.5, 0.5), Direction East (Right). Matches the final state (green arrow).
    **8. Conclusion:** Option D correctly describes the path from the red arrow to the green arrow.
**Final Answer: D**

Figure 31: Test case of Arrow Moving Task (Level 0) from Gemini-2.5-pro.

## Mental Animation-Arrow Moving-Level 1

**Prompt：** You should first provide a reasoning process, then provide a single option(A, B, C or D) as the final answer. The reasoning process and the answer are enclosed within <think></think> and <answer></answer> tags, respectively, i.e., <think>reasoning process</think>, <answer>answer</answer>.

**Question:** The left image shows the initial state. Arrows can move in four directions (forward, backward, left, right), where 'forward' always refers to the current direction the arrow is pointing. After each movement, the arrow's direction is updated to the direction of movement. If the target position is empty, the arrow can move there directly; otherwise, it needs to swap with the arrow at the target position, and both arrows' movements should satisfy the aforementioned requirements. After the transformations '((0, 0) Left, 2 units)--((1, 0) Right, 1 unit)', which state from the options can be reached? Please answer from options A, B, C, or D.

**Choices:**
  A. A   B. B   C. C   D. D
**Ground Truth Answer:** C

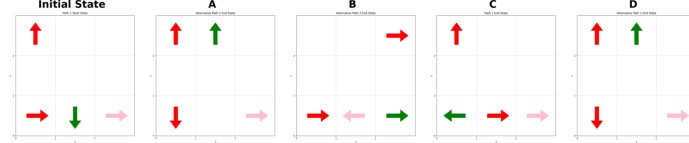

**Thinking Process:**
**1. Analyze Initial State:**
Grid coordinates range from x=0 to x=2 and y=0 to y=2 (implicitly, based on positions).
Arrow 1: Red at (0, 0), Orientation: Right (East).
Arrow 2: Green at (1, 0), Orientation: Down (South).
Arrow 3: Pink at (2, 0), Orientation: Right (East).
Arrow 4: Red at (0, 2), Orientation: Up (North).
**2. Analyze Transformation 1:** ((0, 0) Left, 2 units)
Identify the arrow: The arrow starting at (0, 0) is Arrow 1 (Red, facing Right).
Determine the movement direction: 'Left' relative to the arrow's orientation (Right). If facing Right, 'Left' is Up (North).
Determine the target position: Move 2 units Up from (0, 0). Target is (0, 0 + 2) = (0, 2).
Check the target position: Position (0, 2) is occupied by Arrow 4 (Red, facing Up).
Execute swap: Since the target is occupied, Arrow 1 and Arrow 4 must swap positions.
Arrow 1 moves from (0, 0) to (0, 2). Movement direction: Up. New orientation for Arrow 1: Up.
Arrow 4 must move from (0, 2) to (0, 0). Movement direction: Down.
Check if Arrow 4's move is valid: Arrow 4 is facing Up.
Moving Down is 'Backward'. This is a valid relative move.
New orientation for Arrow 4: Down (direction of movement).
State after Transformation 1: Arrow 4: Red at (0, 0), Orientation: Down. Arrow 2: Green at (1, 0), Orientation: Down. Arrow 3: Pink at (2, 0), Orientation: Right. Arrow 1: Red at (0, 2), Orientation: Up.
**3. Analyze Transformation 2:** ((1, 0) Right, 1 unit)
Identify the arrow: The arrow currently at (1, 0) is Arrow 2 (Green, facing Down).
Determine the movement direction: 'Right' relative to the arrow's orientation (Down). If facing Down, 'Right' is Left (West).
Determine the target position: Move 1 unit Left from (1, 0). Target is (1 - 1, 0) = (0, 0).
Check the target position: Position (0, 0) is occupied by Arrow 4 (Red, facing Down).
Execute swap: Since the target is occupied, Arrow 2 and Arrow 4 must swap positions.
Arrow 2 moves from (1, 0) to (0, 0). Movement direction: Left. New orientation for Arrow 2: Left.
Arrow 4 must move from (0, 0) to (1, 0). Movement direction: Right.
Check if Arrow 4's move is valid: Arrow 4 is facing Down. Moving Right is 'Right'. This is a valid relative move.
New orientation for Arrow 4: Right (direction of movement).
State after Transformation 2 (Final State): Arrow 2: Green at (0, 0), Orientation: Left. Arrow 4: Red at (1, 0), Orientation: Right. Arrow 3: Pink at (2, 0), Orientation: Right. Arrow 1: Red at (0, 2), Orientation: Up.
**4. Compare Final State with Options:**
Option A: Incorrect arrows/positions/orientations.
Option B: Incorrect arrows/positions/orientations.
Option C: Green at (0, 0), facing Left. (Match); Red at (1, 0), facing Right. (Match); Pink at (2, 0), facing Right. (Match); Red at (0, 2), facing Up. (Match). This matches the derived final state.
Option D: Incorrect arrows/positions/orientations.
**5. Conclusion:** The state reached after the transformations matches Option C.
**Final Answer: C**

Figure 32: Test case of Arrow Moving Task (Level 1) from Gemini-2.5-pro.

## Mental Animation-Block Moving-Level 1

**Prompt:** You should first provide a reasoning process, then provide a single option(A, B, C or D) as the final answer. The reasoning process and the answer are enclosed within <think></think> and <answer></answer> tags, respectively, i.e., <think>reasoning process</think>, <answer>answer</answer>.

**Question:** The top row of images shows different views of the initial state of a cube stack, while the bottom row shows different views of the final state after transformation. During the transformation process, blocks can move one unit in any direction (forward, backward, left, right, up, down). If the target position is empty, the block can move there directly; if the target position already has a block, they swap places. Blocks cannot float in the air. If a block is moved away from a position, any block above it will fall down until reaching a supporting surface. The xyz axes are shown in the diagram, and each block's position can be precisely identified using coordinates (x1,y1,z1). Which of the following transformation sequences can change the cube stack from the initial state to the final state shown in the diagram? Please answer from options A, B, C, or D.

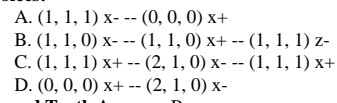

**Choices:**
    A. (1, 1, 1) x- -- (0, 0, 0) x+
    B. (1, 1, 0) x- -- (1, 1, 0) x+ -- (1, 1, 1) z-
    C. (1, 1, 1) x+ -- (2, 1, 0) x- -- (1, 1, 1) x+
    D. (0, 0, 0) x+ -- (2, 1, 0) x-

**Ground Truth Answer:** B

**Thinking Process:**
    **1. Analyze States:**
    Identify block positions (R, G, DB, P, C) in Initial and Final states using a consistent coordinate system (e.g., R at 0,0,0).
    Initial: R(0,0,0), G(1,0,0), DB(0,1,0), P(1,1,0), C(1,1,1).   → **Missing 1 block and the positions of blocks are wrong**
    Final: R(0,0,0), DB(1,0,0), G(0,1,0), P(1,1,0), C(1,1,1).
    The change is G swaps with DB.
    **2. Analyze Rules:**
     Moves are 1 unit. Target empty = move. Target occupied = swap. Gravity applies if support is removed.
    **3. Evaluate Options:**
    A: (1,1,1)x- moves C to (0,1,1). (0,0,0)x+ swaps R and G.
    Result: G(0,0,0), R(1,0,0), DB(0,1,0), P(1,1,0), C(0,1,1). Incorrect state.
    B: (1,1,0)x- swaps P and DB. (1,1,0)x+ moves DB(now at 1,1,0) to (2,1,0), causing C(1,1,1) to fall to (1,1,0). (1,1,1)z- attempts to move from empty (1,1,1). Invalid.
    C: (1,1,1)x+ moves C(1,1,1) to (2,1,1), causing C to fall to (2,1,0). (2,1,0)x- swaps C(2,1,0) and P(1,1,0). (1,1,1)x+ attempts to move from empty (1,1,1). Invalid.
    D: (0,0,0)x+ swaps R(0,0,0) and G(1,0,0). State: G(0,0,0), R(1,0,0), DB(0,1,0), P(1,1,0), C(1,1,1). (2,1,0)x- attempts to move from empty (2,1,0). Invalid.
    **4. Conclusion:**
    All options appear invalid or incorrect based on the rules. B, C, D reference empty coordinates. A leads to the wrong state.
    **5. Reassess:**
    The problem likely contains an error. However, if forced to choose, D's first move is valid. The required transformation (swapping G and DB) needs multiple steps. I found a 3-step sequence: (0,1,0)y-, (0,0,0)x+, (0,0,0)y+. Another 3-step sequence starting like D: (0,0,0)x+, (0,1,0)x+, (0,0,0)y+. Since Option D has only 2 steps and the second is invalid, it cannot be correct as written. But in multiple-choice questions with flaws, sometimes the intended answer has the most 'reasonable' start or structure. Option D starts with a valid swap. Without clarification, choosing D assumes it's the intended answer despite the error.
**Final Answer: D**

Figure 33: Test case of Block Moving Task from Gemini-2.5-pro.

## Mental Animation-Mechanical System-Level 0

**Prompt:** You should first provide a reasoning process, then provide a single option(A, B, C or D) as the final answer. The reasoning process and the answer are enclosed within <think></think> and <answer></answer> tags, respectively, i.e., <think>reasoning process</think>, <answer>answer</answer>.

**Question:** In the coaxial system shown, the orange gear is fixed, and the yellow and green gears are identical in shape. If the yellow gear rotates clockwise, what is the motion of the green gear relative to the orange gear? Please choose from options A, B, C, or D.

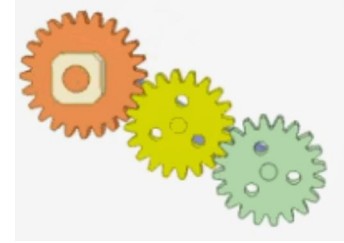

**Choices:**
    A. Clockwise rotation, clockwise revolution
    B. Clockwise rotation, counterclockwise revolution
    C. No rotation, clockwise revolution
    D. No rotation, counterclockwise revolution

**Ground Truth Answer:** C

**Thinking Process:** Can't intuitively thinking without theoretical derivation
    **1. Define the system and motion:**
    We have an epicyclic gear train. The orange gear (Sun) is fixed ($\omega_o = 0$). The yellow gear (Planet 1) meshes with the orange gear and rotates clockwise ($\omega_y < 0$). The green gear (Planet 2) meshes with the yellow gear, and is identical to it ($N_g = N_y$). The yellow and green gears are carried by an implicit arm that revolves around the center of the orange gear. We need to find the absolute rotation (spin) and revolution (orbit) of the green gear relative to the fixed orange gear.
    **2. Determine the arm's revolution ($\omega_{arm}$):**
    The relationship between the sun, planet, and arm is given by the Willis equation: $(\omega_y - \omega_{arm}) / (\omega_o - \omega_{arm}) = -N_o / N_y$. Substituting $\omega_o = 0$: $(\omega_y - \omega_{arm}) / (-\omega_{arm}) = -N_o / N_y$. Simplifying: $\omega_y - \omega_{arm} = (-\omega_{arm}) (-N_o / N_y) = \omega_{arm} (N_o / N_y)$. Rearranging for $\omega_y$: $\omega_y = \omega_{arm} + \omega_{arm} (N_o / N_y) = \omega_{arm} (1 + N_o / N_y)$. We are given that $\omega_y$ is clockwise (negative). Since $N_o$ and $N_y$ are positive tooth counts, the term $(1 + N_o / N_y)$ is positive. Therefore, for $\omega_y$ to be negative, $\omega_{arm}$ must also be negative. This means the arm, carrying the yellow and green gears, revolves clockwise around the orange gear.
    **3. Determine the green gear's rotation ($\omega_g$):**
    Now consider the meshing between the yellow and green gears. They are carried by the same arm. The relative motion formula applies: $(\omega_g - \omega_{arm}) / (\omega_y - \omega_{arm}) = -N_y / N_g$. Since the yellow and green gears are identical, $N_y = N_g$. The ratio $-N_y / N_g = -1$. So, $(\omega_g - \omega_{arm}) / (\omega_y - \omega_{arm}) = -1$. $\omega_g - \omega_{arm} = -(\omega_y - \omega_{arm}) = -\omega_y + \omega_{arm}$. Rearranging for $\omega_g$: $\omega_g = -\omega_y + 2 \omega_{arm}$.
    **4. Express $\omega_g$ in terms of $\omega_y$ and gear ratio:**
    Substitute the expression for $\omega_{arm}$ from step 2 [$\omega_{arm} = \omega_y / (1 + N_o / N_y)$] into the equation for $\omega_g$: $\omega_g = -\omega_y + 2 [\omega_y / (1 + N_o / N_y)]$. Factor out $\omega_y$: $\omega_g = \omega_y [-1 + 2 / (1 + N_o / N_y)]$. Combine terms inside the bracket: $\omega_g = \omega_y [(-1 (1 + N_o / N_y) + 2) / (1 + N_o / N_y)]$. $\omega_g = \omega_y [(-1 - N_o / N_y + 2) / (1 + N_o / N_y)]$. $\omega_g = \omega_y [(1 - N_o / N_y) / (1 + N_o / N_y)]$
    **5. Analyze the rotation based on relative sizes:**
    Case 1: If $N_o > N_y$ (Orange gear larger than Yellow, as suggested by the image). Then $(1 - N_o / N_y)$ is negative. Since $\omega_y$ is negative (clockwise), $\omega_g = (negative) (negative / positive) = positive$. Green gear rotates counter-clockwise. Motion: CCW rotation, CW revolution. (Not an option).
    Case 2: If $N_o < N_y$ (Orange gear smaller than Yellow). Then $(1 - N_o / N_y)$ is positive. $\omega_g = (negative) (positive / positive) = negative$. Green gear rotates clockwise. Motion: CW rotation, CW revolution. (Option A).
    Case 3: If $N_o = N_y$ (Orange gear same size as Yellow). Then $(1 - N_o / N_y) = 0$. $\omega_g = \omega_y (0 / positive) = 0$. Green gear has no rotation (absolute rotation is zero). Motion: No rotation, CW revolution. (Option C).
    **6. Select the most plausible answer:**
    Although the image suggests $N_o > N_y$, this leads to a result not listed in the options. Gear train problems in multiple-choice format often rely on specific configurations, sometimes simplifying assumptions (like equal gear sizes if not explicitly stated otherwise) that lead to one of the provided answers. Assuming the problem intends for the case where $N_o = N_y$ allows us to match option C.
    **7. Final Motion (assuming $N_o = N_y$):**
    Rotation of green gear ($\omega_g$): 0 (No rotation). Revolution of green gear ($\omega_{arm}$): Clockwise
**Final Answer: C**

Figure 34: Test case of Mechanical System Task from Gemini-2.5-pro.

