# OpenReview forum: "SpatialViz-Bench: A Cognitively-Grounded Benchmark for Diagnosing Spatial Visualization in MLLMs"
_ICLR.cc/2026/Conference — ICLR 2026 Poster_

### Official Review · Reviewer_nVip · 2025-10-25

**Soundness:** 3
**Presentation:** 2
**Contribution:** 3
**Rating:** 4
**Confidence:** 5

**Summary:**

This paper introduces SpatialViz-Bench, a new benchmark for diagnosing spatial visualization abilities in MLLMs. The authors address limitations and potential data contamination in existing benchmarks by creating a scalable framework that programmatically generates 1,180 novel problems across 12 distinct tasks. Their evaluation of 27 MLLMs reveals wide performance variations and shows that even state-of-the-art models struggle with these tasks. The diagnostic analysis suggests that these failures stem from fundamental deficits in perception and spatial transformation rather than high-level reasoning.

**Strengths:**

1. The use of a synthetic dataset is a significant strength, as it effectively mitigates the risk of data contamination and ensures comprehensive coverage across a wide range of tasks.
2. The benchmark demonstrates strong discriminative power, effectively highlighting performance differences across a broad spectrum of MLLMs.
3. The statistical error analysis, which breaks down failures into fundamental perceptual and spatial transformation deficits, provides valuable insights into the core limitations of current models.

**Weaknesses:**

1. Figure 1 is dense and difficult to parse due to information overload. The upper-right legend, in particular, is unclear: the meaning of the "error mark" and the reference to "existing evaluation" are not immediately evident.
2. The Related Work section requires revision. It currently reads more like an extension of the Introduction rather than an objective survey of the field. The tone is heavily opinion-based, with most existing work being cited primarily to emphasize their limitations. Furthermore, the section appears to neglect several of the most recent or concurrent benchmarks focused on spatial reasoning.[1-6].


References:

[1] Ramakrishnan, Santhosh Kumar, et al. "Does Spatial Cognition Emerge in Frontier Models?." arXiv preprint arXiv:2410.06468 (2024).

[2] Stogiannidis, Ilias, Steven McDonagh, and Sotirios A. Tsaftaris. "Mind the gap: Benchmarking spatial reasoning in vision-language models." arXiv preprint arXiv:2503.19707 (2025).

[3] Xu, Wenrui, et al. "Defining and Evaluating Visual Language Models' Basic Spatial Abilities: A Perspective from Psychometrics." arXiv preprint arXiv:2502.11859 (2025).

[4] Li, Chengzu, et al. "11plus-bench: Demystifying multimodal llm spatial reasoning with cognitive-inspired analysis." arXiv preprint arXiv:2508.20068 (2025).

[5] Tang, Kexian, et al. "LEGO-Puzzles: How Good Are MLLMs at Multi-Step Spatial Reasoning?." arXiv preprint arXiv:2503.19990 (2025).

[6] Yin, Baiqiao, et al. "Spatial Mental Modeling from Limited Views." arXiv preprint arXiv:2506.21458 (2025).

**Questions:**

1. The methodology for labeling error categories is unclear. Are these categories mutually exclusive (uni-label), or can a single failure case be assigned multiple labels (multi-label)? If it is a uni-label system, clarification is needed on how the "critical error" is determined or prioritized over other potential errors in a single instance.

2. Regarding the counter-intuitive finding that CoT prompting degrades performance on open-source models: have the authors investigated whether this could be attributed to issues such as parsing failures in the CoT responses?

---

> ### Author Response · Authors · 2025-11-19
> **Q1 to Reviewer nVip**
>
> Thank you for your feedback on presentation and methodology. We revised **Figure 1** to clarify the legend and capability gaps, and rewrote **part 2 of Related Work (Section 2)** to include missing references with an objective tone. We also clarified the multi-label error classification, reporting an inter-annotator agreement of $\kappa=0.88$ (**Appendix E.6.3**), and verified that CoT performance drops are not due to parsing failures (**Table 4, Section 4.2.3**). Revisions are highlighted in blue.
>
> &nbsp;
>
> ## Q1: Clarify the error labeling methodology (uni-label vs. multi-label) and how critical errors are prioritized.
>
> We clarify that we employed a multi-label classification system for error analysis.
>
> - **Rationale:** A single incorrect reasoning chain often involves cascading failures. For instance, an initial "Perceptual Error" (misidentifying a shape) can lead to a subsequent "Spatial Transformation Error" (rotating the wrong shape).
> - **Implementation:** Annotators were instructed to label all distinct error types present in a failure case. Consequently, the total frequency of error tags in our analysis exceeds the total number of incorrect questions.
> - **Validation:** To ensure the reliability of this multi-label approach, we calculated inter-annotator agreement using a binary decomposition method (treating each category as a Yes/No decision), achieving a macroscopic average Cohen's $\kappa$ of **0.88**. This is detailed in **Appendix E.6.3** of the revision.
>
> | **Error Category**      | **Cohen's κ** |
> | ----------------------- | ------------- |
> | Perceptual              | 0.90          |
> | Spatial Transformation  | 0.81          |
> | Methodological          | 0.75          |
> | Instruction Following   | 0.96          |
> | Spatial Memorization    | 0.89          |
> | Calculation & Reasoning | 1.00          |
> | **Average**             | **0.88**      |

---

> ### Author Response · Authors · 2025-11-19
> **Q2 to Reviewer nVip**
>
> ## Q2: Could the "CoT Paradox" (degradation in open-source models) be attributed to parsing failures in CoT responses?
>
> ### **1. Ruling out Parsing Failures (Sensitivity Analysis)**
>
> We investigated this possibility thoroughly and confirmed that parsing failures are **not** the cause.
>
> We compared our baseline **Extract Rule A** (truncated letter matching) against **Extract Rule B** (customized full-format regex matching). While Rule A is generic enough to cover diverse output types and Rule B targets specific patterns, the inherent flexibility of CoT responses means neither guarantees 100% extraction success. Crucially, the discrepancy between rules was negligible (<1.2%), and the negative impact of CoT persisted under both. This confirms that the "CoT Paradox" is a genuine reasoning failure, not a parsing artifact of Rule A.
>
> **Sensitivity to Extraction Rules (Acc. Drop%)** in Table 4 (b)
>
> | Model           | Rule A ↓ | Rule B ↓ |   Δ   |
> | :-------------- | :------: | :------: | :---: |
> | SAIL-VL-1.5-2B  |  -8.22   |  -7.29   | +0.93 |
> | Deepseek-VL2-3B |  -5.18   |  -5.01   | +0.17 |
> | Kimi-VL-16B     |  -8.47   |  -9.66   | -1.19 |
>
> ### **2. Statistical Root Cause Analysis**
>
> Having ruled out parsing artifacts, we performed paired significance tests ($p < 0.05$) to pinpoint the nature of the failure. The analysis reveals that the degradation is not random (as one might expect from formatting errors) but is highly concentrated in "pure-visual" spatial tasks (e.g., 3-View Projection, 3D Rotation).
>
> | **Model**            | **Source** | **CoT Impact** | **Significant (p<0.05)** |
> | -------------------- | ---------- | -------------- | ------------------------ |
> | Kimi-VL-A3B-Instruct | Open       | Negative       | Yes (p=0.0192)           |
> | Deepseek-VL2-tiny    | Open       | Negative       | Yes (p=0.0463)           |
> | InternVL2.5-78B      | Open       | Negative       | Yes (p=0.0368)           |
>
> This statistical evidence supports the hypothesis that for models not specifically optimized for reasoning, the mandate to generate text interferes with the holistic processing required for pure spatial tasks.
>
> ### **3. Qualitative Failure Analysis**
>
> Finally, to understand how this interference manifests, we analyzed specific cases where models succeeded with Direct Answering but failed with CoT ("Non-CoT ✓ $\rightarrow$ CoT ×"). We identified two primary mechanisms by which CoT actively induces errors:
>
> | **Failure Mode**           | **Description**                                              | **Representative Case**                                      |
> | -------------------------- | ------------------------------------------------------------ | ------------------------------------------------------------ |
> | **Visual Hallucination**   | Textual reasoning contradicts visual input, effectively "overwriting" correct visual perception with an erroneous text chain. | In **Three-View Projection**, the CoT correctly identifies shapes initially but then "hallucinates" a mirror flip operation that did not visually occur, leading to an incorrect choice despite accurate initial recognition. |
> | **Enforced Serialization** | The model attempts to decompose a parallel process (e.g., folding) into linear text steps, introducing logical errors. | In **Cube Unfolding**, the CoT attempts to verify face adjacencies sequentially. It fails to track simultaneous topological changes, logically inferring that two adjacent faces are "opposite" due to the linear constraints of text generation. |
>
> These findings confirm that CoT degradation is not merely a distraction but a conflict between **enforced serialization** (text generation) and **holistic visual processing** in models that lack a strong internal "world model" for spatial simulation.

---

> ### Author Response · Authors · 2025-11-19
> **Q3 to Reviewer nVip**
>
> ## Q3 (from Weakness 1): Figure 1 is dense and the legend ("error mark", "existing evaluation") is unclear.
>
> We appreciate this feedback and have revised Figure 1 in the updated paper to improve clarity.
>
> - **Legend Clarification:** We clarified the legend to explicitly convey our core motivation: while existing benchmarks cover **Spatial Perception** and **Spatial Memorization** (indicated by checks $\checkmark$), they largely neglect or fail to systematically assess **Spatial Visualization** (indicated by cross ✗).
>
> We have revised the legend to clearly distinguish between capabilities covered by existing benchmarks (marked with a check '✓') and the overlooked 'Spatial Visualization' capability (marked with a cross '✗'), intuitively highlighting the research gap.

---

> ### Author Response · Authors · 2025-11-19
> **Q4 to Reviewer nVip**
>
> ## Q4 (from Weakness 2): The Related Work section is subjective, misses recent references.
>
> We accept this criticism and have significantly revised Section 2 (Related Works) to ensure objectivity and accuracy.
>
> - **Correction of Citations:** We wish to clarify that **SRBench [2] (Stogiannidis et al., 2025)** and **[3] (Xu et al. 2025)** were indeed cited in **Appendix A.2** of our original submission. However, we apologize for a factual error where we inadvertently swapped their core characteristics. We have now corrected this in the revision to accurately state that **SRBench** focuses narrowly on "paper folding tasks," while the work by **Xu et al.** relies on "online psychological tests".
> - **Inclusion of Literature:** We have added the missed reference **[1] (Ramakrishnan et al., 2024)**. Regarding **[4-6] (Li et al., 2025; Tang et al., 2025; Yin et al., 2025)**, we note that these are concurrent works released near the submission deadline, but we have included them in the discussion of the revision.
> - **Tone Adjustment:** We have rewritten the section to objectively state contributions. The argumentative content regarding "data contamination" and the need for "dynamic test banks" has been moved to **Section 1** to serve as motivation rather than a critique of related work.

---

> > ### Comment · Reviewer_nVip · 2025-11-24
> >
> > Thanks for the authors' thorough explanations and newly included experiments about the "CoT Paradox". My concerns are effectively addressed. I'd like to increase the score to 6.

---

> > > ### Author Response · Authors · 2025-11-25
> > >
> > > Thank you for your prompt response. We are delighted that our further explanations and the new experiments concerning the 'CoT Paradox' have resolved your concerns. We are very grateful for your time in reconsidering our work and for the improved rating.

---

### Official Review · Reviewer_9vwG · 2025-10-26

**Soundness:** 2
**Presentation:** 3
**Contribution:** 3
**Rating:** 8
**Confidence:** 4

**Summary:**

The paper introduces SpatialViz-Bench, a cognitively-grounded, programmatically generated benchmark designed to evaluate spatial visualization abilities in Multi-modal Large Language Models (MLLMs). Covering 12 tasks across four sub-abilities (mental rotation, mental folding, visual penetration, and mental animation), the benchmark comprises 1,180 problems and enables scalable, contamination-resistant evaluation. Experiments on 27 MLLMs reveal significant performance gaps with humans and highlight that current models struggle primarily with perceptual and spatial transformation tasks, not high-level reasoning.

**Strengths:**

- Novelty & Relevance: Proposes a comprehensive, cognitively-motivated benchmark specifically targeting spatial visualization in MLLMs, addressing a clear gap in current evaluation practices. The "gap" identified here - whether LLMs can infer unseen relationships through
spatial visualization - is a novel and crucial problem, not intensively researched

- Methodological Rigor: The eye for detail in the error analysis as well as ensuring scalability are good grounds to cover in a benchmark paper. The idea of randomizing generations and dynamically updating the test bank is a valid thought for benchmark's robustness.

- Insightful Analysis: Provides detailed error analysis and ablation studies, revealing that model failures are rooted in perceptual and transformation deficits rather than logic. The fact that CoT degrades model performance for open-source artefacts is an interesting call out and calls for further investigation

**Weaknesses:**

- CoT Degradation Analysis can be fleshed out more :
   - This is a crucial and rather counter-intuitive observation made by the paper. It would be good to provide more information on the details of the framework for this specific study to establish trust on these results.
   - The ablation study is limited to accuracy comparisons with and without CoT, and the explanation for the degradation is largely speculative (e.g., CoT text may distract models not fine-tuned for long-form reasoning). There is a need for more in-depth, systematic root cause analysis—such as controlled experiments isolating prompt length, reasoning style, or model architecture effects.
   - The study does not explore whether prompt engineering, alternative CoT formats, or model-specific tuning could mitigate the observed degradation. Need to ablate on the CoT settings to ensure this is not a framework bias
   - The paper does not provide error breakdowns or qualitative examples specifically illustrating how CoT leads to errors in open-source models.

- Data Contamination solutions needs more validation : the idea of dynamically updating the test bank with randomized generations (while maintaining standardized templates for fair comparisons) is a sound hypothesis to mitigate data contamination issues, but quantification/testing to confirm the validity of this hypothesis is lacking
    - There is a need for basic adversarial or retrieval-based analysis to test if models are “memorizing” or “recalling” similar problems from pretraining.
    - The effectiveness of dynamic test bank updates is asserted, but not validated with concrete evidence or contamination metrics. There is no empirical study or audit quantifying the risk or presence of contamination in current models.

**Questions:**

As covered in the weaknesses, there is need for some validations on the claims and observations made

---

> ### Author Response · Authors · 2025-11-19
> **Q1 to Reviewer 9vwG**
>
> We are grateful for your suggestion to investigate the 'CoT Paradox' further. We performed a comprehensive **robustness analysis** on prompts and extraction rules (**Table 4, Section 4.2.3**) to rule out framework bias and substantiate our findings. Additionally, we addressed contamination concerns by proposing specific validation protocols and clarifying our proactive generation approach. Major updates are marked in blue.
>
> &nbsp;
>
> ## Q1 (from Weakness 1): Address the "CoT Paradox" by providing a robustness analysis (ablation on prompts/formats) to rule out framework bias.
>
> To move beyond speculation and establish this as a robust finding, we conducted a comprehensive ablation study and a qualitative failure analysis. These additions have been integrated into **Section 4.2.2** and **Section 4.2.3** of the revised paper.
>
> To ensure the observed degradation in open-source models was not an artifact of specific prompt engineering or extraction failures, we analyzed sensitivity across prompt variants and extraction rules (see **Table 4** in the revision).
>
> - **Sensitivity to Prompt Variations:** We compared our original template ("CoT A", inspired by DeepSeek-R1) against an alternative template ("CoT B", adapted from EMMA). As shown in the table below, the performance trends remained consistent. For example, **Qwen2.5-VL-72B** consistently underperformed compared to its non-CoT baseline (35.00%) regardless of the prompt used.
>
>   **Sensitivity to Prompt Variations (Accuracy %)**  in Table 4 (a)
>
>   | Model             | CoT A | CoT B |   Δ   |
>   | :---------------- | :---: | :---: | :---: |
>   | Qwen2.5-VL-72B    | 33.31 | 31.19 | -2.12 |
>   | GPT-4o            | 31.10 | 30.81 | -0.29 |
>   | Claude-3.5-sonnet | 32.54 | 28.31 | -4.23 |
>
>
> - **Sensitivity to Extraction Rules:** We compared our baseline **Extract Rule A** (truncated letter matching) against **Extract Rule B** (customized full-format regex matching). While Rule A is generic enough to cover diverse output types and Rule B targets specific patterns, the inherent flexibility of CoT responses means neither guarantees 100% extraction success. Crucially, the discrepancy between rules was negligible (<1.2%), and the negative impact of CoT persisted under both. This confirms that the "CoT Paradox" is a genuine reasoning failure, not a parsing artifact of Rule A.
>
>   **Sensitivity to Extraction Rules (Acc. Drop%)** in Table 4 (b)
>
>   | Model           | Rule A ↓ | Rule B ↓ |   Δ   |
>   | :-------------- | :------: | :------: | :---: |
>   | SAIL-VL-1.5-2B  |  -8.22   |  -7.29   | +0.93 |
>   | Deepseek-VL2-3B |  -5.18   |  -5.01   | +0.17 |
>   | Kimi-VL-16B     |  -8.47   |  -9.66   | -1.19 |

---

> ### Author Response · Authors · 2025-11-19
> **Q2 to Reviewer 9vwG**
>
> ## Q2 (from Weakness 1): Provide a deeper root cause analysis with qualitative examples of how CoT induces errors.
>
> ### **1. Statistical Root Cause Analysis**
>
> We performed paired significance tests ($p < 0.05$) to pinpoint *where* the degradation occurs. The analysis reveals that the negative impact for open-source models is not uniform but is **highly concentrated in "pure-visual" spatial tasks** (e.g., 3-View Projection, 3D Rotation).
>
> | **Model**            | **Source** | **CoT Impact** | **Significant (p<0.05)** |
> | -------------------- | ---------- | -------------- | ------------------------ |
> | Kimi-VL-A3B-Instruct | Open       | Negative       | Yes (p=0.0192)           |
> | Deepseek-VL2-tiny    | Open       | Negative       | Yes (p=0.0463)           |
> | InternVL2.5-78B      | Open       | Negative       | Yes (p=0.0368)           |
> | Claude-3.5-Sonnet    | Closed     | Positive       | Yes (p=0.0007)           |
>
> This statistical evidence supports the hypothesis that for models not specifically optimized for reasoning, the mandate to generate text interferes with the holistic processing required for pure spatial tasks.
>
> ### **2. Qualitative Failure Analysis (Why CoT fails)**
>
> To provide the requested qualitative evidence, we analyzed specific cases where models succeeded with Direct Answering but failed with CoT ("Non-CoT ✓ $\rightarrow$ CoT ×"). We identified two primary mechanisms by which CoT actively induces errors:
>
> | **Failure Mode**           | **Description**                                              | **Representative Case**                                      |
> | -------------------------- | ------------------------------------------------------------ | ------------------------------------------------------------ |
> | **Visual Hallucination**   | Textual reasoning contradicts visual input, effectively "overwriting" correct visual perception with an erroneous text chain. | In **Three-View Projection**, the CoT correctly identifies shapes initially but then "hallucinates" a mirror flip operation that did not visually occur, leading to an incorrect choice despite accurate initial recognition. |
> | **Enforced Serialization** | The model attempts to decompose a parallel process (e.g., folding) into linear text steps, introducing logical errors. | In **Cube Unfolding**, the CoT attempts to verify face adjacencies sequentially. It fails to track simultaneous topological changes, logically inferring that two adjacent faces are "opposite" due to the linear constraints of text generation. |
>
> These findings confirm that CoT degradation is not merely a distraction but a conflict between **enforced serialization** (text generation) and **holistic visual processing** in models that lack a strong internal "world model" for spatial simulation.

---

> ### Author Response · Authors · 2025-11-19
> **Q3 to Reviewer 9vwG**
>
> ## Q3 (from Weakness 2): How can the hypothesis that "dynamically updating the test bank" mitigates data contamination be quantitatively validated?
>
> We confidently propose two specific quantitative experiments as validation protocol to empirically verify this capability:
>
> ### **1. Controlled Contamination Experiment (Immediate Quantification)**
>
> This method simulates contamination to measure the benchmark's resistance.
>
> - **Method:** Using our generation pipeline, create two parallel datasets: **Set A** (Public/Contamination Source) and **Set B** (Private/Dynamic).
> - **Simulation:** Select a baseline model and fine-tune it on Set A to explicitly induce data memorization.
> - **Verification Metric:**
>   - The model will show artificially inflated scores on Set A.
>   - However, its score on the unseen Set B will remain low, reflecting its true generalization capability.
> - **Proof:** The significant performance gap between Set A and Set B provides quantitative evidence of the dynamic bank's ability to resist contamination and isolate true reasoning skills.
>
> ### **2. Longitudinal Tracking Experiment (Long-term Validation)**
>
> This leverages the benchmark's scalability over its lifecycle.
>
> - **Method:** Publicly release the current **v1.0** (1,180 problems) while maintaining an internal, dynamically updated **v1.1** (Private Bank).
> - **Verification Metric:** As future models (potentially trained on v1.0 data) are released, evaluate them on both versions.
> - **Proof:**
>   - Scores on v1.0 are expected to inflate rapidly due to data leakage.
>   - Scores on v1.1 will show a smoother curve, accurately reflecting actual progress in spatial reasoning.
>   - This divergence validates the necessity of dynamic updates for ensuring continuously reliable evaluations.
>
> These experiments demonstrate that our programmatic framework is not just a hypothesis but an actionable tool for maintaining evaluation integrity.

---

> ### Author Response · Authors · 2025-11-19
> **Q4 to Reviewer 9vwG**
>
> ## Q4 (from Weakness 2): Did you conduct adversarial or retrieval-based analysis to test for "memorization" versus "reasoning" and quantify contamination risks?
>
> We did not perform an explicit post-hoc retrieval analysis because our methodology—**Programmatic Generation**—is designed to *proactively* preclude contamination at the source, rendering reactive auditing unnecessary.
>
> - **Proactive Avoidance vs. Reactive Detection:** Traditional benchmarks often rely on public sources (e.g., IQ tests), making them vulnerable to data leakage. In contrast, SpatialViz-Bench comprises **1,180 entirely novel** problems generated from scratch using code, ensuring they fundamentally do not exist in current pre-training corpora.
> - **Performance as Validation:** The low model performance itself serves as strong evidence against memorization. If contamination were present, we would expect inflated scores. Instead, even the top-performing model (Gemini-2.5-pro) achieves only **44.66%**, with near-random accuracy on core 3D tasks. This confirms that models are being forced to reason (often unsuccessfully) rather than recall.
> - **Future-Proofing:** Our framework’s ability to dynamically update the test bank ensures continued resistance to contamination in future training iterations.
>
> Thus, the results on SpatialViz-Bench accurately reflect genuine deficits in spatial reasoning rather than failures of memory.

---

### Official Review · Reviewer_GoVB · 2025-10-31

**Soundness:** 3
**Presentation:** 3
**Contribution:** 3
**Rating:** 6
**Confidence:** 4

**Summary:**

This paper introduces SpatialViz-Bench, a new benchmark designed to evaluate the critical but under-tested skill of spatial visualization in Multi-modal Large Language Models (MLLMs). The authors argue that existing benchmarks fail to adequately assess an MLLM's ability to mentally manipulate unseen objects (e.g., mentally folding a net into a cube) and often rely on publicly sourced problems, risking data contamination.
To address this, SpatialViz-Bench offers a comprehensive suite of 1,180 problems across 12 tasks, which are programmatically generated to ensure fairness, scalability, and resistance to data leakage. The tasks are grounded in cognitive science, covering four core abilities: mental rotation, mental folding, visual penetration, and mental animation. Through an extensive evaluation of 27 MLLMs, the paper reveals several key findings: 1. A significant performance gap exists between all current models, including state-of-the-art systems like Gemini-2.5-pro, and a human baseline, highlighting the difficulty of the tasks. 2. In a counter-intuitive discovery, Chain-of-Thought (CoT) prompting paradoxically degrades the accuracy of many open-source models. 3. Error analysis indicates that model failures stem primarily from fundamental deficits in visual perception and spatial transformation, rather than high-level logical reasoning.
Ultimately, SpatialViz-Bench provides a robust, diagnostic tool for the research community, identifying a crucial weakness in modern MLLMs and guiding future efforts toward improving their core visuospatial intelligence.

**Strengths:**

### 1. Addresses a Critical and Under-Evaluated Research Gap in MLLMs
The paper's strength is its focus on **spatial visualization**, a specific and advanced cognitive skill that has been largely overlooked by existing MLLM benchmarks. While many benchmarks test reasoning on *visible* information (e.g., "What color is the car on the left?"), SpatialViz-Bench is one of the first to systematically evaluate the ability to reason about *unseen* spatial relationships through mental manipulation. This is a crucial capability for real-world applications in fields like engineering (CAD models), medicine (interpreting 3D scans), and robotics (planning object manipulation), making this benchmark both timely and highly relevant.


### 2. Diagnostic Analysis Beyond Simple Accuracy Metrics
The statistical error analysis (Figure 4) and the qualitative case studies (Figure 5) provide deeper insights into the models' failure modes. The conclusion that **"Perceptual and Spatial Transformation errors dominate failures"** is an important finding. It suggests that the primary bottleneck for MLLMs is not high-level logic or instruction following, but rather fundamental deficits in visual perception and mental manipulation. This provides a clear and actionable direction for future research and model development.




### 3. Methodologically Robust and Trustworthy Benchmark Construction
The authors have designed a benchmark that directly confronts the most pressing challenges in modern AI evaluation, particularly data contamination and reliability. The core strength here is the **programmatic generation** of 11 out of the 12 tasks. This approach provides several key advantages:
*   **Mitigates Data Contamination:** By generating novel, synthetic problems, the benchmark ensures that models are tested on their reasoning abilities rather than their ability to recall answers to similar problems scraped from the internet (e.g., from public IQ tests or math competitions).
*   **Ensures Scalability and Longevity:** The programmatic framework allows for the continuous generation of new test cases. This creates a "dynamic test bank" that can be updated over time, making the benchmark a sustainable and long-lasting resource that is resilient to future training data leakage.
*   **Enables Controlled Difficulty and Systematic Analysis:** The generation process allows for precise control over task difficulty (e.g., by increasing the number of folds or the complexity of an assembly). It also enables the systematic creation of distractor options based on common error types, which makes the evaluation more diagnostic.

**Weaknesses:**

**1. The "Cognitively-Grounded" Claim is Overstated**

The paper's implementation of its central claim to be "cognitively-grounded" is shallow. True cognitive grounding goes beyond simply adopting high-level task categories from historical psychology literature; it requires integrating principles of human cognition into task design, particularly concerning difficulty and error analysis.

*   **Absence of Parametric Difficulty Scaling:** The paper's approach to scaling difficulty, as outlined in **Table 1 (lines 218-236)**, relies on intuitive and qualitative heuristics such as "**Larger assemblies,**" "**More folds,**" or "**More system modules.**" This approach fails to incorporate research on spatial reasoning. For example, a deeply grounded mental rotation task would parametrically vary the angle of rotation, as human reaction time and error rates are known to increase linearly with this variable. Similarly, a paper folding task's difficulty could be systematically controlled by the number of folds or the complexity of the resulting hole symmetry. By not grounding its difficulty levels in these quantifiable cognitive-load metrics, the benchmark cannot guarantee that "Level 1" is consistently and measurably more difficult than "Level 0" in a way that reflects cognitive processes.

*   **Implication:** This lack of parametric control weakens the benchmark's diagnostic power. We cannot be certain that a model's performance drop from Level 0 to Level 1 is due to a failure in a specific cognitive step or simply an inability to handle more visual "clutter." The benchmark measures performance on harder problems but doesn't provide the fine-grained control needed to diagnose *why* they are harder from a cognitive standpoint.



**2. Issue with manually designed "Mechanical System" task**

One of the paper's arguments is its use of programmatic generation to create a scalable, dynamic, and contamination-resistant benchmark.  However, this rigor is abandoned for one of the 12 tasks, creating inconsistency.

*   **Contradiction of Core Philosophy:** The authors state in **Section 3.3 (lines 265-267)** that the "**Mechanical System task was manually designed**" using "**representative public simulations.**" This contradicts their own critique of other benchmarks that "**rely heavily on web-sourced problems, risking data leakage**" (**line 096**). This task is now vulnerable to the exact problems the rest of the benchmark was designed to solve. The "public simulations" may very well be part of the pre-training corpora of the models being evaluated, turning a test of reasoning into a test of memorization.

*   **Implication:** This inconsistency compromises the internal validity of the benchmark's results. Performance on the Mechanical System task cannot be reliably compared to the other 11 tasks. As seen in **Table 2 (line 328)**, many open-source models score surprisingly high on this task relative to their poor performance on other 3D tasks like Cube Unfolding or 3D Rotation. Is this because they possess a distinct "intuitive physics" capability, as the authors suggest (**lines 2534-2537**), or is it because they have been exposed to the source material during training? The manual design of this task makes it difficult to disentangle these possibilities, undermining the clarity of the evaluation.




**3. The "Chain-of-Thought Paradox"**

The paper's finding that CoT prompting degrades performance on open-source models is a thought provoking conclusion. However, the analysis supporting this claim is not robust enough to rule out simpler, confounding explanations.

*   **The Flaw of a One-Size-Fits-All Prompt:** The entire CoT analysis rests on a single, standardized prompt template applied to all models, detailed in **Appendix E.2 (lines 2347-2351)**. This is a methodological flaw. Different MLLMs are instruction-tuned on vast and varied datasets, leading to high sensitivity to prompt phrasing, structure, and keywords. A prompt that works well for one model family may be confusing or suboptimal for another. The observed performance degradation may not reflect a fundamental cognitive "interference" but rather a failure of prompt engineering—the models may simply be failing to adhere to an unfamiliar instruction format. A robust study should  include a prompt sensitivity analysis, testing several CoT variations to ensure the effect was not an artifact of their chosen template.

*   **Selection Bias in the Comparison Group:** The paper introduces a selection bias that skews the comparison. The authors explicitly state they "**excluded models... unable to adhere to the format (e.g., InternVL3-2B)**" from the non-CoT evaluation (**lines 368-370**). This means the group of open-source models in the non-CoT test is a pre-filtered subset of the most capable or compliant models. The weaker models that fail even simple formatting are removed. This inflates the baseline non-CoT performance for the open-source category, making the subsequent drop when CoT is enabled appear larger than it may actually be.

*   **Speculative Explanation:** The authors hypothesize that CoT "**interferes with their intrinsic visual judgment**" (**line 395**). The paper provides no qualitative analysis of the generated CoT content itself. A rigorous analysis would involve manually inspecting the reasoning steps to see *how* they fail. Do the models make perceptual errors in their text (e.g., "the orange face is next to the green face" when it is not)? Or are the errors logical, showing a flawed transformation process? The error analysis in **Figure 4 (line 390)** categorizes final outcomes but does not correlate specific error types with the use (or non-use) of CoT. Without this evidence, alternative hypotheses—such as the models simply generating irrelevant, distracting text—are equally plausible.



**4. A Foundational Claim Lacking Empirical Verification: "Vision-Dependence"**

The paper rightly positions its benchmark as highly "vision-dependent," stating that "**textual input alone is insufficient**" for problem-solving (**line 309**). While intuitively true, this is a foundational, empirical claim that is asserted rather than proven. The necessity of the visual input could have been easily and powerfully demonstrated with a simple control experiment. By feeding the text-only components of the prompts to a powerful text-only LLM (like GPT-4) and recording its performance, the authors could have provided definitive quantitative evidence for their claim. A resulting accuracy at or near the random-chance baseline (**25.08%**, noted in **Table 2**) would have formally established the visual modality as indispensable for this benchmark, thereby strengthening the paper's core motivation and contribution.

**Questions:**

*   **The CoT Prompt:** The paper makes a strong claim about CoT, but the specifics of the prompt used are relegated to an appendix.
    *   Can you give examples of the CoT prompt?
    *   Why was a single, standardized prompt chosen for all models? It is well-known that prompt effectiveness is highly model-dependent. The observed negative effect might be an artifact of a suboptimal prompt for certain models, not an inherent issue with CoT itself.


*   **Use of External Datasets:** For the "Three-View Projection" task, Table 1 mentions "Real engineering parts (DeepCAD)." It's unclear if DeepCAD is another procedurally generated source or a fixed dataset. If it's a fixed dataset, how does this align with the goal of creating a dynamically updatable benchmark?

---

> ### Author Response · Authors · 2025-11-19
> **Q1 to Reviewer GoVB**
>
> Thank you for highlighting the "CoT Paradox" and "Vision-Dependence". We addressed these by conducting a **text-only control experiment** to confirm visual dependency (**Table 2**, **Section 4.2.1**)  and sensitivity analyses on prompt variants (**Table 4, Section 4.2.3**). We have modified **Section 3.3** for concerns of the use of  DeepCAD dataset, reinforcing the benchmark's cognitive grounding and clarifying safeguards for the Mechanical System task. Please refer to **Appendix B** for more detailes of task construction. Changes are highlighted in blue.
>
> &nbsp;
>
> ## Q1: Provide examples of the CoT prompts used, and explain why was a single standardized prompt chosen for all models instead of accounting for model-specific sensitivities.
>
> We have provided the exact, verbatim CoT prompt template, along with the "Direct Answer" (non-CoT) template used for comparison, in Appendix E.2.
>
> - **Standardization for Fair Comparison:** Our core objective was to ensure fairness, controllability, and scalability in our evaluation. Individually tuning prompts for all 27 models would introduce a massive confounding variable, rendering cross-model comparisons meaningless.
>
> - **Assessing Intrinsic Capability:** By using a unified, standardized prompt, we aim to assess a model's intrinsic spatial capability rather than our prompt engineering skills. We believe that a model's ability to effectively execute a standard CoT instruction without specific tuning is itself a key capability being evaluated.
>
> - **Sensitive Study:**
>
> Our original submission used a CoT template (“CoT A”) inspired by DeepSeek-R1:
>
> ```
> You should first provide a reasoning process, then provide a single option (A, B, C or D) as the final answer. The reasoning process and the answer are enclosed within <think></think> and <answer></answer> tags, respectively, i.e., <think>reasoning process</think>, <answer>answer</answer>.
> ```
>
> To assess prompt sensitivity, we evaluated an additional CoT template (“CoT B”) adapted from EMMA [1]:
>
>
> ```
> Answer with the option’s letter from the given choices and put the letter in one '\\boxed{}'. Please solve the problem step by step.
> ```
>
> **Sensitivity to Prompt Variations (Accuracy %)**  in Table 4 (a)
>
> | Model             | CoT A | CoT B |   Δ   |
> | :---------------- | :---: | :---: | :---: |
> | Qwen2.5-VL-72B    | 33.31 | 31.19 | -2.12 |
> | GPT-4o            | 31.10 | 30.81 | -0.29 |
> | Claude-3.5-sonnet | 32.54 | 28.31 | -4.23 |
>
> The results show that our main findings are not highly sensitive to minor prompt changes, with the overall accuracy difference being minimal (<= 5%) (detailed in Section 4.2.3).

---

> ### Author Response · Authors · 2025-11-19
> **Q2 to Reviewer GoVB**
>
> ## Q2: Is the DeepCAD dataset mentioned in the "Three-View Projection" task programmatically generated or a fixed dataset?
>
> We clarify that DeepCAD itself is a fixed dataset. However, utilizing it does not compromise the "programmatic generation" or "contamination-resistant" nature of our benchmark for the following reasons:
>
> - **Massive Scale:** DeepCAD [1] contains 178,238 models, providing an extremely rich source of raw material.
> - **Programmatic Problem Generation:** As detailed in **Appendix B.1**, our core innovation lies in the *problem generation process*, not the raw model creation. We randomly sample CAD models and then employ programmatic methods to generate novel distractors.
> - **Dynamic Distractor Mechanism:** The specific generation algorithm is detailed in Algorithm 5 of Appendix B.4. We use image processing algorithms to extract internal lines and then programmatically generate deceptive negative samples through random line deletion, view rotation, or view flipping. Consequently, while the source models are fixed, every evaluation question—and crucially, its distractors—is dynamically generated via random algorithms, ensuring novelty and scalability.
>
> > [1] Wu, Rundi, Chang Xiao, and Changxi Zheng. "Deepcad: A deep generative network for computer-aided design models." *Proceedings of the IEEE/CVF* *International Conference on Computer Vision*. 2021.

---

> ### Author Response · Authors · 2025-11-19
> **Q3 to Reviewer GoVB**
>
> ## Q3 (from Weakness 1): Respond to the criticism that the "cognitively-grounded" claim is overstated and that difficulty scaling relies on intuition rather than quantifiable cognitive load.
>
> We acknowledge that the high-level descriptions in Table 1 (e.g., "More folds") might have appeared heuristic due to space constraints. However, we clarify that our difficulty design is fully quantitative and parametrically controlled:
>
> - **Quantitative Definition:** As detailed in **Appendix B.1 (Task Construction)** and the pseudocode in **Appendix B.4**, every difficulty level (L0, L1, L2) corresponds to strict quantitative parameters. For example:
>   - **Paper Folding:** Controlled by grid size (L0: 3x3, L1: 4x4, L2: 5x5), fold count (L0/L1: 2, L2: 3), and punch count (L0: 1, L2: 2, L3: 3).
>   - **Cube Counting:** Defined by view count (L0: 2, L1/L2: 3) and component scale.
>   - **Arrow Moving:** Defined by step count and interaction rules (single vs. multi-arrow).
>   - **Cube Tasks:** Controlled by pattern complexity (e.g., symmetry) and the operations required for negative sample generation (e.g., internal rotation/flipping).
> - **Programmatic Implementation:** Since 11/12 tasks are programmatically generated, these parameters are hard-coded into our generation algorithms (see **Appendix B.4**), ensuring difficulty is systematic and reproducible, not subjective.
> - **Cognitive Validity:** We validated these gradients against human performance. As shown in **Appendix F.1 (Tables 7-10)**, human accuracy shows a clear negative correlation with our difficulty levels (e.g., Paper Folding accuracy drops from 100% at L0 to 87.5% at L2), confirming our quantitative metrics align with human cognitive load.

---

> ### Author Response · Authors · 2025-11-19
> **Q4 to  Reviewer GoVB**
>
> ## Q4 (from Weakness 2): Address the contradiction of using a manually designed "Mechanical System" task with "public simulations" against the "anti-contamination" philosophy, and explain how to rule out data leakage.
>
> We clarify that programmatically generating random, physically consistent mechanical systems is technically prohibitive, requiring expert physics engine development. Conversely, open-source simulation animations are abundant. To ensure comprehensiveness, we chose this pragmatic approach, but with strict safeguards against contamination:
>
> - **Novel Question Design:** As stated in **Section 3.3 and Appendix B.3**, all evaluation questions were newly designed by human experts with mechanical engineering backgrounds *after* viewing the simulations.
> - **Rigorous interrogation:** To ensure rigor and comprehensiveness, we generate multiple distinct questions (e.g., testing both clockwise and counter-clockwise motion scenarios) from a single reference image. This ensures the textual questions are novel and have never appeared in training sets.
> - **Necessity of the Task:** We included this task because "Mechanical Intelligence"—visualizing object relationships and physical operations—is a foundational component of spatial ability research [2]. Excluding "Mental Animation" of mechanical dynamics would render our spatial visualization assessment incomplete.
>
> > [2] Thorndike, Edward L. "On the Organization of Intellect." *Psychological Review* 28.2 (1921): 141.

---

> ### Author Response · Authors · 2025-11-19
> **Q5 to Reviewer GoVB**
>
> ## Q5. a (from Weakness 3.a): Address the methodological flaw of selection bias in the non-CoT analysis by excluding "non-compliant" models.
>
> Our exclusion criteria for the Direct Answer (non-CoT) evaluation were based on two clear standards to ensuring a valid baseline, not to inflate performance (illustrated in Appendix E.6.1):
>
> - **Reasoning-Centric Architectures:** Models explicitly designed for extended reasoning (e.g., o1, Gemini-2.5, Kimi-thinking, Llama-4 series) were excluded, as inhibiting CoT contradicts their core design principles.
>
> - **Instruction Adherence:** Models unable to suppress reasoning traces despite strict formatting prompts (specifically InternVL3-2B) were excluded. This failure reflects a limitation in instruction following rather than reasoning capability.
>
> Therefore, our non-CoT evaluation retains only models capable of reliably providing a single-letter answer. This ensures the comparison is methodologically sound by testing models on a task format they can actually perform. The lower number of closed-source models in this comparison simply reflects that most top-tier closed-source models are now reasoning-centric by design.
>
> &nbsp;
>
> ## Q5. b (from Weakness 3.b): Qualitative analysis of the generated CoT content to support the speculative hypothesis that CoT "interferes with intrinsic visual judgment".
>
> To substantiate our hypothesis, we conducted a focused qualitative analysis of specific failure cases where models succeeded with Direct Answering but failed with CoT (i.e., "Non-CoT ✓→ CoT ×").
>
> This analysis, summarized in the table below, provide evidence that CoT does not merely distract, but actively induces errors by forcing serialized, analytical reasoning onto tasks that require holistic visuospatial judgment.
>
> | **Failure Mode**           | **Description**                                              | **Representative Case**                                      |
> | -------------------------- | ------------------------------------------------------------ | ------------------------------------------------------------ |
> | **Visual Hallucination**   | Textual details contradict visual input, effectively "overwriting" correct visual perception with an erroneous text chain. | In **Three-View Projection**, the CoT text correctly identifies the initial shapes but subsequently hallucinates a "mirror flip" operation that simply did not occur in the visual data, leading to an incorrect final choice despite accurate initial recognition. |
> | **Enforced Serialization** | The model attempts to decompose a holistic, parallel process (e.g., mental folding) into linear steps, introducing logical errors during sequential verification. | In **Cube Unfolding**, the CoT attempts to verify face adjacencies sequentially text-by-text. However, it fails to track simultaneous topological changes, logically inferring that two actually adjacent faces are "opposite" due to the linear constraints of the text generation. |
>
> This qualitative evidence directly supports our statistical findings: the enforced serialization of CoT conflicts with the parallel nature of holistic visual processing, actively generating reasoning artifacts that derail correct perceptual judgments.

---

> ### Author Response · Authors · 2025-11-19
> **Q6 to Reviewer GoVB**
>
> ## Q6 (from Weakness 4): Verify the claim of "vision-dependence" by conducting a control experiment with a text-only LLM.
>
> We appreciate this insightful suggestion to strengthen our foundational claim. In response, we conducted a control experiment using **Qwen2.5-72B-Instruct** (a powerful text-only LLM) on the text-only components of our prompts.
>
> ### 1. Quantitative Verification
>
> As hypothesized, the text-only model achieved an overall accuracy of 25.86%, which is negligibly different from the random-chance baseline of 25.08%. This statistically confirms that the visual modality is indispensable for solving SpatialViz-Bench tasks. We have integrated these results into Section 4.2.1 and Table 2 of the revised paper.
>
> | **Model**                            | **Overall** | **Mental Rotation** |         |         |         | **Mental Folding** |        |        |         | **Visual Penetration** |        |        |         | **Mental Animation** |        |        |         |
> | ------------------------------------ | ----------- | ------------------- | ------- | ------- | ------- | ------------------ | ------ | ------ | ------- | ---------------------- | ------ | ------ | ------- | -------------------- | ------ | ------ | ------- |
> |                                      |             | **2DR**             | **3DR** | **3VP** | **Avg** | **CU**             | **PF** | **CR** | **Avg** | **CS**                 | **CC** | **CA** | **Avg** | **AM**               | **BM** | **MS** | **Avg** |
> | **Qwen2.5-72B-Instruct (Text-only)** | **25.86**   | 15.00               | 35.00   | 15.00   | 21.67   | 16.67              | 23.33  | 26.67  | 22.22   | 20.00                  | 33.33  | 45.00  | 31.25   | 25.00                | 30.00  | 30.00  | 28.33   |
>
> ### 2. Qualitative Analysis of Text-Only Failures
>
> Our analysis of the model's outputs reveals three primary failure modes caused by the lack of visual grounding:
>
> - **Type 1: Logical but Groundless Analysis.** The model attempts to reason logically about the *options* but lacks the visual "ground truth" (e.g., the initial grid state) to verify them, leading to random guessing.
>
>   > *Problem Statement:* "The left image shows a colored grid with a red square marking one corner. Which grid can be obtained by rotating the left grid only, without flipping or other changes?"
>   >
>   > *Model Output:* "**Option A**: Check if it matches... **Option B**: Check if it matches... If none of the options match, then the correct answer is D."
>
>   **Type 2: Visual Hallucination.** The model invents visual details (e.g., initial orientation) that are critical for the solution but absent in the text description.
>
>   > *Problem Statement:* "In the diagram, the red arrow is the initial arrow, and the green arrow is the final arrow. The arrow can move in four directions... Which of the following paths can make the arrow move from the starting position to the ending position?"
>   >
>   > *Model Output:* "The red arrow (initial) points upwards." (When the text did not specify the initial orientation).
>
>   **Type 3: Generic Theoretical Assumptions.** The model relies on general knowledge (e.g., physics principles) rather than the specific system configuration, which is impossible to solve without seeing the connections.
>
>   > *Problem Statement:* "In the current setup, when the blue shaft rotates clockwise, what is the resulting motion of the green object?"
>   >
>   > *Model Output:* "Given the typical setup... the most likely scenario is that the green object will rotate in the opposite direction..."
>
> These findings confirm that SpatialViz-Bench effectively isolates spatial visualization skills that cannot be solved via textual shortcuts or pattern matching.

---

### Official Review · Reviewer_XXzx · 2025-11-01

**Soundness:** 3
**Presentation:** 3
**Contribution:** 3
**Rating:** 6
**Confidence:** 3

**Summary:**

This paper introduces SpatialViz-Bench, a benchmark that isolates and evaluates spatial visualization in multimodal LLMs across 4 sub-abilities mental rotation, mental folding, visual penetration and mental animation. This is introduced using 12 tasks and 1,180 programmatically generated problems with controllable difficulty. Eleven tasks are created with a Python + FreeCAD pipeline to enable scalable expansion and minimize contamination; one task (mechanical systems) is manually curated. The authors evaluate 27 MLLMs (8 closed-source, 19 open-source) in zero-shot settings with both Direct Answer and CoT prompting. Results show a large gap to humans and reveal an interesting finding that CoT hurts many open-source models, widening the closed vs. open-source gap. The diagnostic error analysis finds failures dominated by Perceptual and Spatial Transformation errors, suggesting limits in visuo-spatial processing rather than high-level logic. The benchmark aims to provide a stable, expandable, and contamination-resistant testbed to improve progress on visuo-spatial capabilties in MLLMs.

**Strengths:**

* Centers spatial visualization (not just perception or generic “spatial reasoning”) and grounds it in cognitive science, decomposing ability into 4 sub-abilities with 12 targeted tasks.
* Programmatic generation (Python + FreeCAD) for 11/12 tasks yields controllable difficulty, systematic distractors, and an expandable test bank to mitigate contamination; standardized templates reduce instruction confounds.
* Broad coverage (rotation, folding, penetration, animation) plus error taxonomy (perceptual, transformation, memorization, instruction following, methodological, calculation/reasoning) surfaces where models fail rather than just how much.
* Clear hierarchy among models with large human–model gap and near-random performance on core 3D tasks for many models; CoT ablation highlights that enforced reasoning text can degrade open-source model accuracy.

**Weaknesses:**

* Lacks uncertainty estimates (CIs/SEs) and significance testing across models/tasks.
* The “CoT hurts” result may be prompt and format-sensitive. More prompts, temperature sweeps, and output-length controls are needed to ensure the effect isn’t an artifact of prompt adherence or extraction errors.
* Pure multiple-choice framing may enable elimination strategies viz. no free-form diagram/pose prediction, 3D pose estimation, or procedural rollouts that might better stress genuine mental simulation.
* There is limited evidence that gains on SpatialViz-Bench transfer to robotics, CAD, surgical planning, or navigation
* The error analysis is “primarily manual” with model assistance; inter-annotator agreement (e.g., Cohen’s κ) is not reported.

**Questions:**

1. How sensitive are CoT results to prompt variants, answer extraction rules, and output-length limits? Reporting accuracy across a prompt suite and per-model variance bars would provide more insights.
2. Is possible to also publish 95% CIs per task/model and paired significance tests across prompts (CoT vs. Direct), and across difficulty levels to substantiate the key conclusions.
3. Do improvements on SpatialViz-Bench correlate with performance on downstream visuospatial tasks or questions, VQA on CAD/DeepCAD?
4. Was there any ablations on image resolution, number of views, and rendering style (lighting, line thickness) to determine sensitivity to low-level vision vs. spatial reasoning.
5. Was there an analysis on controlled difficulty and its relation to performance of the models to identify where does it break?
6. For the six error categories, is it possible to report inter-annotator agreement (κ)?

---

> ### Author Response · Authors · 2025-11-19
> **Q1 to Reviewer XXzx**
>
> We appreciate your constructive feedback on statistical rigor. We have strengthened our analysis by adding **95% Confidence Intervals** and **paired significance tests** to substantiate our findings (**Figure 4**, **Table 3**). We also conducted sensitivity analyses on prompt variants and extraction rules (**Table 4, Section 4.2.3**), and reported inter-annotator agreement ($\kappa$) in **Section 4.3.1, Appendix E.6.3**. All revisions are marked in blue.
>
> &nbsp;
>
> ## Q1: How sensitive are the CoT results to prompt variants, answer extraction rules, and output-length limits?
>
> We analyzed the sensitivity of our Chain-of-Thought (CoT) results across three aspects: (1) prompt variants, (2) answer extraction rules, and (3) output-length limits.
>
> ### **1. Prompt Variants**
>
> Our original submission used a CoT template (“CoT A”) inspired by DeepSeek-R1 [1]:
>
> ```
> You should first provide a reasoning process, then provide a single option (A, B, C or D) as the final answer. The reasoning process and the answer are enclosed within <think></think> and <answer></answer> tags, respectively, i.e., <think>reasoning process</think>, <answer>answer</answer>.\n
> ```
>
> To assess prompt sensitivity, we evaluated an additional CoT template (“CoT B”) adapted from EMMA [2]:
>
> ```
> Answer with the option’s letter from the given choices and put the letter in one '\\boxed{}'. Please solve the problem step by step.\n
> ```
>
> **Sensitivity to Prompt Variations (Accuracy %)**  in Table 4 (a)
>
> | Model             | CoT A | CoT B |   Δ   |
> | :---------------- | :---: | :---: | :---: |
> | Qwen2.5-VL-72B    | 33.31 | 31.19 | -2.12 |
> | GPT-4o            | 31.10 | 30.81 | -0.29 |
> | Claude-3.5-sonnet | 32.54 | 28.31 | -4.23 |
>
> The results show that our main findings are not highly sensitive to minor prompt changes, with the overall accuracy difference being minimal (<= 5%) (detailed in Section 4.2.3).
>
> ### **2. Answer Extraction Rules**
>
> - Our original **Extract Rule A** inspired by MME-CoT [3] (Appendix E.4) employs a truncated letter matching approach designed to broadly cover diverse output formats.  Incorrect Matches:
>
>   `<answer>All three other options are incorrect</answer>` would be misread as 'A'.
>
>   `<answer>(Right, 1 unit)--(Backward, 2 units)--(Backward, 2 units)</answer>` would be misread as B.
>
> - **Extract Rule B** utilizes a customized, full-format regex strategy, using `\b([A-D])\b` to match standalone letters and then iterating through a prioritized list of patterns. We re-evaluated all models. For **14 of the 27 models, the results were identical**. For the other 13, the differences were minor, confirming our original logic was already highly accurate.
>
> Given the inherent flexibility of CoT responses, neither method guarantees 100% extraction success.
>
> To directly address whether the "CoT Paradox" was an artifact of the faulty rule, we performed a targeted analysis on the most-affected open-source models. The table below shows the change in CoT performance (vs. Direct Answer) using the old rule (Extract Rule A) versus the new rule (Extract Rule B).
>
> **Sensitivity to Extraction Rules (Acc. Drop%)** in Table 4 (b)
>
> | Model           | Rule A ↓ | Rule B ↓ |   Δ   |
> | :-------------- | :------: | :------: | :---: |
> | SAIL-VL-1.5-2B  |  -8.22   |  -7.29   | +0.93 |
> | Deepseek-VL2-3B |  -5.18   |  -5.01   | +0.17 |
> | Kimi-VL-16B     |  -8.47   |  -9.66   | -1.19 |
>
> This analysis confirms our key finding is robust. The "CoT Paradox" persists, and in the case of Kimi-VL, the negative impact is even *stronger* with the more accurate extraction rule. This strengthens our conclusion that for certain models, CoT interferes with intrinsic visuo-spatial judgment.
>
> ### **3. Output-Length Limits**
>
> We configured the maximum output token limit to the model's context capacity (or a sufficiently high upper bound). An analysis of our logs confirms that **nearly 100% of responses concluded naturally via the End-of-Sequence (EOS) token**, rather than being truncated by length limits. This ensures that the observed reasoning failures are due to model capability, not insufficient generation space. We believe this is the fairest comparison method. A sensitivity analysis on reduced length was not performed, as this would predictably and artificially degrade accuracy by cutting off a model's reasoning or final answer, rather than providing new insight into the CoT effect itself.
>
> > [1] Guo, Daya, et al. "Deepseek-r1: Incentivizing reasoning capability in llms via reinforcement learning." *arXiv preprint arXiv:2501.12948* (2025).
> >
> > [2] Hao, Yunzhuo, et al. "Can mllms reason in multimodality? emma: An enhanced multimodal reasoning benchmark." *arXiv preprint arXiv:2501.05444* (2025).
> >
> > [3] Jiang, Dongzhi, et al. "Mme-cot: Benchmarking chain-of-thought in large multimodal models for reasoning quality, robustness, and efficiency." *arXiv preprint arXiv:2502.09621* (2025).

---

> ### Author Response · Authors · 2025-11-19
> **Q2 to Reviewer XXzx**
>
> ## **Q2: Publish 95% CIs per task/model and paired significance tests across prompts (CoT vs. Direct), and across difficulty levels to substantiate the key conclusions.**
>
> We have conducted the requested statistical analyses, calculating 95% Wilson confidence intervals (CIs) and performing paired significance tests. These additions provide robust statistical backing for our core conclusions and have been integrated into the revised paper (see Figure 4 and Table 3).
>
> ### 1. 95% Confidence Intervals per Model (Section 4.2.1)
>
> The calculated CIs (illustrated in **Figure 4a**) yield three critical insights:
>
> - **Validation of Performance:** The CI for the top model, Gemini-2.5-pro ([41.85%, 47.51%]), does not overlap with the random baseline ([22.69%, 27.64%]), statistically confirming that top-tier models perform significantly better than random guessing1.
> - **Statistical Support for Capability Gaps:** The analysis confirms a significant gap between proprietary and open-source models. The CI for Gemini-2.5-pro has no overlap with the top open-source model, Llama-4-Scout ([31.58%, 36.99%])2.
> - **Identification of Performance Tiers:** The CIs allow us to group models into statistically indistinguishable tiers. For instance, the CIs for Llama-4-Scout and Qwen2.5-VL-72B highly overlap, placing them in the same performance tier3.
>
> ### 2. Statistical Significance of the "CoT Paradox" (Section 4.2.2)
>
> We performed paired significance tests ($p < 0.05$) comparing CoT and Direct prompting (see **Table 3**). The results confirm that the "CoT Paradox" is statistically significant. While CoT significantly aids Claude-3.5-Sonnet, it significantly degrades performance for several open-source models.
>
> | **Model**            | **Source** | **CoT Impact** | **Significant (p<0.05)** |
> | -------------------- | ---------- | -------------- | ------------------------ |
> | Kimi-VL-A3B-Instruct | Open       | Negative       | Yes (p=0.0192)           |
> | Deepseek-VL2-tiny    | Open       | Negative       | Yes (p=0.0463)           |
> | InternVL2.5-78B      | Open       | Negative       | Yes (p=0.0368)           |
> | Claude-3.5-Sonnet    | Closed     | Positive       | Yes (p=0.0007)           |
>
> Crucially, this degradation is concentrated in "pure-visual" spatial tasks (e.g., 3D Rotation, 3-View Projection), supporting the hypothesis that enforced text generation interferes with visual-spatial judgment in these models5.
>
> ### 3. Difficulty Level Sensitivity Analysis (Section 4.2.1)
>
> We analyzed model sensitivity to difficulty gradients (DG) to determine where capabilities break, utilizing the data presented in Figure 4b and Figure 4c of the revised paper:
>
> - **Performance Floor:** The analysis reveals a widespread "performance floor" at L0. **10 models** showed $\le 1$ significant change across difficulty levels, indicating they struggle even at the baseline difficulty. In contrast, the top-performing **Gemini-2.5-pro** was the most sensitive, exhibiting **7 significant changes**, reflecting its ability to solve L0 tasks while statistically revealing the difficulty spike at L1/L2.
>
> - **Task-Specific Insights:**
>
>   - **Cube Reconstruction vs. Unfolding:** A stark contrast was observed between **Cube Reconstruction** (12 models showed significant sensitivity) and its symmetric counterpart **Cube Unfolding** (only 1 model showed significant sensitivity). This suggests models are significantly better at reasoning about symmetry from unfolded views than reconstructing them.
>
>   - **Block Moving:** This task proved universally difficult; **0 models** showed a significant difficulty gradient. The task is challenging at both levels, rendering any performance drop statistically invisible.
>
>   - **3D Rotation:** Crucially, only **2 models** (the top performers, Gemini-2.5-pro and o1) exhibited a significant difficulty gradient. This confirms our core claim: only top-tier models achieve non-random L0 accuracy, making them the only ones capable of showing a statistically significant collapse at L1.

---

> ### Author Response · Authors · 2025-11-19
> **Q3 to Reviewer XXzx**
>
> ## Q3: Do improvements on SpatialViz-Bench transfer to downstream visuospatial tasks (e.g., CAD/DeepCAD VQA)?
>
> - **Direct Evaluation on CAD Data:** Our benchmark explicitly incorporates real engineering scenarios. Specifically, the **Three-View Projection (Level 1)** task is constructed using real engineering parts derived from the **DeepCAD** dataset. Therefore, performance on this subset of SpatialViz-Bench serves not just as a proxy, but as a **direct evaluation** of a model's capability to handle easy CAD VQA tasks (understanding orthogonal projections of engineering parts).
> - **Foundational Necessity:** While we did not run simulations for robotics or surgery, the spatial skills we test (Mental Rotation, Mental Animation) are widely recognized in cognitive science as the **prerequisites** for these downstream applications. If a model fails to mentally rotate a simple block (SpatialViz-Bench), it inherently lacks the capability to plan a robotic arm's trajectory in 3D space. Thus, our benchmark acts as a necessary 'unit test' for these downstream capabilities.

---

> ### Author Response · Authors · 2025-11-19
> **Q4 to Reviewer XXzx**
>
> ## Q4: Ablations on image resolution, number of views, and rendering style.
>
> Rather than varying these parameters to test sensitivity, we adopted a **standardized programmatic generation** strategy to **proactively eliminate** low-level visual ambiguities, ensuring the benchmark isolates spatial visualization capabilities.
>
> - **Standardization over Ablation:** We utilized a Python + FreeCAD pipeline to generate images with unified lighting, high contrast, and clear lines. This design choice is intentional: we aim to test the upper limits of **spatial reasoning**, not the robustness of visual encoders to noise or blur.
> - **Strategic View Selection:** The number of views was **carefully considered during the initial design phase** to ensure solvability while maintaining difficulty, rather than being arbitrarily determined:
>   - **Three-View Projection:** We intentionally provide limited isometric views with occlusion to force the model to rely on orthogonal projections for inference.
>   - **Cube Assembly:** The generation algorithm ensures that a single view is sufficient by enforcing physical stability rules (e.g., no floating cubes), allowing models to infer hidden supports.
>   - **Cross-Section:** We selected optimal camera angles to ensure all necessary geometric information is visible.
> - **Error Analysis Validation:** Our diagnostic analysis supports the effectiveness of this isolation. While "Perceptual Errors" account for 29.1% of failures, these are defined as high-level failures to interpret complex patterns (e.g., 2D grid patterns) or topological relationships, rather than failures to detect edges or low-level features. Given that modern MLLMs (e.g., Qwen2.5-VL, Gemini-2.5) support high-resolution inputs (our benchmark averages 2k resolution), low-level visual recognition is rarely the bottleneck.

---

> ### Author Response · Authors · 2025-11-19
> **Q5 to Reviewer XXzx**
>
> ## Q5: Analysis on controlled difficulty and its relation to performance of the models to identify where does it break.
>
> We conducted a detailed difficulty sensitivity analysis (see **Appendix F.1**, **Figure 4**, and **Tables 7-10**). Our data reveals a nuanced conclusion: the "breaking point" varies fundamentally by task type.
>
> - **Immediate Failure at Level 0 (The "Performance Floor"):** For visually complex tasks or those with obscure physical rules, most models hit a performance floor immediately at **Level 0**, showing no statistically significant performance drop because they fail from the start.
>   - **Block Moving:** This task proved universally difficult with **0 models** showing a significant difficulty gradient, rendering L0 and L1 performance effectively indistinguishable (e.g., Gemini-2.5-pro scored **27.50%** at L0, near random).
>   - **Cube Unfolding:** Unlike its inverse task (Reconstruction), only **1 model** showed sensitivity here, suggesting a widespread inability to reason about symmetry from unfolded views even at the simplest level.
> - **Collapse at Level 1 (The "Fragility Break"):** For tasks where models demonstrate competence at Level 0, we observed a clear breaking point at **Level 1** as spatial complexity increased.
>   - **Cube Reconstruction:** In stark contrast to Cube Unfolding, **12 models** exhibited a significant performance drop. Notably, Gemini-2.5-pro collapsed from **45.00%** (Level 0) to **10.00%** (Level 1).
>   - **3D Rotation:** While generally difficult, top-tier models (Gemini-2.5-pro, o1) achieved non-random L0 accuracy, allowing a statistically significant collapse to emerge at L1.
>   - **Cube Counting:** Gemini-2.5-pro's accuracy dropped significantly from **80.00%** (Level 0) to **52.50%** (Level 1) as constraints increased from two to three views.
>
> In summary, our analysis identifies two failure modes: 1) **Foundational Deficit**, where models fail immediately at L0 (e.g., Block Moving); and 2) **Fragility**, where models capable of solving L0 tasks collapse immediately when spatial dimensions or reasoning steps increase at L1 (e.g., Cube Reconstruction, Cube Counting).

---

> ### Author Response · Authors · 2025-11-19
> **Q6 to Reviewer XXzx**
>
> ## Q6: Report inter-annotator agreement ($\kappa$) for the six error categories.
>
> We have conducted a rigorous inter-annotator agreement analysis, which has been added to Appendix E.6.3 of the revised paper.
>
> - **Methodology:** Since our error analysis is a **multi-label classification task** (a single failure can stem from multiple error sources), a global Cohen's $\kappa$ is not directly applicable. Instead, we adopted a standard binary decomposition approach, treating each error category as an independent "Yes/No" classification3. Two authors independently annotated 100 randomly sampled failure cases.
> - **Results:** As shown in **Table 5**, the agreement levels are high across all categories.
>   - The **Methodological** category showed substantial agreement ($\kappa=0.75$).
>   - All other categories achieved almost perfect agreement ($\kappa > 0.81$), with **Calculation & Reasoning** reaching perfect consensus ($\kappa=1.00$).
>   - The macroscopic average Cohen's $\kappa$ is **0.88**, indicating an almost perfect level of consistency.
>
> | **Error Category**      | **Cohen's κ** |
> | ----------------------- | ------------- |
> | Perceptual              | 0.90          |
> | Spatial Transformation  | 0.81          |
> | Methodological          | 0.75          |
> | Instruction Following   | 0.96          |
> | Spatial Memorization    | 0.89          |
> | Calculation & Reasoning | 1.00          |
> | **Average**             | **0.88**      |
>
> This result quantitatively confirms the reliability and replicability of our error taxonomy and reinforces the validity of our diagnostic findings regarding model failure modes.

---

> ### Author Response · Authors · 2025-11-19
> **Q7 to Reviewer XXzx**
>
> ## Q7 (from Weakness 3): Why not use free-form evaluation like pose prediction?
>
> We employed MCA as a deliberate, pragmatic design choice to ensure scalability and feasibility, while proactively mitigating its limitations through diverse formatting and rigorous validation.
>
> - **Mitigation Strategies:** To prevent reliance on simple elimination heuristics, we diversified the question formats.
>   - **"All Wrong" Options:** 310 questions (approx. 26%) utilize an "A/B/C/All three other options are incorrect" format. This structure forces the model to perform genuine eliminative reasoning to rule out all distractors, rather than simply picking the best fit.
>   - **Mixed Formats:** We included 242 unique text answers and 120 numeric answers to prevent models from overfitting to a single multiple-choice style.
> - **Quantified Performance Gap:** We validated the difficulty gap between MCA and free-form generation in **Appendix F.2**. When converting the **Cube Counting** task from MCA to a fill-in-the-blank format, all models suffered significant performance drops (e.g., **GPT-4o: -32.50%**, **Gemini-2.5-pro: -14.17%**). This confirms that while MCA provides some scaffolding, the relative performance hierarchy remains consistent, and the gap authentically exposes deficits in precise reasoning.
> - **Technical Feasibility:** For core tasks like **3D Rotation** or **Paper Folding**, the "correct answer" is often a complex image. Requiring MLLMs to strictly generate these precise geometric visual outputs is a frontier research challenge beyond the scope of current evaluation capabilities. MCA allows us to assess the *mental simulation* of these transformations efficiently without being bottlenecked by the model's image generation limitations.

---

### Author Response · Authors · 2025-11-19
**Summary of Major Changes**

We sincerely thank all reviewers (XXzx, GoVB, 9vwG, nVip) for their constructive feedback and encouraging comments. Their insightful suggestions have been instrumental in strengthening the arguments and enhancing the persuasiveness of our work. Guided by these valuable inputs, we have revised the manuscript significantly to address the concerns regarding statistical rigor, contamination risks, and the root cause of CoT degradation.

&nbsp;

We have performed a significant revision based on the reviewers’ collective feedback to bolster the statistical rigor and analytical depth of our work. Most notably, to address concerns regarding **evaluation robustness** (Reviewers XXzx & nVip), we introduced 95% confidence intervals (Section 4.2.1, Para 2; Figure 4a) and conducted significance tests on difficulty gradients to quantify performance distinctness across levels (Section 4.2.1, Para 4; Figure 4b, c). We further added a comprehensive **robustness analysis** for prompting and extraction strategies (Section 4.2.3, Table 4), complemented by a rigorous inter-annotator agreement analysis (Appendix E.6.3, Table 5). In response to the critical discussion on the **"CoT Paradox"** (Reviewers XXzx, GoVB, 9vwG & nVip), we conducted paired significance tests for CoT and non-CoT (Table 3) and modified Section 4.2.2. To address the concern of **vision-dependence** (Reviewer GoVB), we empirically validated this by integrating a text-only control experiment (Table 2; Section 4.2.1, Para 1). Furthermore, we expanded Section 3.3 to clarify the **task construction** and refined Section 2, Para 2 and Figure 1 to ensure **objective** positioning against concurrent works.

---

### Meta-Review · Area_Chair_HDJc · 2025-12-24

**Summary:**

The paper presents SpatialViz-Bench, a benchmark designed to evaluate spatial visualization capabilities in MLLMs. It distinguishes itself through programmatic problem generation to mitigate contamination, and a focus on cognitive sub-abilities such as mental rotation and folding. The reviewers initially recognized the novelty and relevance of the topic, but raised significant concerns regarding the statistical rigor of the results, the robustness of the counter-intuitive finding that CoT prompting degrades open-source model performance, and the validity of the claims regarding cognitive grounding and vision dependence.

The authors provided a comprehensive rebuttal that included new statistical analyses (95% confidence intervals, paired significance tests), sensitivity studies on prompts and answer extraction, and a text-only control experiment. These additions largely support the benchmark’s discriminative power and strengthen the evidence for the CoT degradation effect. Given the methodological improvements and the explicit endorsement from Reviewer nVip after the revisions, the paper makes a solid contribution to the evaluation of spatial visualization in MLLMs.

**Reviewer Concerns:**

**Addressed Concerns:**

- **Statistical rigor:** Reviewer XXzx’s request for uncertainty estimates and significance testing was addressed. The authors added 95% Wilson confidence intervals and paired significance tests to better substantiate performance tiers and key comparisons.
- **Robustness of CoT findings:** Multiple reviewers (XXzx, GoVB, 9vwG, nVip) questioned whether the CoT degradation was an artifact of prompting or parsing. The authors conducted sensitivity analyses across prompt templates and extraction rules and added qualitative analysis. Overall, the revisions reduce the likelihood that the effect is purely a prompt or parsing artifact.
- **Vision dependence:** Reviewer GoVB’s concern that the benchmark might be solvable from text alone was addressed via a control experiment in which a text-only model achieved near-random performance.
- **Inter-annotator agreement:** Reviewer XXzx and nVip’s concerns about the reliability of the error taxonomy were addressed by reporting high inter-annotator agreement (k = 0.88).
- **Presentation and related work:** Reviewer nVip’s feedback on figure clarity and the tone and coverage of the related work section was incorporated into the revised manuscript.

**Outstanding Concerns:**

- **Mechanical System task contamination:** Reviewer GoVB noted that the Mechanical System task relies on public simulations, posing a higher contamination risk than the procedurally generated tasks. While the authors argue the questions are newly written and diversified, this component remains less secure than the fully programmatically generated portions.
- **Cognitive grounding:** Reviewer GoVB felt the cognitively grounded claim was overstated due to the lack of clearly articulated parametric difficulty scaling in the main text. While the authors clarified the underlying parameters in the appendix, the top-level presentation may still appear heuristic to some readers.

**Reviewer Scores:**

Reviewer XXzx: Score likely unchanged at 6. The added confidence intervals, significance tests, and sensitivity analyses directly address the main methodological requests. Remaining issues (transfer, multiple-choice framing, limited ablations on rendering factors) were not resolved with decisive new evidence, so an upgrade is not clearly warranted.

Reviewer GoVB: Score likely unchanged at 6. The rebuttal strengthens the paper on vision dependence, CoT robustness, and difficulty parameterization. However, skepticism about contamination risk for the manually curated mechanical system task, along with concerns about prompt standardization and selection effects, is only partially mitigated by clarification rather than definitive audits or new controls.

Reviewer 9vwG: Score likely unchanged at 8. The rebuttal improves confidence in the CoT finding via robustness checks and provides clearer discussion of contamination mitigation. However, requests for concrete contamination validation and deeper causal analysis are only partially addressed (some proposals and qualitative examples, but limited direct verification), so there is not enough new evidence to justify an upward adjustment.

Reviewer nVip: Score increased from 4 to 6 based on their explicit follow up after the revision. The reviewer already reflected this change.

---

### Decision · Program_Chairs · 2026-01-26

Accept (Poster)